# SLICEFINE: THE UNIVERSAL WINNING-SLICE HYPOTHESIS FOR PRETRAINED NETWORKS

## ABSTRACT

This paper presents a theoretical framework that explains why fine-tuning small, randomly selected subnetworks (*slices*) within pre-trained models is sufficient for downstream adaptation. We prove that pretrained networks exhibit a **universal winning slice** property, arising from two phenomena: (1) **spectral balance**—the eigenspectra of different weight matrix slices are remarkably similar—and (2) **high task energy**—their backbone representations (pretrained weights) retain rich, task-relevant features. This leads to the *Universal Winning Slice Hypothesis*, which provides a theoretical foundation for parameter-efficient fine-tuning (PEFT) in large-scale models. Inspired by this, we propose *SliceFine*, a PEFT method that uses this inherent redundancy by updating only selected slices of the original weights—introducing zero new parameters, unlike adapter-based approaches. Empirically, *SliceFine* matches the performance of SOTA PEFT methods across various language and vision tasks, while significantly improving training speed, memory efficiency, and model compactness. Our work bridges theory and practice, offering a theoretically grounded alternative to existing PEFT techniques.

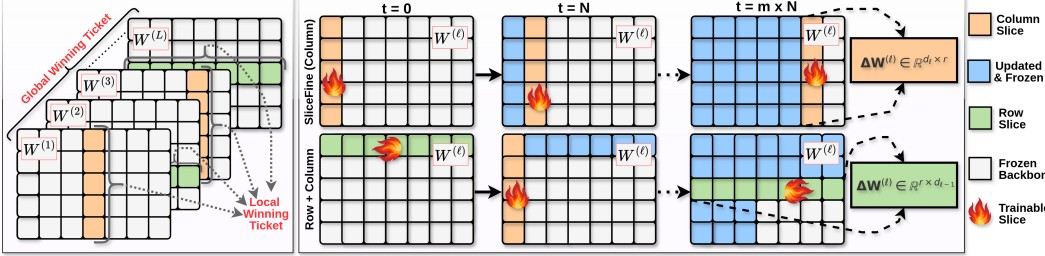

Figure 1: **(Left) Winning Tickets.** In a pretrained network, a *randomly chosen slice* of a layer $W^{(\ell)} \in \mathbb{R}^{d_\ell \times d_{\ell-1}}$ *acts as a local winning ticket*: tuning only that slice lowers the loss while keeping the backbone frozen. A few such slices (row, column, or row-column) selected across layers constitute a *global winning ticket*. **(Right) SliceFine.** At step $t$, only a slice of the weight matrix $W^{(\ell)}$ is updated; all other entries remain fixed. Every $N$ steps, we activate a new slice at a different position for learning; the previously active slice retains its learned update but is frozen. Top: *column sweep*—the slice slides across columns. Bottom: *row–column alternation*—the slice alternates between a column block and a row block to cover complementary directions. Similarly, In *row sweep*—the slice slides across rows. This schedule updates only a tiny portion of the model at a time while gradually covering many regions; applying it across several layers yields a global winner.

## 1 INTRODUCTION

Large pretrained models work well across many tasks, yet we still lack a simple picture of *why* small changes are often enough to adapt them. Popular parameter-efficient fine-tuning (PEFT) methods add small modules (e.g., adapters, low-rank layers) and train only those parts while freezing the rest (Hu et al., 2021; Kowsher et al., 2024b; Zhang et al., 2023b; Liu et al., 2024b; Prottasha et al., 2025). This raises a basic question: why do these small modules work, and do we actually need to *add* new parameters—or is there already enough useful structure inside the pretrained weights themselves?

This paper addresses this question by introducing the *Universal Winning Slice Hypothesis*, which provides a simple, testable explanation for fine-tuning behavior in large-scale pretrained models and

guides a very light form of adaptation. First, we briefly fix terminology and state the *Universal Winning–Slice Hypothesis*.

**Terminology.** Let $f_{\theta_0} : \mathbb{R}^d \to \mathbb{R}^k$ denote a pretrained neural network of depth $L$, where each layer $\ell$ is parameterized by a weight matrix $W^{(\ell)} \in \mathbb{R}^{d_\ell \times d_{\ell-1}}$. A *slice* of a layer $l$ is a contiguous set of rows or columns of a single weight matrix $W^{(\ell)}$, specified by a mask $M^{(\ell)} \in \{0,1\}^{d_\ell \times d_{\ell-1}}$. A *winning slice* (local winner) is a slice whose update alone reduces the loss at $\theta_0$. A *slice set* $\mathcal{T} = \{M_i\}_{i=1}^m$ is a small collection of slices across layers. When the joint update over $\mathcal{T}$ attains near full fine-tuning performance, $\mathcal{T}$ is a *global winning ticket*. We call $r$ the *slice rank* (number of selected columns or rows). *Slice rank is not the matrix rank of $W^{(\ell)}$;* it measures the slice *width* (capacity).

**Universal Winning–Slice Hypothesis (UWSH).** *In a dense, pretrained network, any random slice with sufficient width has a local winning ticket: training only that slice while freezing the rest improves downstream performance (a local winner). Moreover, tuning a small set of such slices across layers can match full fine-tuning accuracy while updating far fewer parameters (a global winner).*

This view differs from the Lottery Ticket Hypothesis (LTH) (Frankle & Carbin, 2019; Yu et al., 2021; Chen et al., 2020), which posits a *special*, sparse subnetwork ("winning ticket") that must be found by pruning and then trained—often from its original initialization—to match full–network performance. Most analyses of LTH rely on carefully identifying such subnetworks via iterative pruning by randomly (Bai et al., 2022) or importance-based selection (You et al., 2022). In contrast, we study large-scale, dense, pretrained networks and establish a theoretical framework showing that a random simple *slice* can serve as a winning ticket. Consequently, our goal is to fine-tune only these slices—without adding new parameters—as a PEFT strategy for adapting pretrained models to downstream tasks. The reason every sufficiently wide slice can be a winning ticket is twofold: First, *spectral balance*. Take any layer and split its weight matrix into simple groups of rows or columns (slices). In Figure 2, if we look at each group's covariance and its eigenvalues, we find that the average size of the eigenvalues and the way they decay are very similar across groups, implying that all slices have comparable capacity for fine-tuning. Second, *high task energy*. After pretraining, the backbone features of a model already line up a large share of their variation with directions that are useful for downstream tasks. This shows up as high cumulative explained variance of the top PCA components of the centered features (details in Lemma 2.5), or, equivalently, a "lazy" NTK spectrum where a few top directions dominate. When most of the useful signal lives in a small set of directions, a small slice that has enough rank will inevitably touch those directions and can move the model in the right way.

Together, these two facts give a simple rule of thumb. Because slices are balanced in strength, *which slice we pick is not critical*. Because the backbone already concentrates task energy, a slice in $W^{(\ell)}$ overlaps with the task-relevant subspace, produces a nonzero restricted gradient, and reduces the loss. This picture also clarifies common PEFT observations: adapters help because the backbone already carries most of what is needed (see Corollary E.2).

*As every sufficiently wide slice in $W^{(\ell)}$ has a local winning ticket, can we fine-tune* only *a slice?* Based on the UWSH, we propose ***SliceFine***, which trains a set of slices $\mathcal{T}$ across different layers (to get a global winning ticket) while keeping the backbone frozen and **adds no new parameters**. The per-layer cost scales with the slice rank, and in practice very small ranks—often $r=1$—already match full fine-tuning or strong PEFT baselines on downstream tasks. Although Figure 12 shows good accuracy on downstream tasks when training only a fixed slice in $W^{(\ell)}$, to cover more directions over time we freeze the active slice after a fixed number of steps $N$ and then activate a new slice at a different position, effectively performing block-coordinate descent across the layer. In practice, we do not need to traverse all possible positions; switching the active slice every $N = 500$ training steps and making only 5–10 switches in total achieves optimal performance across all of our experiments. Figure 1 (right top) illustrates a *column-slice* schedule: at $t=0$ the first column group is trained; after $N$ steps the slice shifts to the next column group, and so on. Figure 1 (right bottom) shows an *alternating* schedule: at $t=0$ a column slice is trained; at $t=N$ the slice switches to a row group; subsequent shifts continue alternating between columns and rows.

**Our contributions are summarized as follows**: (i) A theoretical and testable hypothesis, **UWSH**: in pretrained networks, an arbitrary sufficiently wide slice is a *local winning ticket*; multiple slices across layers form a *global winning ticket*. (ii) A PEFT method *SliceFine* by following UWSH: trains only a set of slice $\mathcal{T}$ across different layers, with no new parameters. (iii) Broad empirical study

across language and vision showing accuracy on par with—or better than—strong PEFT baselines, with lower memory, faster throughput, and smaller model footprints. (iv) Ablations that connect rank choice, slice selection, switching interval, initialization, and backbone pruning to the simple picture above, using explained variance/NTK laziness as a guide.

## 2 WINNER SLICES IN PRETRAINED NETWORKS

Let $Y \in \mathbb{R}^{c \times n}$ be the label encoding matrix for $n$ downstream examples across $c$ classes, and let $P_Y \in \mathbb{R}^{n \times n}$ be the orthogonal projector onto the column space of $Y$. The *task covariance* at layer $\ell$ is $\Sigma_{\text{task}} := \frac{1}{n} \Phi_\ell P_Y \Phi_\ell^\top$ and task subspace $k_{\text{task}} := \text{rank}(\Sigma_{\text{task}})$.

Consider a network $f_{\theta_0}$ is *pretrained* on a large dataset so that, for each layer $\ell$, the intermediate representations $\phi_\ell(x) \in \mathbb{R}^{d_\ell}$ satisfy: $\text{rank}(\Phi_\ell) = \text{rank}([\phi_\ell(x_1), \ldots, \phi_\ell(x_n)]) \geq k_{\text{task}}$, where $\Phi_\ell \in \mathbb{R}^{d_\ell \times n}$ is the matrix of downstream representations and $k_{\text{task}}$ is the intrinsic dimension of the downstream task in the representation space at layer $\ell$, i.e., the smallest number of orthogonal directions needed to achieve perfect linear classification of the labels.

Formally, the slice parameters $M^{(\ell)} \odot W^{(\ell)}$ (where $\odot$ denotes element-wise multiplication) are trainable, while all other entries remain fixed at their pretrained values. *Unlike pruning in LTH (Yu et al., 2021; Chen et al., 2020), the frozen weights are not zeroed out; they continue to contribute during forward propagation, providing a representational scaffold that allows the trainable slice to adapt effectively.* The learned increment is $\Delta W^{(\ell)} = M^{(\ell)} \odot U^{(\ell)}$, where $U^{(\ell)}$ denotes the gradient-driven update. Given a downstream dataset $D = \{(x_i, y_i)\}_{i=1}^n$ and convex per-sample loss $\ell$, the fine-tuning objective is: $\mathcal{L}(\theta) = \frac{1}{n} \sum_{i=1}^n \ell(f_\theta(x_i), y_i)$.

Let $U_{k_{\text{task}}} \in \mathbb{R}^{d_\ell \times k_{\text{task}}}$ denote the top $k_{\text{task}}$ left singular vectors of $\Phi_\ell$, spanning the task-relevant subspace. The projection operator is $P_{U_{k_{\text{task}}}} = U_{k_{\text{task}}} U_{k_{\text{task}}}^\top$. The quantity $k_{\text{task}}$ captures the minimum number of feature-space directions required for perfect linear separation of the downstream labels. In *SliceFine*, we must determine whether each slice of a pretrained weight matrix overlaps enough with the $k_{\text{task}}$-dimensional task subspace to reduce downstream loss. The lemma below shows that, in pretrained networks $f_{\theta_0}$, slices from the $W^{(\ell)}$ have similar average spectral energy and comparable eigenvalue decay—a property we call *spectral balance*.

**Lemma 2.1** (Spectral Balance Across Slices). *Consider a pretrained layer $W^{(\ell)}$ partitioned into $k$ disjoint groups $\{W_g\}_{g=1}^k$. Let $\Sigma_g := W_g W_g^\top$ and let $\lambda_1(\Sigma_g) \geq \cdots \geq \lambda_{d_\ell/k}(\Sigma_g)$ denote its eigenvalues in descending order. Although each group is* anisotropic—*its eigenvalues decay from large to small—the groups exhibit* spectral balance *in the sense that:* $\frac{\frac{1}{d_\ell/k} \sum_{i=1}^{d_\ell/k} \lambda_i(\Sigma_g)}{\frac{1}{d_\ell/k} \sum_{i=1}^{d_\ell/k} \lambda_i(\Sigma_{g'})} \approx 1, \quad \forall g, g' \in \{1, \ldots, k\},$ *and their decay profiles are boundedly similar:* $\max_{1 \leq i \leq d_\ell/k} \frac{|\lambda_i(\Sigma_g) - \lambda_i(\Sigma_{g'})|}{\lambda_i(\Sigma_{g'})} \leq \rho,$ *for some small $\rho \ll 1$ independent of $g, g'$. Thus, no group is disproportionately "weak" or "strong" in average spectral energy, and each slice retains non-trivial overlap with the task-relevant subspace.*

This property guarantees that no slice is degenerate with respect to the task subspace: all slices retain comparable spectral energy and a non-zero projection onto task-relevant directions. Consequently, each slice admits a non-zero restricted gradient with respect to the downstream loss and can independently reduce the loss—i.e., it qualifies as a *local winning slice*.

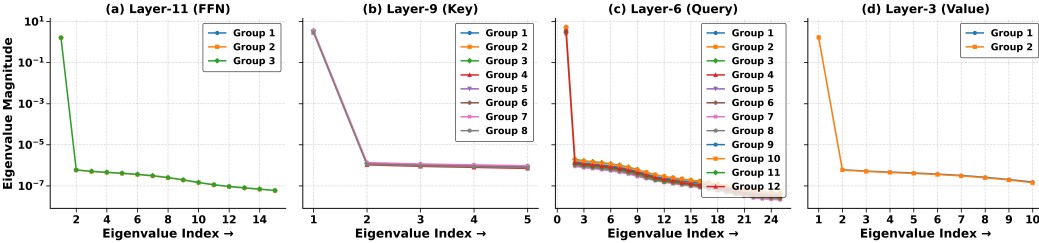

Figure 2: Eigenvalue spectra of FFN, Key, Query, and Value weight matrices from different layers of a pretrained RoBERTa-base model. For each matrix, weights are partitioned into groups, and the eigenvalues of the within-group covariance $\Sigma_g = W_g^{(\ell)} W_g^{(\ell)\top}$ are plotted in descending order.

Figure 2 shows that, although each group's eigen-spectrum is anisotropic—dominated by a few large eigenvalues—the decay profiles and the average spectral energy are nearly identical across groups. Groups are constructed by partitioning the weight matrix into $n$ equal blocks: for a row-wise partition (first two figures) we split the $d_\ell$ rows into $n$ groups (each of rank $r = d_\ell/n$), and for a column-wise partition (last two figures) we split the $d_{\ell-1}$ columns into $n$ groups (each of rank $r = d_{\ell-1}/n$). Empirically, the average inter-group variance metric $\rho$ computed across all groups and eigen-components is $0.00125$, $0.00192$, $0.00255$, and $0.00102$ respectively, indicating negligible dispersion. This supports the *spectral balance* property of Lemma 2.1, implying that no group is disproportionately weak or strong; the detailed proof is in Appendix A.

Spectral balance is the key precondition that motivates the following two definitions. We first define a *local winning slice* to capture the ability of a slice to reduce downstream loss, and then a *global winning ticket* to describe a set of slices $\tau$ across different layers whose joint optimization achieves full fine-tuning performance.

**Definition 2.2** (Local Winner). A mask $M$ is a *local winning slice* if there exists an update $U$ such that $\mathcal{L}(\theta_0 + M \odot U) \leq \mathcal{L}(\theta_0) - \delta$ for some $\delta > 0$, and the restricted gradient is non-zero:

$$\|\nabla_M \mathcal{L}(\theta_0)\|_2 = \left\| \frac{\partial \mathcal{L}(\theta)}{\partial (M \odot W^{(\ell)})} \Big|_{\theta=\theta_0} \right\|_2 > 0.$$ Under spectral balance, every slice has comparable spectral energy and retains overlap with the task subspace, ensuring its restricted gradient is non-zero and enabling it to independently reduce loss.

**Definition 2.3** (Global Winner). A set of masks $\mathcal{T} = \{M_i\}_{i=1}^m$ forms a *global winning ticket* if there exist updates $\{U_i\}_{i=1}^m$ such that $\mathcal{L}\left(\theta_0 + \sum_{i=1}^m M_i \odot U_i\right) \leq \epsilon$, for some $\epsilon > 0$, achieving performance close to full fine-tuning while updating only a small fraction of weights.

While the local winning slice definition ensures each slice can make independent progress, the global winning ticket definition captures the compositional effect: if different slices contribute complementary directions in the task subspace, their combined span can cover the full $k_{\text{task}}$ subspace, matching full fine-tuning performance.

**Theorem 2.4** (Universal Winning Ticket). *Let $f_{\theta_0}$ be a pretrained network of depth $L$. For each layer $\ell$, let $\Phi_\ell = [\phi_\ell(x_1), \ldots, \phi_\ell(x_n)]$ have singular values $\sigma_1 \geq \sigma_2 \geq \ldots$ satisfying $\frac{\sum_{j=1}^{k_{\text{task}}} \sigma_j^2}{\sum_j \sigma_j^2} \geq \eta$, for some $\eta \in (0,1)$. Assume further that every slice vector $w_j$ of $W^{(\ell)}$ has non-trivial projection into the top-$k_{\text{task}}$ subspace: $\|P_{U_{k_{\text{task}}}} w_j\|_2^2 \geq \gamma > 0$.*

*Then: 1. (Local winners) For any binary mask $M$ selecting a slice of weights in layer $\ell$, if the restricted Jacobian $J_M(x) := \nabla_{M \odot W^{(\ell)}} f_{\theta_0}(x)$ satisfies $\text{rank}(J_M) \geq k_{\text{task}} > 0$, then the gradient on the slice is nonzero: $\left\| \frac{\partial \mathcal{L}(\theta)}{\partial (M \odot W^{(\ell)})} \right\|_F \Big|_{\theta=\theta_0} > 0$. Consequently, there exists a perturbation $U$ supported on $M$ such that, for sufficiently small $\eta > 0$, $\mathcal{L}(\theta_0 + \eta\, M \odot U) < \mathcal{L}(\theta_0) - \delta$, making the slice a local winner. This follows from spectral balance and projection properties, which ensure $J_M$ retains task-relevant directions.*

*2. (Global tickets) There exists a collection $\{M_i\}_{i=1}^m$, with $m \ll \sum_\ell d_\ell$, such that joint or sequential optimization yields $\mathcal{L}\left(\theta_0 + \sum_{i=1}^m M_i \odot U_i\right) \leq \epsilon$, for arbitrarily small $\epsilon$. This holds because the combined Jacobians $\dim \text{span}(J_{M_1}(x), \ldots, J_{M_m}(x)) \to k_{\text{task}}$, incrementally span the task-relevant subspace, enabling near full fine-tuning performance while updating only a small subset of weights.*

Theorem 2.4 follows naturally: spectral balance ensures that *any* slice is a local winner (Part 1), while the finite rank $k_{\text{task}}$ determines the number of slices required to assemble a global ticket (Part 2). Details of the proof are provided in Appendix B.

We now examine how large a slice must be to act as a winner. Empirically, our ablation study (Appendix F.1) shows that training a *slice* with rank one ($r = 1$ i.e., updating a single column or a single row of a pretrained weight matrix) is often sufficient. Theoretically, we find that this rank requirement is determined by the spectral structure of the representation space, which can be understood via the relationship between the linearized NTK kernel and the PCA decomposition of the features.

We observe a strong dependence on the domain. When the feature spectrum exhibits high cumulative explained variance (CEV) and the model remains in a lazy NTK regime after fine-tuning, a slice rank $r$ suffices to capture the task directions. Conversely, when the spectrum has lower CEV and the NTK is less lazy, larger slices are needed to cover enough task-relevant directions. Since, in the linearized regime, the NTK spectrum matches the PCA spectrum of the centered features (Lemma 2.5), the PCA curve gives a direct guide: a steeper curve suggests a slice suffices; a flatter curve calls for a larger slice.

**Lemma 2.5** (PCA Decomposition of the Representation/Linearized NTK Kernel). *Let $H \in \mathbb{R}^{n \times d}$ be hidden representations and $\tilde{H} := H - \frac{1}{n}\mathbf{1}\mathbf{1}^\top H$ the centered features. Let $\Sigma := \frac{1}{n-1}\tilde{H}^\top \tilde{H}$ with eigendecomposition $\Sigma = V \Lambda V^\top$, where $V \in \mathbb{R}^{d \times r}$ has orthonormal columns and $\Lambda = \mathrm{diag}(\lambda_1, \ldots, \lambda_r)$ with $\lambda_i > 0$.*

*Define $P := \tilde{H}V \in \mathbb{R}^{n \times r}$. Then the (centered) feature Gram matrix (a.k.a. linearized representation kernel) $K := \tilde{H}\tilde{H}^\top \in \mathbb{R}^{n \times n}$ admits the exact decomposition $\boxed{K = PP^\top}$.*

*Moreover, the nonzero eigenvalues of $K$ are $(n-1)\lambda_1, \ldots, (n-1)\lambda_r$. Consequently, for any $k \leq r, \frac{\sum_{i=1}^k \lambda_i}{\sum_{j=1}^r \lambda_j} = \frac{\sum_{i=1}^k \mu_i}{\sum_{j=1}^r \mu_j}, \qquad \mu_i := (n-1)\lambda_i$, so PCA explained-variance ratios in feature space match those of $K$.*

**Corollary 2.6** (Minimal Slice Rank from PCA/NTK Spectrum). *Let $k_{\mathrm{task}}(\tau)$ be the smallest integer $m$ such that $\sum_{i=1}^m \lambda_i / \sum_{j=1}^r \lambda_j \geq \tau$ for a threshold $\tau \in (0, 1]$. If the slice rank $r_{\mathrm{slice}} \geq k_{\mathrm{task}}(\tau)$, then with high probability the slice intersects the task-relevant subspace, and thus can serve as a local winner. For familiar domains, $k_{\mathrm{task}}(\tau)$ is small due to a steep PCA/NTK spectrum; for unfamiliar domains, it is larger due to a flatter spectrum.*

The proof of Lemma 2.5 is given in Appendix C. Additional discussion and empirical evidence connecting slice rank, NTK laziness, and PCA variance profiles are provided in Appendix D.

Now we turn our attention to analyzing how much *task energy* the frozen backbone already carries for an $r$-rank slice to be a winner. We measure this with the PCA/NTK cumulative explained variance of the features: when the CEV is high, most of the feature energy lies in a small set of directions that separate the labels (the "task subspace"). From a small calibration set, estimate $k_{\mathrm{task}}(\tau)$: the fewest principal components that explain at least a fraction $\tau$ of the feature variance. Then set the slice size $r_{\mathrm{slice}} \geq k_{\mathrm{task}}(\tau)$. Under the same spectral-balance assumption as in Lemma E.1, any slice of this size has nonzero overlap with the task subspace, so its gradient is nonzero and the loss decreases when we update it. In short: if the backbone shows high task energy, a slice that is large enough will train well. This also explains the failure modes. If we weaken the backbone (e.g., heavy pruning or strong domain shift), the CEV curve becomes flatter, the effective task subspace grows, and the slice's overlap shrinks. Lemma E.1 then predicts smaller gradients and smaller guaranteed improvement. To recover performance we must increase $r_{\mathrm{slice}}$ or combine several slices until their total span reaches $k_{\mathrm{task}}(\tau)$. These trends match our experiments: familiar domains (high CEV) work with very small slices, while unfamiliar or degraded backbones require larger slices (Details in Appendix E).

# 3 SLICEFINE: SLICE AS EFFICIENT FINE-TUNING

Section 2 established two key facts: (i) *spectral balance*—different row/column groups in a pretrained layer have similar spectral strength—and (ii) *high task energy*—the frozen features already concentrate variance along task-relevant directions. Theorem 2.4 implies a simple recipe: every slice whose rank satisfies $r_{\mathrm{slice}} \geq k_{\mathrm{task}}(\tau)$ has a winning ticket. This observation directly motivates a practical PEFT method *SliceFine*, that trains one small slice at a time and moves it across positions to accumulate task-aligned directions. A slice is a row or column mask of rank $r$ applied to $W^{(\ell)} \in \mathbb{R}^{d_\ell \times d_{\ell-1}}$. At step $t$, only entries in the active mask $M_\ell(t)$ are trainable and we apply an increment $U_t^{(\ell)}$ supported on $M_\ell(t)$, $W^{(\ell)} \leftarrow W^{(\ell)} + M_\ell(t) \odot U_t^{(\ell)}$, leaving all non–masked entries fixed at their pretrained values (plus any past increments when they were previously active). After every $N$ steps in Figure 1, we *move* the mask to a new position, i.e., $M_\ell(t + N) \in \mathcal{M}_\ell$ where $\mathcal{M}_\ell$ is the set of admissible row/column slices of rank $r$; in practice we use a simple cyclic sweep (left–to–right for row, top–to–bottom for column), though

random or coverage–aware choices are also possible (Appendix L). Over $K$ distinct positions $\{M_{\ell,i}\}_{i=1}^{K}$, the accumulated update equals $\Delta W^{(\ell)} = \sum_{i=1}^{K} M_{\ell,i} \odot U_i^{(\ell)}$, and the linearized effect is $f_{\theta_0+\Delta\theta}(x) \approx f_{\theta_0}(x) + \sum_{i=1}^{K} J_{M_{\ell,i}}(x) \, \text{vec}(U_i^{(\ell)})$, i.e., block–coordinate descent over Jacobian blocks. Empirically, we apply *SliceFine* to all linear layers or to selected layers (e.g., nn.Linear in PyTorch) to obtain a global winning ticket, as detailed in Table 4.

*SliceFine* has three design knobs: *rank $r$*, *slice selection*, and *switching interval $N$*. Rank controls capacity; by Corollary 2.6, taking $r \geq k_{\text{task}}(\tau)$ ensures the active slice intersects the task subspace, yielding a nonzero restricted gradient and a guaranteed decrease (Lemma E.1). We estimate $k_{\text{task}}(\tau)$ once from frozen features on a small calibration set; empirically, very small rank (often $r{=}1$) is sufficient for efficient fine-tuning.

For slice selection, spectral balance implies robustness to position: any row or column slice of rank $r$ behaves similarly. We therefore initialize the mask at the first (index-0) position and perform a deterministic sweep across positions, which simplifies implementation and makes the training dynamics easy to track. We also verify that purely random mask placements perform comparably (Appendix L). The interval $N$ trades off per–slice adaptation and coverage: small $N$ explores more positions but can slow early convergence; large $N$ adapts deeper on each slice but delays coverage.

Our ablation F.2 shows a broad, task–dependent sweet spot (e.g., $N \approx 100$–$500$ for STS–B/QNLI (Wang et al., 2018) and $N \approx 500$–$1500$ for MRPC/SST–2 (Wang et al., 2018)), consistent with the theory that each visited slice contributes additional task–aligned directions until the combined span matches the task dimension. Computationally, per–layer cost scales as $O(d_\ell \times r)$ (row) or $O(d_{\ell-1} \times r)$ (column) with zero auxiliary parameters, and optimizer state is maintained only for active entries, yielding favorable memory and throughput characteristics (Details in Section 4). The implementation details are presented in the Appendix M.

## 4 EXPERIMENTS

In this section, we conduct experiments across a diverse set of downstream tasks spanning text, image, and video, thereby covering commonsense reasoning, mathematical reasoning, natural language understanding, image classification, and video action recognition. This broad evaluation design allows us to examine whether the theoretical guarantees of the Winner Slice Hypothesis translate into consistent practical gains across modalities. We report results for six variants that differ only in slice rank and orientation: *SliceFine-1R*, *SliceFine-1C*, *SliceFine-1RC*, *SliceFine-5R*, *SliceFine-5C*, *SliceFine-5RC*. Here "1R/5R" denotes a *row* slice of rank $r \in \{1, 5\}$; "1C/5C" denotes a *column* slice of the same rank; and "1RC/5RC" alternates row and column slices with the same total rank per switch. Additionally, Table 4 reports performance across slice ranks $r \in \{1, 2, 4, 8, 32, 64, 128\}$ (see Appendix F.3 for details) and Figure 3(a) shows increasing higher rank tends to gradually overfit. Across all tables, the best result is highlighted in blue and the second best in orange. For baseline LoRA-type methods such as LoRA, VeRA, AdaLoRA, DoRA, BoNE, RoCoFT, etc., we use a rank of 5 and apply each PEFT adapter to all linear weight matrices (i.e., nn.Linear). For prompt- and prefix-based PEFT methods, we use five virtual tokens. All other hyperparameters—including initialization strategies, warm-start options, optimizer settings, and placement configurations—follow the recommendations provided in each method's original paper, ensuring that every baseline is evaluated under its optimal or standard configuration. Appendix J outlines the full set of hyperparameters and training protocols. The complete experimental setup is described in detail in the appendices. Appendix H provides dataset descriptions, Appendix G lists all baseline PEFT methods, and Appendix I details the backbone models used in each modality.

**Main Results.** On commonsense and mathematical reasoning with *LLaMA-3B*, *SliceFine* (Table 1) matches or surpasses strong PEFT baselines—including LoRA, AdaLoRA, and RoCoFT—while using fewer trainable parameters. For example, *SliceFine-5RC* attains the highest average score (82.13%) in math reasoning, outperforming AdaLoRA by +1.07 points while requiring significantly fewer trainable parameters. Even rank one column or row slice consistently surpasses lighter baselines such as BitFit and Prefix Tuning.

On visual downstream tasks (Table 2), SliceFine matches or outperforms strong low-rank baselines such as VeRA and AdaLoRA, while updating only 0.08M–0.41M parameters. For instance, on VTAB-1K image classification using ViT-Base-Patch16-224, *SliceFine-5R* reaches 88.85% av-

| LLM | Method | #TTPs | Commonsense Reasoning | | | | | | | | | Math Reasoning | | | | | |
|---|---|---|---|---|---|---|---|---|---|---|---|---|---|---|---|---|---|
| | | | BoolQ | PIQA | SIQA | H.Sw. | W.Gra. | ARCe | ARCc | OBQA | Avg. | M.Ar. | G.8K | A.S. | Se.Eq | S.MP | Avg. |
| LLaMA-3₃ᵦ | Prefix | 38.09 | 70.85 | 81.91 | 78.66 | 79.98 | 75.11 | 74.40 | 59.88 | 73.08 | 74.23 | 87.29 | 71.66 | 84.24 | 82.93 | 54.20 | 76.06 |
| | AdaLoRA | 33.05 | 71.05 | 82.11 | 82.27 | 91.76 | 79.31 | 79.17 | 61.91 | 77.14 | 77.71 | 91.57 | 79.68 | 89.22 | 83.97 | 63.51 | 81.41 |
| | VeRA | 1.50 | 69.32 | 81.20 | 80.39 | 91.15 | 79.47 | 78.26 | 62.02 | 76.73 | 77.32 | 92.16 | 78.36 | 88.91 | 84.48 | 61.71 | 81.07 |
| | LoRA | 13.77 | 71.25 | 82.01 | 79.47 | 91.96 | 80.29 | 79.83 | 62.22 | 77.55 | 78.12 | 91.47 | 78.15 | 84.99 | 84.96 | 62.02 | 80.32 |
| | RoCoFT | 6.90 | 72.47 | 82.62 | 80.79 | 91.67 | 79.78 | 79.17 | 62.22 | 79.37 | 78.51 | 91.38 | 80.49 | 88.51 | 83.46 | 63.13 | 81.55 |
| | HRA | 6.25 | 72.27 | 82.82 | 80.29 | 91.97 | 79.37 | 79.07 | 62.52 | 79.27 | 78.44 | 92.18 | 80.29 | 89.42 | 83.36 | 62.83 | 81.62 |
| | *SliceFine-1R* | 1.25 | 70.88 | 81.16 | 79.12 | 90.19 | 78.04 | 78.58 | 61.95 | 78.73 | 77.33 | 91.65 | 79.85 | 88.36 | 84.83 | 62.55 | 81.45 |
| | *SliceFine-1C* | 1.25 | 70.94 | 81.08 | 78.95 | 90.98 | 78.00 | 77.56 | 61.14 | 77.70 | 77.04 | 91.44 | 78.80 | 87.21 | 82.72 | 61.73 | 80.38 |
| | *SliceFine-1RC* | 1.25 | 71.02 | 82.46 | 81.26 | 92.61 | 80.28 | 79.80 | 61.93 | 79.18 | 78.66 | 92.12 | 80.11 | 89.77 | 82.18 | 63.55 | 81.55 |
| | *SliceFine-5R* | 6.25 | 71.96 | 82.40 | 81.20 | 92.58 | 80.23 | 79.77 | 62.88 | 78.92 | 78.74 | 92.15 | 81.05 | 89.70 | 84.11 | 63.50 | 82.08 |
| | *SliceFine-5C* | 6.25 | 72.49 | 82.86 | 80.97 | 91.97 | 79.71 | 79.82 | 62.47 | 78.40 | 78.73 | 92.11 | 80.52 | 89.11 | 83.55 | 63.08 | 81.72 |
| | *SliceFine-5RC* | 6.25 | 72.01 | 82.45 | 81.25 | 92.64 | 80.27 | 79.82 | 62.92 | 78.97 | 78.79 | 92.10 | 81.10 | 89.75 | 84.16 | 63.53 | 82.13 |

Table 1: Commonsense and math reasoning with LLaMA-3B. SliceFine (shaded) rivals or surpasses baselines such as LoRA and AdaLoRA while using far fewer trainable parameters (#TTPs).

| PEFT | Image | | | | | | | | | Video | | | | |
|---|---|---|---|---|---|---|---|---|---|---|---|---|---|---|
| | Caltech | Flowers | Pets | Camel. | Euro. | Retino. | KITTI | Avg | #TTPs | UCF101 | Kinetics | HMDB | Avg | #TTPs |
| Full | 89.92 | 97.41 | 85.87 | 81.65 | 88.12 | 73.62 | 77.93 | 84.93 | 85.83 | 92.30 | 55.23 | 65.79 | 74.99 | 86.65 |
| VeRA | 91.53 | 99.19 | 91.04 | 86.45 | 92.97 | 74.25 | 77.92 | 87.62 | 0.240 | 92.28 | 57.21 | 66.77 | 72.09 | 0.242 |
| BitFit | 90.58 | 98.93 | 91.06 | 86.73 | 93.07 | 74.39 | 79.26 | 87.72 | 0.101 | 92.38 | 57.31 | 66.87 | 72.19 | 0.104 |
| DiffFit | 90.26 | 99.24 | 91.73 | 86.79 | 92.53 | 74.46 | 81.01 | 88.00 | 0.148 | 92.80 | 57.73 | 67.29 | 72.61 | 0.150 |
| SHiRA | 88.72 | 99.14 | 89.64 | 81.85 | 91.93 | 74.28 | 80.57 | 86.59 | 0.667 | 91.82 | 57.75 | 63.31 | 70.96 | 0.668 |
| LayerNorm | 89.79 | 99.13 | 91.41 | 86.29 | 92.88 | 74.82 | 78.16 | 87.50 | 3.041 | 92.16 | 57.09 | 66.65 | 71.97 | 3.867 |
| MiSS | 89.83 | 99.18 | 91.48 | 86.74 | 91.59 | 73.21 | 85.64 | 88.24 | 0.667 | 93.18 | 58.11 | 67.67 | 72.99 | 0.668 |
| Propulsion | 91.55 | 99.25 | 91.44 | 86.23 | 94.77 | 72.36 | 80.27 | 87.96 | 0.165 | 93.47 | 57.40 | 66.96 | 72.61 | 0.166 |
| LoHA | 92.13 | 99.28 | 91.07 | 85.84 | 93.38 | 73.25 | 80.83 | 87.97 | 1.667 | 93.63 | 57.56 | 67.12 | 72.77 | 1.669 |
| DoRA | 91.86 | 99.27 | 91.08 | 85.88 | 91.42 | 75.28 | 80.46 | 87.89 | 0.834 | 92.84 | 57.77 | 67.33 | 72.65 | 0.836 |
| Bone | 92.14 | 99.23 | 91.05 | 86.28 | 92.83 | 73.71 | 80.91 | 88.02 | 0.414 | 93.82 | 57.75 | 67.31 | 72.96 | 0.415 |
| RoCoFT | 92.56 | 99.31 | 91.19 | 87.84 | 93.54 | 74.27 | 79.63 | 88.33 | 0.415 | 93.14 | 58.07 | 67.63 | 72.95 | 0.417 |
| HRA | 92.16 | 99.32 | 91.36 | 86.74 | 93.82 | 74.87 | 78.18 | 88.06 | 0.491 | 92.72 | 57.65 | 67.21 | 72.53 | 0.493 |
| LoRA | 92.03 | 99.18 | 90.92 | 87.73 | 92.65 | 74.23 | 80.42 | 88.08 | 0.833 | 93.88 | 57.81 | 67.37 | 73.02 | 0.835 |
| Ada-LoRA | 91.62 | 99.24 | 91.18 | 87.54 | 92.37 | 73.84 | 79.94 | 87.96 | 2.011 | 94.43 | 57.66 | 67.22 | 73.04 | 2.027 |
| *SliceFine-1R* | 91.64 | 99.12 | 91.26 | 86.88 | 93.83 | 74.18 | 80.11 | 88.15 | 0.084 | 93.14 | 57.16 | 66.29 | 72.20 | 0.087 |
| *SliceFine-1C* | 91.78 | 99.72 | 91.12 | 86.63 | 94.24 | 74.10 | 79.73 | 88.19 | 0.084 | 93.35 | 57.72 | 66.31 | 72.46 | 0.087 |
| *SliceFine-1RC* | 91.74 | 99.19 | 91.51 | 86.41 | 94.70 | 74.46 | 74.46 | 88.17 | 0.084 | 93.36 | 57.10 | 67.30 | 72.59 | 0.087 |
| *SliceFine-5R* | 92.75 | 99.73 | 91.64 | 87.72 | 94.70 | 75.28 | 80.11 | 88.85 | 0.415 | 94.54 | 57.16 | 67.49 | 73.06 | 0.417 |
| *SliceFine-5C* | 92.64 | 99.64 | 91.15 | 87.78 | 94.42 | 75.14 | 80.39 | 88.74 | 0.415 | 94.53 | 57.68 | 67.44 | 73.22 | 0.417 |
| *SliceFine-5RC* | 92.28 | 99.53 | 91.71 | 87.58 | 94.89 | 75.20 | 80.89 | 88.83 | 0.415 | 94.17 | 57.69 | 67.40 | 73.09 | 0.417 |

Table 2: Performance comparison on VTAB-1K image and video benchmarks. SliceFine achieves competitive or superior performance to SOTA PEFT baselines with significantly fewer parameters.

erage accuracy, surpassing LoRA (88.08%) and AdaLoRA (87.96%). On video recognition using `VideoMAE-base`, *SliceFine-5RC* achieves 73.09%, outperforming HRA (72.53%) and MiSS (72.99%), highlighting that slice winners generalize beyond static perception to spatio-temporal modeling.

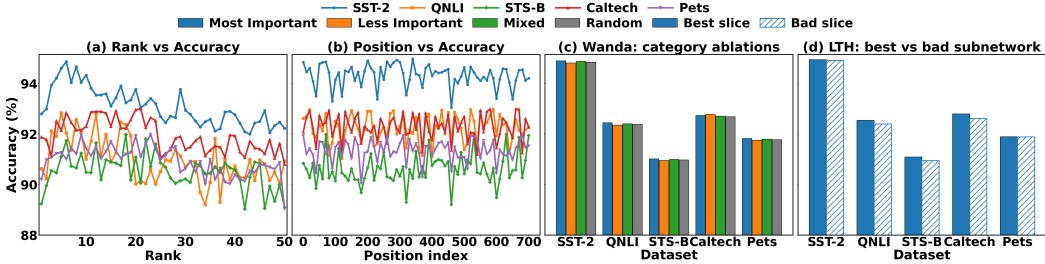

Figure 3: Empirical evidence for the robustness of slice selection strategies across tasks. (a) **Rank vs. Accuracy:** Increasing the slice rank improves accuracy up to a point, after which validation accuracy declines, indicating gradual overfitting. (b) **Position vs. Accuracy:** accuracy remains stable across slice positions, within ±1% of the anchor accuracy. (c) **Wanda category ablations:** accuracy is insensitive to whether slices are chosen from most important, less important, mixed, or random weights. (d) **LTH comparison:** even "bad" slices perform comparably to the "best" slices, supporting the *winner-slice property*—pretrained networks contain many capable subnetworks.

| Methods | Space | Time | #TTPs | #APs |
|---|---|---|---|---|
| FT | $O(d_\ell \times d_{\ell-1})$ | $O(d_\ell \times d_{\ell-1})$ | $d_\ell \cdot d_{\ell-1}$ | 0 |
| $(IA)^3$ | $O(d_k + d_v + d_{ff})$ | $O(d_k + d_v + d_{ff})$ | $d_k + d_v + d_{ff}$ | $d_k + d_v + d_{ff}$ |
| Prompt | $O(d_\ell \times l_p)$ | $O(d_\ell \times l_p)$ | $l_p \cdot d_\ell$ | $l_p \cdot d_\ell$ |
| Prefix | $O(L \times d_\ell \times l_p)$ | $O(L \times d_\ell \times l_p)$ | $L \cdot l_p \cdot d_\ell$ | $L \cdot l_p \cdot d_\ell$ |
| LoRA | $O((d_\ell + d_{\ell-1}) \times r)$ | $O((d_\ell + d_{\ell-1}) \times r)$ | $d_\ell r + d_{\ell-1} r$ | $d_\ell r + d_{\ell-1} r$ |
| LoRA-FA | $O((d_\ell + d_{\ell-1}) \times r)$ | $O((d_\ell + d_{\ell-1}) \times r)$ | $d_\ell r$ | $d_\ell r + d_{\ell-1} r$ |
| AdaLoRA | $O((d_\ell + d_{\ell-1} + r) \times r)$ | $O((d_\ell + d_{\ell-1} + r) \times r)$ | $(d_\ell + d_{\ell-1})r + r^2$ | $(d_\ell + d_{\ell-1})r + r^2$ |
| LoHA | $O(2r(d_\ell + d_{\ell-1}))$ | $O(2r(d_\ell + d_{\ell-1}))$ | $2(d_\ell r + d_{\ell-1} r)$ | $2(d_\ell r + d_{\ell-1} r)$ |
| Propulsion | $O(d_\ell)$ | $O(d_\ell)$ | $d_\ell$ | $d_\ell$ |
| VeRA | $O((d_\ell + d_{\ell-1})r + r + d_\ell)$ | $O((d_\ell + d_{\ell-1})r + r + d_\ell)$ | $d_\ell + r$ | $(d_\ell + d_{\ell-1})r + d_\ell + r$ |
| *SliceFine* (row) | $O(d_\ell \times r)$ | $O(d_\ell \times r)$ | $r d_\ell$ | 0 |
| *SliceFine* (column) | $O(d_{\ell-1} \times r)$ | $O(d_{\ell-1} \times r)$ | $r d_{\ell-1}$ | 0 |

Table 3: Comparison of space and time complexity, total trainable parameters (#TTPs), and additional parameters (#APs) introduced by different PEFT methods for a single layer $W^{(\ell)} \in \mathbb{R}^{d_\ell \times d_{\ell-1}}$. We denote $d_k, d_v$, and $d_{ff}$ as the dimensions of the three learned vectors in $IA^3$, and $l_p$ as the prompt length in prompt tuning and prefix tuning. For LoRA-type methods, $r$ denotes the rank. SliceFine achieves $O(d_\ell \times r)$ or $O(d_{\ell-1} \times r)$ complexity with no additional parameters, in contrast to other methods that incur higher asymptotic costs or parameter overhead.

**Is any slice a winner?** Empirically, we investigate the *Universal Winning–Slice Hypothesis* through three complementary experiments, all of which suggest that slice selection is largely insensitive—supporting the *winner-slice property*. As shown in Figure 3(b), performance remains stable across slice positions. Using a fixed slice rank of 5, we evaluate accuracy when training slices at different positions of the weight matrix. Accuracy remains within $\pm 1\%$ of the anchor across all positions, indicating that row and column slices contribute comparably to the task-relevant subspace.

In Figure 3(c), we adopt the Wanda pruning heuristic (Sun et al., 2023b) to rank weights by importance. Given a weight matrix $W \in \mathbb{R}^{d_\ell \times d_{\ell-1}}$ and activations $X \in \mathbb{R}^{s \times d_{\ell-1}}$ from a sequence of length $s$, Wanda defines the importance of entry $(i,j)$ as $S_{ij} = |W_{ij}| \cdot \|X_{\cdot j}\|_2$, where $\|X_{\cdot j}\|_2$ is the $\ell_2$ norm of the $j$-th input feature across the batch. To score a slice, we aggregate over its entries: $S_{slice} = \sum_{(i,j) \in slice} S_{ij}$. We then select slices from the most important, least important, mixed, or random categories. Figure 3(c) shows that all categories yield nearly identical accuracy, confirming that slice winners emerge regardless of weight importance.

Finally, Figure 3(d) compares "good" and "bad" slices using the Lottery Ticket Hypothesis framework following Frankle & Carbin (2019). Standard LTH seeks a sparse binary mask $M \in \{0,1\}^d$ such that the pruned subnetwork $\theta \odot M$ matches the accuracy of the full model: $\mathcal{L}(f_{\theta \odot M}(x), y) \approx \mathcal{L}(f_\theta(x), y)$. We extract both "winning" and "losing" subnetworks and use their masks to define slices. Surprisingly, even slices derived from "bad" subnetworks perform comparably to those from "good" ones, further reinforcing that pretrained networks contain many capable subnetworks and that every slice can be a winner.

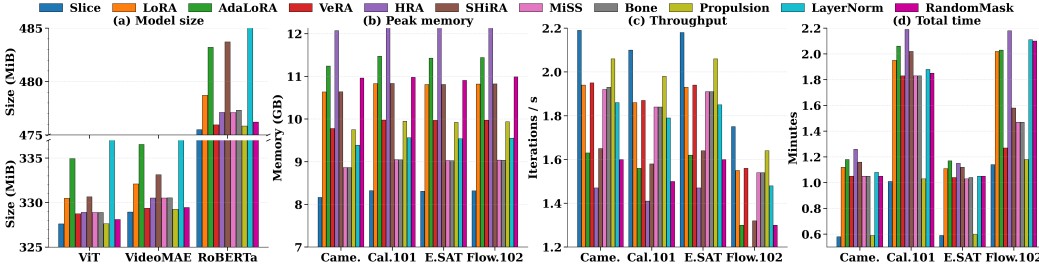

Figure 4: Comparison of PEFT methods on (a) PEFT model size, (b) peak memory, (c) throughput, and (d) total training time across **ViT**, **VideoMAE**, **RoBERTa**, and multiple datasets.

**Efficiency Analysis.** In this section we discuss the efficiency of *SliceFine*. All methods are trained under identical conditions: 10 epochs, batch size 64, fixed sequence lengths, BF16 precision, and the same learning schedules. Reported runtime is the wall-clock time for a full training run, averaged over three seeds, and throughput is measured in iterations per second on identical hardware (NVIDIA A100 80GB). Table 3 summarizes asymptotic costs. Slice training scales as $O(d_\ell \times r)$ (row) or $O(d_{\ell-1} \times r)$ (column) and introduces no additional parameters (#APs = 0). In contrast,

LoRA-style and adapter-style baselines incur $2r(d_\ell + d_{\ell-1})$ or higher parameter overhead, significantly increasing both space and time complexity.

Figure 4 presents empirical results across ViT, VideoMAE, and RoBERTa backbones on four VTAB-1K datasets. Figure 4(a) shows that slices yield compact model sizes, reducing storage by 3–5% compared to LoRA and AdaLoRA. Figure 4(b) demonstrates reduced peak memory usage: slices consistently use 2–4GB less memory ($\approx 18\%$ savings on average) compared to HRA and AdaLoRA, enabling training under stricter GPU memory budgets. Figure 4(c) reports throughput, where slices achieve the fastest iteration rates, averaging 2.05 iterations/s versus 1.78 for LoRA and 1.62 for HRA, representing a 15–25% speedup. Finally, Figure 4(d) shows total wall-clock training time: slice training completes 10 epochs in 1.05 minutes on average, compared to 1.83 minutes for HRA and 2.12 minutes for LayerNorm tuning, corresponding to a 42–50% reduction.

**Can Random Masks Act as Winning Slices?** To test whether unstructured subsets of weights can also exhibit the winning-slice property, we replace structured row/column slices with an *unstructured random mask* in each layer. As shown in Table 9 and Table 10, random masks attain competitive accuracy, supporting our claim that pretrained models contain many potential winning slices.

However, random masks introduce substantial practical inefficiencies. Unstructured sparsity breaks the dense matrix layout used by modern hardware, requiring gather/scatter indexing and disabling fused GEMM kernels, which significantly slows training. Moreover, random masking requires additional metadata (mask indices) and prevents selectively disabling gradients or optimizer states for the frozen weights. As a result, peak memory and wall-clock time are both higher than with SliceFine's structured slices.

Figure 4 illustrates these efficiency differences: random masks slightly increase model size and noticeably increase peak memory and training time. Further analysis is provided in Appendix L. Overall, while random subsets can act as winning slices, SliceFine's contiguous row/column slices offer a far better balance between expressivity and hardware efficiency.

Overall: (i) *SliceFine* supports the UWSH means any slice with sufficient rank acts as a local winner, so training slice suffices for PEFT without adding adapters; (ii) empirically, a rank–1 slice already delivers competitive performance; and (iii) *SliceFine* matches or exceeds baseline accuracy while using far fewer trainable parameters, and thus training faster, and consuming less memory. The more detailed results are provided in Appendix K. In addition, we present detailed ablation studies in Appendix F. The slice–rank ablation (§F.1) shows that accuracy initially increases with rank, as larger slices capture more task-relevant directions within the representation subspace. However, beyond a certain point, the gains saturate and adding additional parameters yields diminishing returns.

The switching-interval ablation (§F.2), summarized in Figure 11, further shows that switching slices too frequently degrades accuracy, while switching too late produces stable performance but not the best results. Intermediate switching intervals consistently achieve the strongest results, with the optimal value depending on the dataset.

We also study the optimal slice rank in §F.3. Table 4 reports results across ranks $r \in \{1, 2, 4, 8, 32, 64, 128\}$ and across different module placements—using SliceFine only on the value matrix; on query/key/value; on all attention matrices; on the transformer block; or on all layers. Next, in Figure 12, we compare *dynamic* versus *static* slice training, and find that dynamic slicing consistently outperforms static slicing, details is in §F.4.

Finally, we include a multi-seed robustness analysis on a larger LLM in §F.5 (Table 5), using ten seeds on the GPT-OSS-20B model. The results show consistently *low variance* across seeds, indicating that SliceFine is stable and robust to random initialization.

## 5 RELATED WORK

**Lottery Ticket Hypothesis.** LTH (Frankle & Carbin, 2019) says a large dense network contains sparse subnetworks ("winning tickets") that, after pruning and retraining, can match the full model. Later work tested this on modern architectures (Burkholz, 2022) and surveyed the evidence (Liu et al., 2024a). Pruning methods to find these subnetworks were studied in Zhou et al. (2019). The *strong* LTH (Ramanujan et al., 2020) shows that some untrained sparse subnetworks already perform

well, with further analysis and CNN results in Pensia et al. (2020); da Cunha et al. (2022). In pre-trained models, similar findings appear: BERT has winning-ticket subnetworks (Chen et al., 2020; Yu et al., 2021); pruning before fine-tuning can find them (Wu et al., 2022); and ImageNet-pretrained CNNs also contain sparse subnetworks that keep downstream accuracy (Chen et al., 2021; Iofinova et al., 2022). Recent work applies LTH to LLM adaptation: KS-Lottery finds certified winning tick-ets in LLaMA embeddings that match full fine-tuning for multilingual transfer (Yuan et al., 2024a), while Lottery Ticket Adaptation (LoTA) identifies sparse task vectors that avoid multi-task interfer-ence (Panda et al.). For code models, PLEX selects influential parameters via a lottery-ticket strat-egy and achieves state-of-the-art efficiency (Lee et al., 2025). Theoretical advances also show that Ramanujan-graph–based sparse subnetworks can preserve strong connectivity and accuracy even at extreme sparsity (Pal et al., 2022).

**PEFT.** PEFT adapts large models by updating a small subset of parameters while freezing most of the backbone. Prior work spans prompt-based approaches that optimize soft prompts or pre-fixes (Lester et al., 2021; Li & Liang, 2021), adapter layers inserted into residual blocks (Houlsby et al., 2019), and weight-space updates such as LoRA (Hu et al., 2021), BitFit and LayerNorm tun-ing (Zaken et al., 2021; Zhao et al., 2023), and multiplicative gating like IA$^3$ (Liu et al., 2022). Numerous refinements improve placement, rank, and overhead: AdaLoRA adapts rank during train-ing (Zhang et al., 2023b), LoRA-FA and VeRA reduce parameter and optimizer cost (Zhang et al., 2023a; Kopiczko et al., 2023), and other variants explore alternative factorizations or injection points (Hyeon-Woo et al., 2021; Edalati et al., 2022; Bałazy et al., 2024). Beyond LoRA, DoRA stabilizes updates via weight decomposition (Liu et al., 2024b), READ augments with residual adapters (Wang et al., 2024), LoRA+ and KronA target better efficiency (Hayou et al., 2024; Edalati et al., 2022), and Q-PEFT conditions adaptation on task queries (Peng et al., 2024). Orthogonal lines combine PEFT with sparsity: structured and unstructured pruning (e.g., SparseGPT, Wanda, LLM-Pruner) remove redundancy with limited loss (Frantar & Alistarh, 2023; Sun et al., 2023a; Ma et al., 2023), and domain adaptation strategies help avoid overfitting (Gururangan et al., 2020). Closer to sub-network training, RoCoFT trains a fixed subset of rows/columns and reports strong results, but the chosen subnetwork remains static over training (Kowsher et al., 2024a).

Recent PEFT methods also explore the spectral or structural properties of pretrained weights. Spec-tral Adapter performs fine-tuning directly in the top singular subspace obtained from the pretrained SVD and improves the rank expressivity of low-rank updates (Zhang & Pilanci, 2024). XoRA intro-duces expander-masked LoRA factors using Ramanujan graphs to sparsify low-rank matrices while preserving connectivity (Amaljith et al., 2024). RandLoRA replaces low-rank factors with learned combinations of full-rank random bases, addressing LoRA's representational limits while keeping parameter count low (Albert et al., 2025). *SFT* proposes a sparse fine-tuning mechanism that dy-namically updates, prunes, and regrows parameter indices, achieving strong performance on LLaMA models while maintaining low memory overhead (Ansell et al., 2024). These approaches modify the structure or basis of the update itself, rather than selecting a subset of pretrained weights. In contrast to methods that add auxiliary parameters (adapters, prompts, low-rank factors) or remove weights via pruning, *SliceFine* updates a set of slices across layers of the original network—without adding parameters—and can move the active slice during training, yielding block–coordinate adaptation motivated by spectral balance and high task energy in the pretrained backbone.

## 6 CONCLUSION

This work proposes the *Universal Winning–Slice Hypothesis (UWSH)*: in pretrained networks sat-isfying spectral balance and high task energy, every sufficiently wide slice of a weight matrix acts as a local winning ticket, and a small set of slices across layers forms a global winning ticket. Building on this view, *SliceFine* fine-tunes only a small, moving slice per layer—adding no new pa-rameters—while matching strong PEFT baselines in accuracy and improving efficiency in memory, speed, and model size. Experiments across text, image, and video tasks show competitive perfor-mance compared to SOTA PEFT methods.

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

CONTENTS

## THE USE OF LARGE LANGUAGE MODELS (LLMS)

We used a large language model (GPT) solely for minor writing assistance, such as grammar checking, language polishing, and improving readability. No content generation, ideation, experimental design, or analysis was performed by the LLM. All research contributions, technical content, and results presented in this paper are entirely the work of the authors.

## A   THEORETICAL JUSTIFICATION OF SPECTRAL BALANCE

We consider a single pretrained layer $W \in \mathbb{R}^{d_\ell \times d_{\ell-1}}$ with rows $w_1^\top, \ldots, w_{d_\ell}^\top$. The $d_\ell$ rows are partitioned uniformly at random into $k$ disjoint groups $G_1, \ldots, G_k$ of equal size $r = d_\ell/k$. For each group $G_g$, let $W_g \in \mathbb{R}^{r \times d_{\ell-1}}$ collect the rows indexed by $G_g$, and define

$$C_g := W_g^\top W_g \in \mathbb{R}^{d_{\ell-1} \times d_{\ell-1}}, \qquad C := W^\top W \in \mathbb{R}^{d_{\ell-1} \times d_{\ell-1}}, \qquad \Sigma_g := W_g W_g^\top \in \mathbb{R}^{r \times r}.$$

Note that $C_g$ and $\Sigma_g$ share the same nonzero eigenvalues (up to multiplicity).

We assume the following mild conditions, empirically satisfied by pretrained layers: (A1) There exists $\mu \geq 1$ such that $\|w_i\|_2^2 \leq \mu \|W\|_F^2/d_\ell$ for all $i \in [d_\ell]$. (A2) The Gram matrix $\frac{1}{d_\ell}C$ satisfies $\lambda_{\min}(\frac{1}{d_\ell}C) \geq \alpha > 0$.

Now, under *(A1)–(A2)*, with probability at least $1 - \delta$ over the random partition, the following hold simultaneously for all $g, g' \in [k]$:

$$\left\| \tfrac{1}{r}C_g - \tfrac{1}{d_\ell}C \right\|_{\mathrm{op}} \leq \varepsilon, \tag{1}$$

$$\left| \tfrac{1}{r}\operatorname{tr}(C_g) - \tfrac{1}{d_\ell}\operatorname{tr}(C) \right| \leq c_1 \frac{\|W\|_F^2}{d_\ell} \sqrt{\frac{\mu k \log(k/\delta)}{d_\ell}}, \tag{2}$$

where, for universal constants $c_0, c_1 > 0$,

$$\varepsilon = c_0 \frac{\mu \|W\|_F^2}{d_\ell^{3/2}} \sqrt{k \log \frac{kd_{\ell-1}}{\delta}}. \tag{3}$$

For each $i \in [d_\ell]$, define $X_i := w_i w_i^\top \in \mathbb{R}^{d_{\ell-1} \times d_{\ell-1}}$, so that $C = \sum_{i=1}^{d_\ell} X_i$ and $C_g = \sum_{i \in G_g} X_i$. Introduce the centered matrices

$$Z_i := X_i - \frac{1}{d_\ell}C \quad \text{so that} \quad \sum_{i=1}^{d_\ell} Z_i = 0,$$

and define

$$Y_g := \frac{1}{r}\sum_{i \in G_g} Z_i = \frac{1}{r}C_g - \frac{1}{d_\ell}C.$$

Then $\mathbb{E}[Y_g] = 0$ and equation 1 is a bound on $\|Y_g\|_{\mathrm{op}}$.

By *(A1)* we have $\|X_i\|_{\mathrm{op}} = \|w_i\|_2^2 \leq \mu \|W\|_F^2/d_\ell$. Also $\left\| \tfrac{1}{d_\ell}C \right\|_{\mathrm{op}} \leq \|W\|_F^2/d_\ell$. Hence

$$\|Z_i\|_{\mathrm{op}} \leq (\mu + 1)\frac{\|W\|_F^2}{d_\ell} =: L.$$

Define $V := \sum_{i=1}^{d_\ell} Z_i^2$; then using $\|A^2\|_{\mathrm{op}} \leq \|A\|_{\mathrm{op}}^2$,

$$\|V\|_{\mathrm{op}} \leq \sum_{i=1}^{d_\ell} \|Z_i\|_{\mathrm{op}}^2 \leq d_\ell L^2 = (\mu + 1)^2 \frac{\|W\|_F^4}{d_\ell}.$$

When we sample $r$ indices uniformly without replacement to form $G_g$, the covariance of the sample mean satisfies the finite-population correction (Hoeffding),

$$\mathbb{E}[Y_g Y_g^\top] = \frac{d_\ell - r}{r \, d_\ell \, (d_\ell - 1)} V.$$

Therefore

$$\left\|\mathbb{E}[Y_g Y_g^\top]\right\|_{\mathrm{op}} \leq \frac{d_\ell - r}{r\, d_\ell\,(d_\ell - 1)} \|V\|_{\mathrm{op}} \leq \frac{k(\mu+1)^2\, \|W\|_F^4}{d_\ell^3} =: \sigma^2,$$

since $r = d_\ell/k$ and $(d_\ell - r)/(d_\ell - 1) \leq 1$.

Applying the matrix Bernstein inequality (Tropp, 2012) with variance proxy $\sigma^2$ and summand bound $L$ yields, for any $t > 0$,

$$\Pr\left(\|Y_g\|_{\mathrm{op}} \geq t\right) \leq 2d_{\ell-1} \exp\left(-\frac{t^2/2}{\sigma^2 + Lt/3}\right).$$

Choosing $t = c\sqrt{\sigma^2 \log(\frac{kd_{\ell-1}}{\delta})}$ for a universal $c > 0$ and union-bounding over $g \in [k]$ gives, with probability at least $1 - \delta$,

$$\|Y_g\|_{\mathrm{op}} \leq c_0\, \frac{\mu\, \|W\|_F^2}{d_\ell^{3/2}} \sqrt{k \log \frac{kd_{\ell-1}}{\delta}} = \varepsilon,$$

which establishes equation 1. For the trace, note that $\frac{1}{r}\operatorname{tr}(C_g) = \frac{1}{r}\sum_{i \in G_g} \|w_i\|_2^2$. The scalar Bernstein inequality for sampling without replacement, combined with (A1), gives equation 2.

By Weyl's inequality (Weyl, 1912),

$$\left|\lambda_i(\tfrac{1}{r}C_g) - \lambda_i(\tfrac{1}{r}C_{g'})\right| \leq \left\|\tfrac{1}{r}C_g - \tfrac{1}{d_\ell}C\right\|_{\mathrm{op}} + \left\|\tfrac{1}{r}C_{g'} - \tfrac{1}{d_\ell}C\right\|_{\mathrm{op}} \leq 2\varepsilon.$$

Multiplying both sides by $r$ gives

$$\left|\lambda_i(C_g) - \lambda_i(C_{g'})\right| \leq 2r\,\varepsilon. \tag{4}$$

Since $C_g$ and $\Sigma_g$ share the same nonzero eigenvalues, it follows that

$$\left|\lambda_i(\Sigma_g) - \lambda_i(\Sigma_{g'})\right| \leq 2r\,\varepsilon = \frac{2d_\ell}{k}\,\varepsilon.$$

Finally, using (A2) we have $\lambda_i(\frac{1}{r}C_{g'}) \geq \alpha - \varepsilon$, hence $\lambda_i(\Sigma_{g'}) = \lambda_i(C_{g'}) \geq r(\alpha - \varepsilon)$, and therefore

$$\frac{\left|\lambda_i(\Sigma_g) - \lambda_i(\Sigma_{g'})\right|}{\lambda_i(\Sigma_{g'})} \leq \frac{2r\,\varepsilon}{r(\alpha - \varepsilon)} = \frac{2\varepsilon}{\alpha - \varepsilon} =: \rho. \tag{5}$$

Taking traces in equation 1 and using equation 2,

$$\left|\frac{1}{r}\sum_{i=1}^{r} \lambda_i(C_g) - \frac{1}{d_\ell}\sum_{i=1}^{d_\ell} \lambda_i(C)\right| \leq c_1\, \frac{\|W\|_F^2}{d_\ell} \sqrt{\frac{\mu k\, \log(k/\delta)}{d_\ell}}.$$

Transferring to $\Sigma_g$ (nonzero spectra coincide) gives, for all $g, g'$,

$$\frac{\frac{1}{r}\sum_{i=1}^{r} \lambda_i(\Sigma_g)}{\frac{1}{r}\sum_{i=1}^{r} \lambda_i(\Sigma_{g'})} = 1 \pm O\left(\sqrt{\frac{\mu k\, \log(k/\delta)}{d_\ell}}\right),$$

matching the "average-energy $\approx 1$" part of Lemma 2.1.

## B    PROOF OF THEOREM UNIVERSAL WINNING TICKET

***Proof of Theorem 2.4.*** Let $\mathcal{L}(\theta) = \frac{1}{n}\sum_{i=1}^{n} \ell(f_\theta(x_i), y_i)$ denote the empirical downstream loss on dataset $D = \{(x_i, y_i)\}_{i=1}^{n}$, with pretrained initialization $\theta_0$. For any slice $M \subseteq W^{(\ell)}$, define the restricted gradient

$$g_M = \nabla_M \mathcal{L}(\theta) = \frac{\partial \mathcal{L}(\theta)}{\partial(M \odot W^{(\ell)})} = \frac{1}{n}\sum_{i=1}^{n} J_M(x_i)^\top \nabla_f \ell(f_\theta(x_i), y_i), \tag{6}$$

where $J_M(x) = \frac{\partial f_\theta(x)}{\partial(M \odot W^{(\ell)})} \in \mathbb{R}^{k \times |M|}$ is the slice Jacobian and $\nabla_f \ell(\cdot, y_i) \in \mathbb{R}^k$ is the gradient with respect to logits. By assumption there exists at least one $i$ with $\nabla_f \ell(f_{\theta_0}(x_i), y_i) \neq 0$, and by the

projection condition each $J_M(x)$ has nontrivial overlap with the $k_{\text{task}}$-dimensional task subspace. Hence $g_M \neq 0$ for all nonempty $M$.

Assume slice-restricted Lipschitz smoothness: there exists $L_M > 0$ such that

$$\mathcal{L}(\theta_0 + M \odot \Delta) \leq \mathcal{L}(\theta_0) + g_M^\top \text{vec}(\Delta) + \frac{L_M}{2}\|\text{vec}(\Delta)\|_2^2 \tag{7}$$

for all $\Delta$ with $\|\Delta\|_2 \leq \rho$. Choosing $\Delta = -\alpha g_M$ yields

$$\mathcal{L}(\theta_0 - \alpha M \odot g_M) \leq \mathcal{L}(\theta_0) - \alpha\|g_M\|_2^2 + \frac{L_M}{2}\alpha^2\|g_M\|_2^2, \tag{8}$$

which is minimized at $\alpha^\star = 1/L_M$. Thus for any $\alpha \in (0, 2/L_M)$ we obtain strict decrease

$$\mathcal{L}(\theta_0 - \alpha M \odot g_M) \leq \mathcal{L}(\theta_0) - \frac{\|g_M\|_2^2}{2L_M} < \mathcal{L}(\theta_0). \tag{9}$$

Therefore every slice admits a loss-decreasing update.

Next, let $\{M_1, \ldots, M_m\}$ be slices chosen such that their Jacobians span the task subspace:

$$\dim \text{span}\{J_{M_1}(x), \ldots, J_{M_m}(x)\} = k_{\text{task}}. \tag{10}$$

By the spectral balance assumption such a finite $m \leq k_{\text{task}}$ exists. Define the combined Jacobian

$$J_{\text{combined}}(x) = [J_{M_1}(x) \mid \cdots \mid J_{M_m}(x)]. \tag{11}$$

In the linearized (NTK) regime,

$$f_{\theta_0 + \Delta\theta}(x) \approx f_{\theta_0}(x) + J(x)\Delta\theta, \tag{12}$$

and restriction to updates supported on $\{M_i\}$ suffices to realize any perturbation in the $k_{\text{task}}$ subspace. Since losses such as cross-entropy are convex in the logits, it follows that for any $\epsilon > 0$ there exist slice updates $\{U_i\}$ such that

$$\mathcal{L}\left(\theta_0 + \sum_{i=1}^m M_i \odot U_i\right) \leq \epsilon. \tag{13}$$

Hence the union of finitely many slices constitutes a global winning ticket. $\qquad\square$

# C    PCA DECOMPOSITION OF THE REPRESENTATION/LINEARIZED NTK KERNEL)

***Proof of Lemma 2.5***. Let $\tilde{H} \in \mathbb{R}^{n \times d}$ for the centered feature matrix. Since centering removes the mean in each feature coordinate, we have $\text{rank}(\tilde{H}) =: r \leq \min\{n - 1, d\}$. Consider the thin singular value decomposition (SVD)

$$\tilde{H} = USV^\top,$$

where $U \in \mathbb{R}^{n \times r}$ and $V \in \mathbb{R}^{d \times r}$ have orthonormal columns ($U^\top U = V^\top V = I_r$), and $S = \text{diag}(s_1, \ldots, s_r) \in \mathbb{R}^{r \times r}$ with singular values $s_1 \geq \cdots \geq s_r > 0$.

The empirical covariance of the centered features is

$$\Sigma = \frac{1}{n-1}\tilde{H}^\top\tilde{H} = \frac{1}{n-1}VS^2V^\top.$$

Thus $V$ diagonalizes $\Sigma$ and its positive eigenvalues are

$$\lambda_i = \frac{s_i^2}{n-1}, \qquad i = 1, \ldots, r,$$

so that $\Lambda = \text{diag}(\lambda_1, \ldots, \lambda_r) = \frac{1}{n-1}S^2$ gives the eigendecomposition $\Sigma = V\Lambda V^\top$ stated in the lemma.

Define $P := \tilde{H}V$. Using the SVD, $P = (USV^\top)V = US$, hence $P \in \mathbb{R}^{n \times r}$ has orthogonal columns and $PP^\top = (US)(US)^\top = US^2U^\top$. On the other hand, the centered feature Gram matrix (a.k.a. linearized representation kernel)

$$K = \tilde{H}\tilde{H}^\top = (USV^\top)(VSU^\top) = US^2U^\top.$$

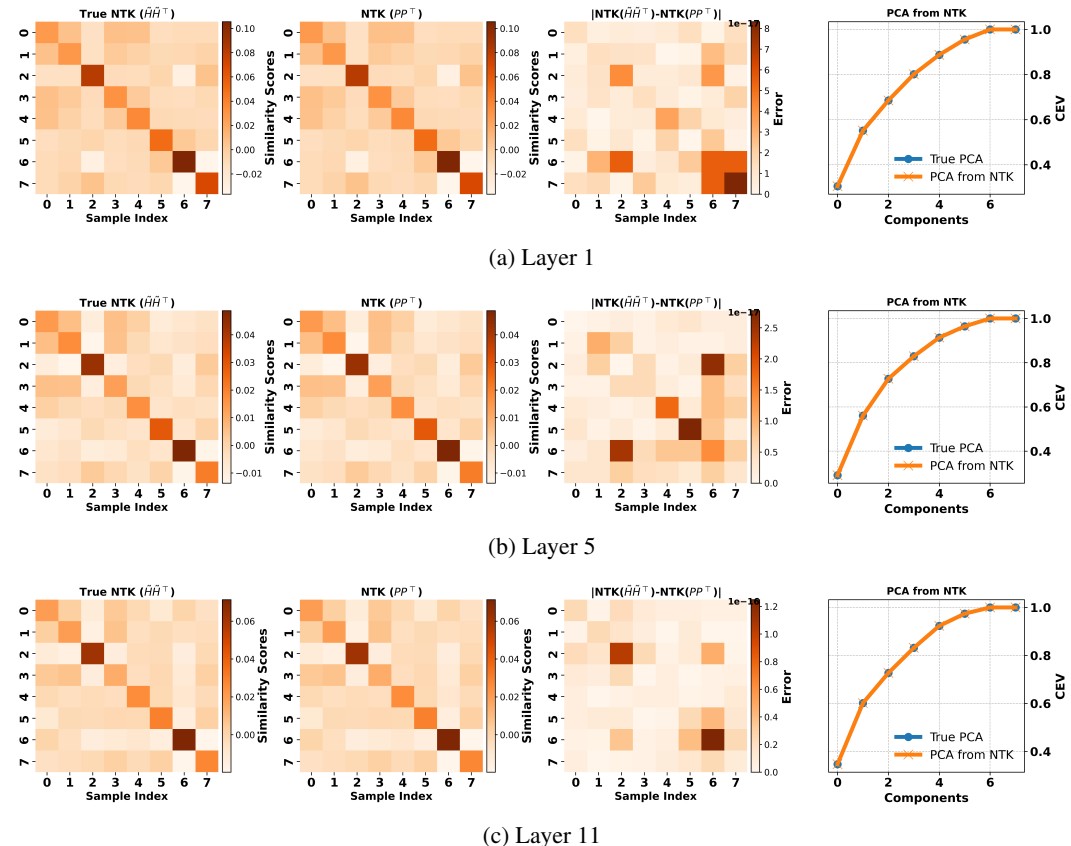

(a) Layer 1

(b) Layer 5

(c) Layer 11

Figure 5: **PCA/NTK agreement across layers.** Each row shows (i) the centered representation kernel $K = \tilde{H}\tilde{H}^\top$, (ii) the reconstruction $PP^\top$ with $P = \tilde{H}V$, (iii) the absolute difference $|K - PP^\top|$, and (iv) cumulative explained variance from PCA on $\Sigma$ versus eigenvalues from $K$. Results are shown for RoBERTa-base layers 1, 5, and 11. Spectra and CEV curves overlap, confirming Lemma 2.5 across depth.

Therefore $K = PP^\top$ exactly, which is the desired decomposition.

The eigen-decomposition of $K$ follows immediately: because $K = US^2U^\top$ with $U$ orthonormal, the nonzero eigenvalues of $K$ are the diagonal entries of $S^2$, namely $s_i^2$, $i = 1, \ldots, r$. Using the relation $s_i^2 = (n-1)\lambda_i$ derived above, the nonzero spectrum of $K$ is $\mu_i := (n-1)\lambda_i$, $i = 1, \ldots, r$. Equivalently, $K$ and $\Sigma$ share the same nonzero eigenvectors in their respective spaces ($U$ in $\mathbb{R}^n$ and $V$ in $\mathbb{R}^d$) and have spectra that differ by the scalar factor $(n-1)$. This scaling also implies equality of explained-variance ratios. Indeed,

$$\frac{\sum_{i=1}^k \lambda_i}{\sum_{j=1}^r \lambda_j} = \frac{\sum_{i=1}^k \frac{1}{n-1}s_i^2}{\sum_{j=1}^r \frac{1}{n-1}s_j^2} = \frac{\sum_{i=1}^k s_i^2}{\sum_{j=1}^r s_j^2} = \frac{\sum_{i=1}^k \mu_i}{\sum_{j=1}^r \mu_j},$$

since $\mu_i = s_i^2$. Hence PCA explained-variance ratios computed from the feature covariance $\Sigma$ coincide exactly with those computed from the Gram kernel $K$.

For completeness, one can also see the spectral correspondence without SVD, using the algebraic fact that $AB$ and $BA$ have the same nonzero eigenvalues (including multiplicities). Taking $A = \tilde{H}$ and $B = \frac{1}{n-1}\tilde{H}^\top$, the products $AB = \frac{1}{n-1}\tilde{H}\tilde{H}^\top$ and $BA = \frac{1}{n-1}\tilde{H}^\top\tilde{H}$ share their nonzero spectra; this gives again that the positive eigenvalues of $K$ equal $(n-1)$ times those of $\Sigma$ and yields the same variance-ratio identity.

Finally, in the context of linearized training around $\theta_0$, when the readout is linear in the representation $\tilde{H}$ or the network is considered in the (first-order) tangent regime, the kernel governing

function-space updates reduces to the representation kernel $K = \tilde{H}\tilde{H}^\top$; the lemma thus characterizes its spectrum through the PCA of $\tilde{H}$, completing the proof. $\square$

Empirically, Figure 5 validates the PCA/NTK correspondence on RoBERTa-base using $n=8$ examples at layers 1, 5, and 11. For each layer we compute the centered representation kernel $K = \tilde{H}\tilde{H}^\top$ ("True NTK" panels), the PCA basis $V$ from $\Sigma = \frac{1}{n-1}\tilde{H}^\top\tilde{H}$, and the reconstruction $PP^\top$ with $P = \tilde{H}V$ (middle-left panels). The absolute difference maps $|K - PP^\top|$ (middle-right) are uniformly small, confirming the exact decomposition $K = PP^\top$ up to numerical error. The rightmost plots compare cumulative explained variance derived from the eigenvalues of $\Sigma$ ("True PCA") versus those induced by $K$ ("PCA from NTK"); the curves overlap across depth, showing that PCA variance ratios in feature space match those of the representation kernel. Together, these results support Lemma 2.5 empirically across multiple layers.

*Remark* C.1 (Relation to the NTK). The identity $K = \tilde{H}\tilde{H}^\top = PP^\top$ describes the (centered) representation kernel. It coincides with the empirical NTK $K_{\theta_0}(x, x') = \langle \nabla_\theta f_{\theta_0}(x), \nabla_\theta f_{\theta_0}(x') \rangle$ when the parameter subset with respect to which the NTK is computed corresponds to a linear readout over fixed features (e.g., last-layer weights with upstream layers frozen). For general multi-layer adaptations, the full NTK requires Jacobians w.r.t. all trainable parameters and does not reduce to a pure feature Gram matrix.

# D RANK, NTK, AND PCA

This section quantifies how *domain familiarity* affects the slice rank needed for good adaptation. The main idea is simple: when the frozen backbone already organizes features along task-relevant directions, a small slice rank is enough; when the task is unfamiliar, a larger rank is needed. We make this precise using PCA on hidden features and the linearized NTK view: a steep PCA spectrum (high cumulative explained variance, CEV) indicates that the task subspace is low-dimensional and small slices should suffice; a flat spectrum indicates the opposite.

At the beginning, we warm-start RoBERTa-base on MRPC for 100 steps (batch size 64, BF16) to bias its features toward paraphrase semantics without fully adapting the model. We then fine-tune with SliceFine at ranks $r \in \{1, 3, 5, 7\}$ on six GLUE tasks, and report three diagnostics that map directly to the analysis.

## D.1 PCA CUMULATIVE EXPLAINED VARIANCE (CEV) ANALYSIS

For layer $\ell$, let the centered features be $\tilde{H}_\ell \in \mathbb{R}^{n \times d_\ell}$, the covariance $\Sigma_\ell = \frac{1}{n-1}\tilde{H}_\ell^\top \tilde{H}_\ell$, and eigenvalues $\lambda_1 \geq \cdots \geq \lambda_{r_\ell} > 0$. The CEV after $k$ components is

$$\text{CEV}_\ell(k) = \frac{\sum_{i=1}^k \lambda_i}{\sum_{j=1}^{r_\ell} \lambda_j},$$

which summarizes how concentrated the feature spectrum is.

Interpretation of Figure 6. MRPC and QQP cross 85% variance with only a few principal components, consistent with a low-dimensional task subspace when the backbone is biased toward paraphrase semantics (As warm-start with MRPC dataset). In contrast, SST-2 and CoLA require more components, pointing to higher intrinsic dimensionality. This matches the rank rule-of-thumb: a steep spectrum suggests that a small slice rank is enough; a flatter spectrum suggests that a larger rank is needed.

## D.2 PREDICTION SHIFT VIA KULLBACK–LEIBLER DIVERGENCE

Let $\theta_b$ be the warm-started backbone and $\theta_r$ the model after SliceFine with rank $r$. For a $k$-class problem with logits $z_\theta(x) \in \mathbb{R}^k$ and probabilities $p_\theta(y \mid x) = \text{softmax}(z_\theta(x))$, define the dataset-averaged divergence

$$\text{KL}_{\mathcal{D}}(\theta_r \parallel \theta_b) = \frac{1}{n}\sum_{i=1}^n \sum_{c=1}^k p_{\theta_r}(c \mid x_i) \log \frac{p_{\theta_r}(c \mid x_i)}{p_{\theta_b}(c \mid x_i)}.$$

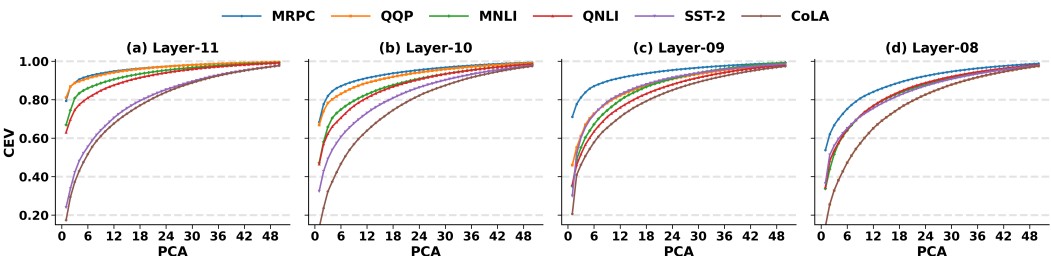

Figure 6: PCA cumulative explained variance (CEV) for eight GLUE tasks. Tasks such as MRPC and QQP reach over 85% variance with few components, indicating a compact task subspace after MRPC warm-start. Tasks such as SST-2 and CoLA accumulate variance more slowly, suggesting higher intrinsic dimensionality and a need for larger slice ranks.

Small KL means the fine-tuned model stays close to the backbone's predictions (a "lazy" update); large KL means stronger shifts.

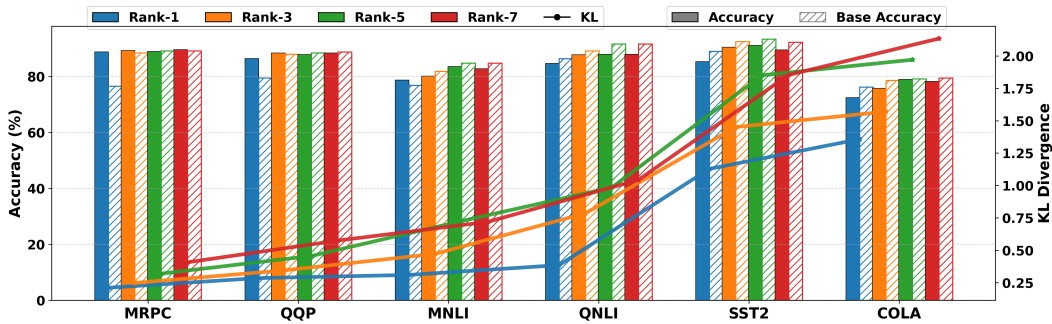

Figure 7: Accuracy (bars), base accuracy (hatched bars), and KL divergence to the warm-started backbone (lines) across six GLUE tasks (MRPC, QQP, MNLI, QNLI, SST-2, CoLA) for different slice ranks. Tasks close to MRPC (MRPC, QQP) saturate at very low rank and show small KL; tasks with flatter CEV (SST-2, CoLA) benefit from higher ranks and show larger KL.

Interpretation of Figure 7. MRPC and QQP achieve high accuracy with $r = 1$ and display low KL, consistent with high task energy already present in the backbone. SST-2 and CoLA gain from larger ranks and show higher KL, indicating that more task-relevant directions must be engaged to move predictions away from the warm-start.

### D.3 LAYER-WISE REPRESENTATION CHANGE VIA CENTERED KERNEL ALIGNMENT (CKA)

For layer $\ell$, let $H_b^{(\ell)}, H_r^{(\ell)} \in \mathbb{R}^{n \times d_\ell}$ be hidden states from $\theta_b$ and $\theta_r$; center by $\tilde{H} := H - \frac{1}{n}\mathbf{1}\mathbf{1}^\top H$. Linear CKA is

$$\mathrm{CKA}\left(H_b^{(\ell)}, H_r^{(\ell)}\right) = \frac{\left\|\tilde{H}_b^{(\ell)\top}\tilde{H}_r^{(\ell)}\right\|_F^2}{\left\|\tilde{H}_b^{(\ell)\top}\tilde{H}_b^{(\ell)}\right\|_F \left\|\tilde{H}_r^{(\ell)\top}\tilde{H}_r^{(\ell)}\right\|_F}.$$

Higher CKA means the representations stay closer to the backbone.

Interpretation of Figure 8. For MRPC and QQP, CKA remains high in final layer even at larger ranks, consistent with a lazy regime: only modest adjustments are needed. For SST-2 and CoLA, CKA decreases as rank grows—especially in upper layers—showing that these tasks require broader subspace updates. CKA is a representation-level proxy; the kernel–PCA link that motivates this interpretation is given in Lemma 2.5.

In summary, warming on MRPC increases CEV for semantically similar tasks, so low ranks work well and changes remain small (low KL, high CKA). For tasks with flatter CEV, larger ranks are

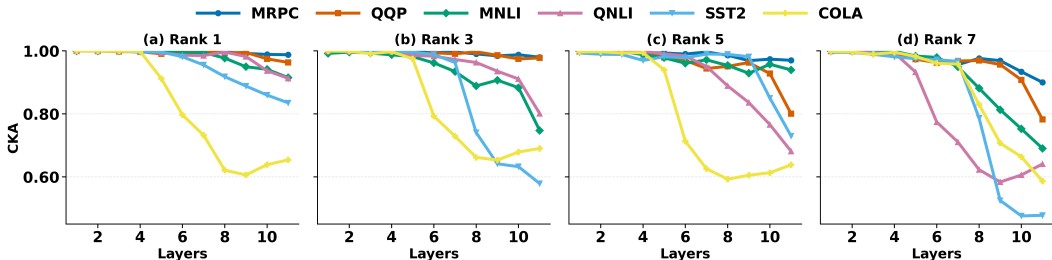

Figure 8: Layer-wise CKA between pre- and post-fine-tuning representations for ranks 1, 3, 5, and 7 across MRPC, QQP, MNLI, QNLI, SST-2, and CoLA. Larger CKA indicates smaller feature drift. MRPC/QQP remain close to the backbone even at higher ranks; SST-2/CoLA show larger drift in upper layers as rank increases.

needed, with larger prediction shifts and representation changes. These trends are exactly what the PCA/NTK view predicts: when the backbone concentrates task energy in a few directions, small slices suffice; when it does not, larger slices are required.

# E  BACKBONE DEPENDENCE

A slice can only reduce loss if its updates move the model along task-relevant directions already encoded by the pretrained backbone. Formally, the loss gradient at the output is propagated back through the frozen features, and the effect of a slice update is controlled by two factors: (i) the *task energy* present in the backbone representations (how much of the feature variance lies in the subspace that separates the labels), and (ii) the *overlap* between the slice Jacobian and that task subspace. If the pretrained network retains high principal energy on the task (large PCA cumulative explained variance), then all slices—by spectral balance—have nontrivial projection onto that subspace, yielding a nonzero restricted gradient and guaranteed decrease. Conversely, if we ablate the backbone (e.g., heavy pruning or strong domain shift), the representation kernel loses energy, the accessible task dimension shrinks, and the slice gradient collapses unless we increase the slice rank.

**Lemma E.1** (Backbone energy & alignment condition for local winners). *Let $\theta_0$ be a pretrained network and, at layer $\ell$, let $\Phi_\ell = [\phi_\ell(x_1), \ldots, \phi_\ell(x_n)]$ have singular values $\{\sigma_j\}$. Let $U_{k_{\text{task}}}$ be the top $k_{\text{task}}$ left singular vectors and assume the task-energy ratio $E := \frac{\sum_{j=1}^{k_{\text{task}}} \sigma_j^2}{\sum_j \sigma_j^2} > 0$. For a slice mask $M$ in layer $\ell$, define the feature-space slice Jacobian $J_M^\phi(x) = \frac{\partial \phi_\ell(x)}{\partial (M \odot W^{(\ell)})} \in \mathbb{R}^{d_\ell \times |M|}$ and set*

$$\gamma_{\min}(M) := \sigma_{\min}\big(U_{k_{\text{task}}}^\top J_M^\phi(x)\big), \qquad \beta(M) := \big\|(I - P_{U_{k_{\text{task}}}}) J_M^\phi(x)\big\|_2.$$

*Assume spectral balance: there exist constants $\underline{\gamma} > 0$ and $c \in [0, 1)$ such that $\gamma_{\min}(M) \geq \underline{\gamma}$ and $\beta(M) \leq c\underline{\gamma}$ for all admissible slices $M$. Let $\mathcal{L}$ be convex in the logits and let $J_{\text{out}}^{(\ell)} = \frac{\partial f_{\theta_0}}{\partial \phi_\ell}$. Define the feature-space gradient $g_\phi = (J_{\text{out}}^{(\ell)})^\top \nabla_f \mathcal{L}$ and decompose $g_\phi = g_{\phi,\text{task}} + g_{\phi,\perp}$ with $g_{\phi,\text{task}} := P_{U_{k_{\text{task}}}} g_\phi$ and $g_{\phi,\perp} := (I - P_{U_{k_{\text{task}}}}) g_\phi$. Assume the gradient alignment condition: for some $\rho \in [0, 1)$,*

$$\|g_{\phi,\perp}\| \leq \rho \|g_{\phi,\text{task}}\|, \qquad \|g_{\phi,\text{task}}\| > 0.$$

*Assume further that $\mathcal{L}$ is $L$-smooth along the slice coordinates (equivalently, $\lambda_{\max}(H_M) \leq L < \infty$ with $H_M := \nabla_M^2 \mathcal{L}(\theta_0)$). Then*

$$\big\|\nabla_M \mathcal{L}(\theta_0)\big\| = \big\|(J_M^\phi(x))^\top g_\phi\big\| \geq (1 - c\rho)\,\underline{\gamma}\,\|g_{\phi,\text{task}}\| > 0,$$

*and the slice-only update $U^\star = -\lambda_{\max}(H_M)^{-1} \nabla_M \mathcal{L}(\theta_0)$ satisfies*

$$\mathcal{L}\big(\theta_0 + M \odot U^\star\big) \leq \mathcal{L}(\theta_0) - \frac{(1 - c\rho)^2\,\underline{\gamma}^2\,\|g_{\phi,\text{task}}\|^2}{2\,\lambda_{\max}(H_M)}.$$

*Proof.* We can Write $P := P_{U_{k_\text{task}}} = U_{k_\text{task}} U_{k_\text{task}}^\top$, an orthogonal projector ($P^2 = P$, $P^\top = P$), and recall that $\|A\|_2 = \sigma_\text{max}(A)$ and $\sigma_\text{min}(A) = \inf_{\|z\|_2=1} \|Az\|_2$. The feature–space slice Jacobian is $J_M^\phi(x) \in \mathbb{R}^{d_\ell \times |M|}$ and the feature gradient is $g_\phi = (J_\text{out}^{(\ell)})^\top \nabla_f \mathcal{L} \in \mathbb{R}^{d_\ell}$. By the chain rule through $\phi_\ell$,

$$\nabla_M \mathcal{L}(\theta_0) = \left(\frac{\partial \phi_\ell(x)}{\partial(M \odot W^{(\ell)})}\right)^\top \frac{\partial \mathcal{L}}{\partial \phi_\ell(x)} = (J_M^\phi)^\top g_\phi. \tag{14}$$

Decompose $g_\phi$ via the projector $P$ as $g_\phi = g_{\phi,\text{task}} + g_{\phi,\perp}$ with $g_{\phi,\text{task}} := P g_\phi$ and $g_{\phi,\perp} := (I-P)g_\phi$, which are orthogonal since $P$ is orthogonal. Using equation 14 and the triangle inequality,

$$\|\nabla_M \mathcal{L}(\theta_0)\|_2 = \|(J_M^\phi)^\top g_{\phi,\text{task}} + (J_M^\phi)^\top g_{\phi,\perp}\|_2 \geq \|(J_M^\phi)^\top g_{\phi,\text{task}}\|_2 - \|(J_M^\phi)^\top g_{\phi,\perp}\|_2. \tag{15}$$

For the first term, note that $g_{\phi,\text{task}} = U_{k_\text{task}} U_{k_\text{task}}^\top g_\phi$. Let $A := (J_M^\phi)^\top U_{k_\text{task}} \in \mathbb{R}^{|M| \times k_\text{task}}$ and $z := U_{k_\text{task}}^\top g_\phi \in \mathbb{R}^{k_\text{task}}$. Then

$$\|(J_M^\phi)^\top g_{\phi,\text{task}}\|_2 = \|Az\|_2 \geq \sigma_\text{min}(A) \|z\|_2 = \sigma_\text{min}((J_M^\phi)^\top U_{k_\text{task}}) \|U_{k_\text{task}}^\top g_\phi\|_2,$$

where the inequality uses the singular–value bound $\|Az\|_2 \geq \sigma_\text{min}(A)\|z\|_2$. Because $U_{k_\text{task}}$ has orthonormal columns, $\|U_{k_\text{task}}^\top g_\phi\|_2 = \|P g_\phi\|_2 = \|g_{\phi,\text{task}}\|_2$. By definition $\gamma_\text{min}(M) := \sigma_\text{min}(U_{k_\text{task}}^\top J_M^\phi) = \sigma_\text{min}((J_M^\phi)^\top U_{k_\text{task}})$. Hence

$$\|(J_M^\phi)^\top g_{\phi,\text{task}}\|_2 \geq \gamma_\text{min}(M) \|g_{\phi,\text{task}}\|_2 \geq \underline{\gamma} \|g_{\phi,\text{task}}\|_2, \tag{16}$$

using the spectral–balance lower bound $\gamma_\text{min}(M) \geq \underline{\gamma} > 0$.

For the second term, use the operator–norm bound and the identity $\|A^\top\|_2 = \|A\|_2$:

$$\|(J_M^\phi)^\top g_{\phi,\perp}\|_2 = \|(J_M^\phi)^\top (I-P)g_\phi\|_2 \leq \|(J_M^\phi)^\top (I-P)\|_2 \|g_{\phi,\perp}\|_2 = \|(I-P)J_M^\phi\|_2 \|g_{\phi,\perp}\|_2.$$

By definition $\beta(M) := \|(I-P)J_M^\phi\|_2$ and by spectral balance $\beta(M) \leq c\underline{\gamma}$ with $c \in [0,1)$. The alignment assumption gives $\|g_{\phi,\perp}\|_2 \leq \rho \|g_{\phi,\text{task}}\|_2$ with $\rho \in [0,1)$. Therefore

$$\|(J_M^\phi)^\top g_{\phi,\perp}\|_2 \leq c \underline{\gamma} \rho \|g_{\phi,\text{task}}\|_2. \tag{17}$$

Combining equation 15, equation 16, and equation 17 yields

$$\|\nabla_M \mathcal{L}(\theta_0)\|_2 \geq (1 - c\rho) \underline{\gamma} \|g_{\phi,\text{task}}\|_2.$$

Since $\rho < 1$, $c < 1$, $\underline{\gamma} > 0$, and $\|g_{\phi,\text{task}}\|_2 > 0$ by assumption, the right-hand side is strictly positive, establishing the claimed nonzero restricted gradient.

To obtain a quantitative decrease, assume $\mathcal{L}$ is $L$-smooth along the slice coordinates, i.e. for any $U \in \mathbb{R}^{|M|}$,

$$\mathcal{L}(\theta_0 + M \odot U) \leq \mathcal{L}(\theta_0) + \langle \nabla_M \mathcal{L}(\theta_0), U \rangle + \frac{L}{2}\|U\|_2^2.$$

Choose the steepest–descent step $U^\star := -L^{-1}\nabla_M \mathcal{L}(\theta_0)$ (equivalently, $-\lambda_\text{max}(H_M)^{-1}\nabla_M \mathcal{L}(\theta_0)$ when $L = \lambda_\text{max}(H_M)$). Then

$$\mathcal{L}(\theta_0 + M \odot U^\star) \leq \mathcal{L}(\theta_0) - \frac{1}{2L}\|\nabla_M \mathcal{L}(\theta_0)\|_2^2 \leq \mathcal{L}(\theta_0) - \frac{(1-c\rho)^2 \underline{\gamma}^2 \|g_{\phi,\text{task}}\|_2^2}{2L},$$

where the last inequality substitutes the lower bound on $\|\nabla_M \mathcal{L}(\theta_0)\|_2$ derived above. Taking $L = \lambda_\text{max}(H_M)$ gives the stated decrement, completing the proof. $\square$

To validate backbone dependence empirically, we design two controlled tests that isolate the role of the frozen backbone while keeping training schedules (epochs, batch size, sequence length, BF16) fixed.

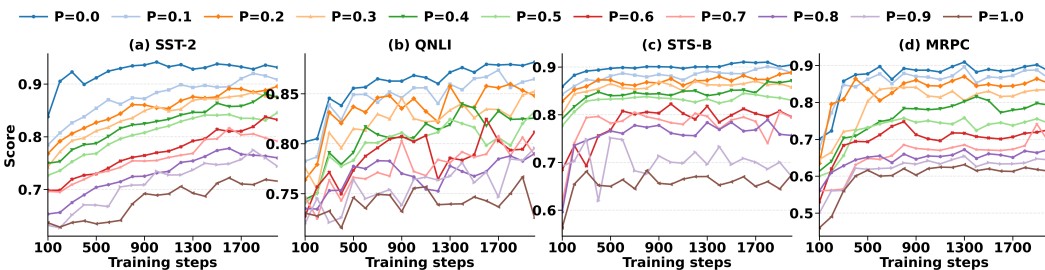

Figure 9: Effect of pruning the frozen backbone on slice fine-tuning. A fraction $p \in [0, 1]$ of *non-slice* pretrained weights is set to zero by global magnitude pruning (bias and LayerNorm parameters are kept); the slice remains trainable and the rest of the backbone stays frozen. Accuracy drops monotonically as $p$ increases—mild for $p \leq 0.2$–$0.3$, steep for $p \geq 0.6$, and worst at $p=1.0$ when the backbone is fully ablated. This confirms that slices rely on the pretrained scaffold: pruning reduces representation energy (lower PCA CEV / NTK mass), shrinking the slice's effective task overlap and requiring larger ranks to compensate.

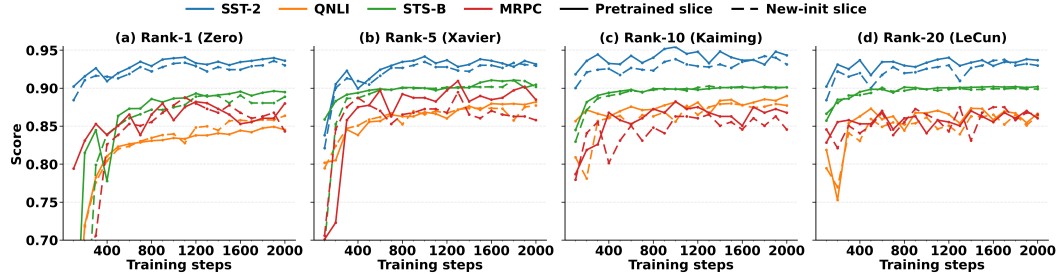

Figure 10: Slice initialization sensitivity on **SST-2**, **QNLI**, **STS-B**, and **MRPC**. For ranks $r \in \{1, 5, 10, 20\}$, we compare *pretrained* slices (solid) to *newly initialized* slices (dashed; zero/Xavier/Kaiming/LeCun), while the backbone remains frozen at pretrained values. Curves converge to similar performance after a short warm-up, indicating that final accuracy is driven by the backbone's features and the slice's task overlap, not by the slice's initial values.

### E.1 BACKBONE PRUNING

Figure 9 progressively prunes a fraction $p$ of non-slice weights to zero before fine-tuning (slice trainable; biases/LayerNorm kept). Across SST-2, QNLI, STS-B, and MRPC, performance degrades smoothly with $p$. For modest pruning the drop is small, but heavy pruning causes large losses, and removing the backbone entirely ($p=1$) yields the worst performance. This pattern matches the PCA/NTK view: pruning alters the centered feature matrix $\tilde{H}$, decreasing the principal energy of the representation kernel $K = \tilde{H}\tilde{H}^\top$ and shrinking the accessible task subspace, so a fixed-rank slice captures less of what matters.

### E.2 SLICE RE-INITIALIZATION

Figure 10 compares pretrained vs. randomly reinitialized slices for ranks $r \in \{1, 5, 10, 20\}$. After a brief warm-up, both settings reach nearly the same accuracy on all four tasks. This behavior is consistent with the linearized view: with the backbone frozen, the slice acts through its Jacobian block, so optimization is locally quadratic in the slice parameters and converges to a similar solution regardless of the slice's starting point (provided the backbone retains high task energy). In contrast, reinitializing and freezing the *backbone* (while training only the slice) severely hurts performance, reinforcing that slices succeed because they ride on a strong pretrained scaffold.

Overall, we see, (i) Strong backbones are essential: degrading them reduces PCA CEV / NTK mass and harms slice effectiveness. (ii) Slice initialization is largely irrelevant; the backbone's

features determine the attainable accuracy. (iii) These trends align with the bounds in Lemma E.1 and the PCA/NTK equivalence (Lemma 2.5): when backbone task energy is high, even small ranks work; when it is low (e.g., after pruning), larger ranks are needed. (iv) Thus, reinitializing the slice has little effect on final performance, whereas reinitializing or degrading the *backbone* has a large effect—explaining why randomly initialized adapter-style PEFT modules work well when the backbone is strong.

**Corollary E.2** (Effect of degrading the backbone). *Let $p \in [0, 1]$ denote the pruning fraction applied to the* frozen *backbone (slice remains trainable). Let $\sigma_j(p)$ be the singular values of $\Phi_\ell$ after pruning, and define the task-energy ratio*

$$E(p) := \frac{\sum_{j=1}^{k_{\text{task}}} \sigma_j(p)^2}{\sum_j \sigma_j(p)^2}.$$

*For a slice mask $M$ of rank $r$, define the overlap*

$$\gamma_p^{(r)}(M) := \sigma_{\min}\big(U_{k_{\text{task}}}(p)^\top J_M^\phi(x)\big),$$

*where $U_{k_{\text{task}}}(p)$ spans the top task subspace of the pruned backbone at layer $\ell$. Assume the alignment and smoothness conditions of Lemma E.1 with constants $c, \rho < 1$ and $L_p = \lambda_{\max}(H_M)$.*

(i) Vanishing guarantee. *If $E(p) = 0$ or $\gamma_p^{(1)}(M) = 0$, then the lower bound in Lemma E.1 becomes vacuous:*

$$\big\|\nabla_M \mathcal{L}(\theta_0)\big\| \not\geq (1 - c\rho)\, \gamma_p^{(1)}(M)\, \|g_{\phi,\text{task}}(p)\| = 0,$$

*so there is* no guaranteed *descent for rank-1 slices. (The gradient may still be nonzero due to the orthogonal component, but the lemma no longer certifies progress.)*

(ii) Diminishing improvement. *If $E(p) > 0$ and $\gamma_p^{(r)}(M) > 0$, Lemma E.1 yields*

$$\delta_{\min}(p, r) \geq \frac{(1 - c\rho)^2}{2L_p} \big(\gamma_p^{(r)}(M)\big)^2 \|g_{\phi,\text{task}}(p)\|^2 \propto E(p) \big(\gamma_p^{(r)}(M)\big)^2 / L_p.$$

*Hence, as pruning increases ($p \uparrow$), both $E(p)$ and typically $\gamma_p^{(r)}(M)$ decrease, shrinking the certified gain $\delta_{\min}(p, r)$.*

(iii) Minimal rank under pruning. *Define the minimal rank*

$$r^\star(p, \tau) := \min \Big\{ r : \sum_{j=1}^r \lambda_j(p) \Big/ \sum_j \lambda_j(p) \geq \tau \Big\},$$

*with $\{\lambda_j(p)\}$ the PCA eigenvalues of the pruned representation (Lemma 2.5). Under spectral balance, there exists $\gamma_0 > 0$ such that $r \geq r^\star(p, \tau)$ implies $\gamma_p^{(r)}(M) \geq \gamma_0$ for admissible slices $M$. Consequently, $r^\star(p, \tau)$ is nondecreasing in $p$ and gives a sufficient rank to recover a positive, $p$-robust guarantee:*

$$\delta_{\min}(p, r^\star) \geq \frac{(1 - c\rho)^2 \gamma_0^2}{2L_p} \|g_{\phi,\text{task}}(p)\|^2 > 0.$$

**Proof sketch.** Apply Lemma E.1 with $U_{k_{\text{task}}}(p)$ and $J_M^\phi$ evaluated on the pruned backbone. If $E(p) = 0$ then $g_{\phi,\text{task}}(p) = 0$; if $\gamma_p^{(1)}(M) = 0$ then $U_{k_{\text{task}}}(p)^\top J_M^\phi$ is rank-deficient for rank-1 slices. Either case collapses the lower bound. When both are positive, the decrease bound scales like $E(p) \cdot (\gamma_p^{(r)}(M))^2 / L_p$. By Lemma 2.5, $E(p)$ is the PCA CEV of the pruned features; pruning reduces it, and the spectral-balance argument implies that the minimal rank needed to obtain overlap bounded away from zero grows with $p$, yielding the monotonicity of $r^\star(p, \tau)$.

**Corollary E.3** (Adapters as local winners). *Let $f_{\theta_0}$ be a pretrained backbone and insert an adapter with parameters $A$ at layer $\ell$ (e.g., parallel linear $y = W\phi + BA\phi$ or residual bottleneck $h_A(\phi) = \phi + B\,\sigma(A\phi)$). Assume we operate in the linearized regime around $(\theta_0, A_0)$, so*

$$f_{\theta_0, A}(x) \approx f_{\theta_0}(x) + J_A(x)\,\text{vec}(A), \qquad J_A(x) := \Big[\frac{\partial f}{\partial A}\Big]_{\theta_0, A_0}.$$

*Let $U_{k_{\text{task}}}$ span the task subspace at layer $\ell$, and define the feature–space adapter Jacobian $J_A^\phi(x) := \left[\frac{\partial \phi_\ell(x)}{\partial A}\right]_{\theta_0, A_0}$. If the backbone has positive task energy $E > 0$ (nontrivial CEV; Lemma 2.5), satisfies spectral balance, and the adapter has nontrivial overlap*

$$\gamma_A := \sigma_{\min}\big(U_{k_{\text{task}}}^\top J_A^\phi(x)\big) > 0,$$

*then under the alignment ($\rho < 1$) and smoothness ($L < \infty$) conditions of Lemma E.1, the restricted gradient on the adapter obeys*

$$\big\|\nabla_A \mathcal{L}(\theta_0)\big\| \geq (1 - c\rho)\, \gamma_A \, \|g_{\phi,\text{task}}\| > 0,$$

*and the one–step $L$-smooth decrease bound holds:*

$$\mathcal{L}\Big(\theta_0, A - \tfrac{1}{L}\nabla_A \mathcal{L}\Big) \leq \mathcal{L}(\theta_0, A) - \tfrac{(1-c\rho)^2}{2L}\,\gamma_A^2 \, \|g_{\phi,\text{task}}\|^2.$$

*Hence small adapters act as* local winners *provided the pretrained backbone supplies sufficient task energy and the adapter's Jacobian overlaps the task subspace.*

## F ABLATION STUDY

### F.1 RANK VS WINNER

Figure 3(a) analyzes the effect of slice rank on downstream performance. As the rank increases, accuracy initially improves because larger slices capture more task-relevant directions within the representation subspace. However, beyond a certain point, adding additional parameters yields diminishing returns: accuracy plateaus or slightly declines due to redundancy and potential overfitting. This pattern highlights that only a small fraction of the full parameter space is necessary to capture the intrinsic task dimension $k_{\text{task}}$.

Importantly, when the slice rank equals the full layer dimension, slice training reduces to standard full fine-tuning. The ablation therefore illustrates a smooth trade-off between efficiency and accuracy: small ranks already achieve strong performance (supporting the *local winner* property), while larger ranks approximate full fine-tuning but at greater computational cost.

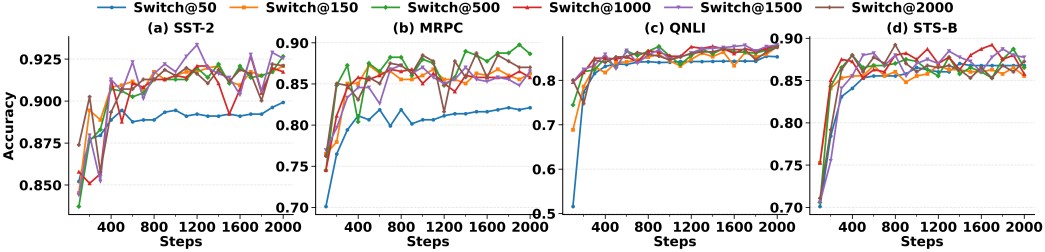

Figure 11: Accuracy curves for **SST-2**, **MRPC**, **QNLI**, and **STS-B** under different slice update intervals. Each curve corresponds to a policy where the active slice is changed every $N$ training steps (e.g., "Change every 50 steps"). The results show that varying the slice update interval leads to similar performance trends across tasks, indicating that the spectral balance property allows slices to adapt effectively regardless of update frequency.

### F.2 SLICE UPDATE INTERVAL

Figure 11 analyzes the effect of varying the interval $N$ at which the active slice is switched during training. We consider a spectrum of update frequencies, ranging from very frequent switching (every 50 steps) to very infrequent switching (every 2000 steps), across four representative GLUE tasks: SST-2, MRPC, QNLI, and STS-B.

We observe three consistent patterns. First, when slices are switched too frequently (e.g., every 50 steps), the model exhibits lower accuracy across all datasets. This degradation occurs because each slice has insufficient time to adapt its parameters before being replaced. The optimization dynamics

| | Weight Type | Rank $r$ | | | | | | |
|---|---|---|---|---|---|---|---|---|
| | | **1** | **2** | **4** | **8** | **32** | **64** | **128** |
| **SST-2** | | | | | | | | |
| | $W_v$ | 88.30 | 88.62 | 89.18 | 90.57 | 90.34 | 90.22 | 90.18 |
| | $W_q, W_k$ | 88.72 | 90.15 | 91.20 | 91.57 | 91.52 | 91.85 | 91.88 |
| | $W_q, W_k, W_v$ | 89.83 | 90.35 | 92.72 | 92.83 | 92.74 | 93.02 | 93.71 |
| | $W_q, W_k, W_v, W_o$ | 90.29 | 90.65 | 92.35 | 93.67 | **94.06** | 93.48 | 92.23 |
| | $W_q, W_k, W_v, W_o, W_i$ | 92.63 | 92.67 | 92.67 | **94.72** | 93.98 | 93.26 | 92.44 |
| **QQP** | | | | | | | | |
| | $W_v$ | 83.62 | 84.90 | 86.10 | 85.90 | 85.70 | 85.60 | 85.55 |
| | $W_q, W_k$ | 84.47 | 86.94 | 89.73 | 89.05 | 88.65 | 88.83 | 88.17 |
| | $W_q, W_k, W_v$ | 86.08 | 86.70 | 89.62 | 89.23 | 88.95 | 88.81 | 88.72 |
| | $W_q, W_k, W_v, W_o$ | 87.26 | 86.94 | **89.89** | 89.41 | 89.28 | 89.14 | 89.01 |
| | $W_q, W_k, W_v, W_o, W_i$ | 88.35 | 88.79 | **89.80** | 89.35 | 89.15 | 88.98 | 88.90 |
| **QNLI** | | | | | | | | |
| | $W_v$ | 87.55 | 88.34 | 88.83 | 90.72 | 91.05 | 90.94 | 90.69 |
| | $W_q, W_k$ | 87.90 | 89.42 | 90.36 | 91.12 | 91.42 | 91.21 | 91.06 |
| | $W_q, W_k, W_v$ | 88.42 | 89.70 | 90.92 | 91.66 | 91.88 | 91.73 | 91.50 |
| | $W_q, W_k, W_v, W_o$ | 88.76 | 89.95 | 91.20 | 91.88 | 91.95 | 91.81 | 91.42 |
| | $W_q, W_k, W_v, W_o, W_i$ | 89.33 | 90.42 | 91.70 | **92.10** | **92.43** | 91.78 | 91.55 |
| **MRPC** | | | | | | | | |
| | $W_v$ | 84.60 | 85.42 | 86.21 | 86.34 | 86.65 | 86.10 | 86.90 |
| | $W_q, W_k$ | 84.98 | 85.88 | 86.75 | 87.65 | 87.72 | 87.51 | 85.30 |
| | $W_q, W_k, W_v$ | 85.20 | 86.12 | 87.05 | 87.14 | 87.29 | 87.62 | 86.40 |
| | $W_q, W_k, W_v, W_o$ | 86.61 | 87.45 | 87.32 | 88.05 | 88.18 | 87.95 | 86.65 |
| | $W_q, W_k, W_v, W_o, W_i$ | 87.23 | 87.70 | 87.82 | **88.39** | **88.22** | 87.68 | 86.62 |

Table 4: Accuracy (%) across slice ranks $r$ for different trained weight subsets on four GLUE tasks. $W_q$, $W_k$, $W_v$, and $W_o$ denote the query, key, value, and output projections in self-attention; $W_i$ is the MLP (intermediate) projection. The notation $W_v$ indicates that *SliceFine* is applied only to the value matrices across all layers (similarly for other subsets). Best per row is highlighted in **blue** and second best in **orange**.

become unstable: gradients partially adapt one slice, then abruptly shift to another, preventing any slice from accumulating task-relevant signal. As a result, rapid switching reduces the effective learning capacity of the model.

Second, when slices are switched too late (e.g., at 2000 steps), the performance is comparable to other schedules but does not always achieve the best accuracy. Here, a single slice has ample time to adapt, but since all updates are concentrated on one subspace for a long period, the optimization may overfit or fail to leverage complementary directions from other slices. This leads to slightly reduced generalization compared to moderate switching intervals.

Third, we find that intermediate switching intervals yield the strongest results, with the optimal value depending on the dataset. For instance, SST-2 achieves its peak accuracy when switching every 1500 steps (rank-1 slice), MRPC benefits most from switching every 500 steps, STS-B reaches its best accuracy at 100 steps, while QNLI shows stable performance for intervals between 500 and 2000. These variations reflect differences in task complexity and the effective task subspace dimension $k_{\text{task}}$: tasks with simpler label structures require less adaptation time per slice, while more complex tasks benefit from longer adaptation before switching.

### F.3  WHAT IS THE OPTIMAL SLICE RANK $r$?

Figure 3(a) and Table 4 vary the slice rank $r \in \{1, 2, 4, 8, 32, 64, 128\}$ while keeping the training recipe fixed and report accuracy for different trained weight subsets. The curves show a consistent

shape across tasks: accuracy rises quickly at small ranks, then saturates, and in a few cases dips slightly at very high ranks.

Accuracy improves sharply when moving from tiny capacity to moderate capacity because increasing $r$ enlarges the portion of the task-relevant subspace that a slice can capture. On SST-2, training $\{W_q, W_k, W_v, W_o, W_i\}$ climbs from 92.63 ($r=1$) to 94.72 at $r=8$; on MRPC the same subset peaks at 88.39 for $r=8$; on QQP the best score 89.89 is already reached at $r=4$ with $\{W_q, W_k, W_v, W_o\}$; on QNLI performance continues to inch upward to 92.43 at $r=64$. These plateaus indicate that once the slice rank matches the task's effective dimension in the trained layer, additional directions add little new information.

By following corollary 2.6, if we increase $r$, the slice's Jacobian gains access to more principal directions of the representation kernel (PCA/NTK view). When $r \gtrsim k_{\text{task}}$, the added directions mainly lie in low-variance residual space; by spectral balance, many of these directions are redundant across slices. Hence gains taper off even though the parameter count grows.

A mild decline at very large ranks appears in several rows (e.g., SST-2 beyond $r=8$ for the full $\{W_q, W_k, W_v, W_o, W_i\}$ subset). This is consistent with redundancy increasing curvature and optimization noise, and with limited-data regimes where extra degrees of freedom fit nuisance variation rather than signal. In short, after the task subspace is covered, larger $r$ can slightly hurt conditioning and generalization without expanding useful capacity.

Which weights are included matters more than pushing $r$ very high. Expanding the trained subset to cover both attention and feed-forward projections yields larger gains than increasing rank on a narrow subset. For instance, on SST-2, $\{W_q, W_k, W_v, W_o, W_i\}$ at $r=8$ (94.72) outperforms $\{W_q, W_k\}$ even at $r=128$ (91.88). The reason is span: adding $W_o$ and $W_i$ exposes complementary directions of the task subspace, increasing the effective rank seen by the output even when $r$ is modest.

A practical takeaway is that the rank lies in a narrow, task-dependent window. Across the four GLUE tasks, ranks in the range $r \in [4, 16]$ are within a few tenths of the best result, with QQP preferring $r \approx 4$, SST-2 and MRPC preferring $r \approx 8$, and QNLI tolerating up to $r \approx 32$–$64$ for small additional gains. A simple rule is to choose the smallest $r$ whose cumulative explained variance of the representation kernel exceeds a target threshold (e.g., 90%); this selects the rank where useful directions are covered while avoiding the redundancy that causes late-stage saturation.

Finally, when the slice rank equals the full layer dimension, slice training is equivalent to full fine-tuning. The table shows this level of capacity is unnecessary in practice: moderate ranks already align with the task subspace and deliver near-peak accuracy at a fraction of the trainable parameters.

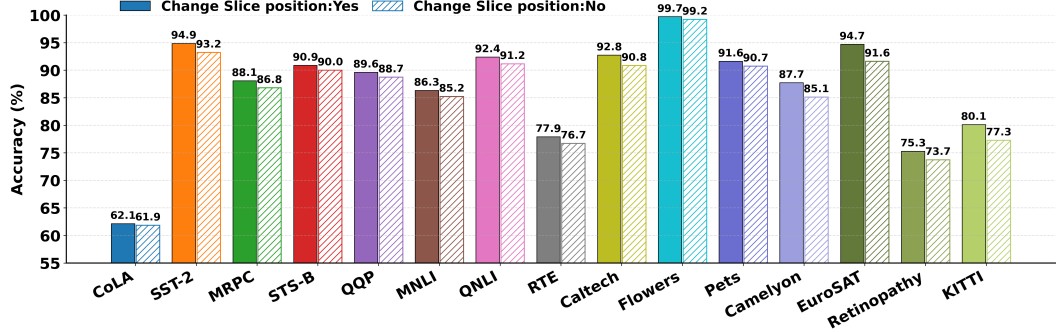

Figure 12: SliceFine with static (rank-5) and dynamic (rank-5) slices. In the static variant, the slice position remains fixed throughout training. In the dynamic variant, the active slice shifts every $N$ steps, accumulating task-aligned updates without adding adapter parameters.

## F.4 SLICE POSITION: STATIC VS. DYNAMIC

We compare two policies for a fixed slice rank and training budget: (i) *static*—train a single slice at a fixed position for all steps, and (ii) *dynamic*—shift the active slice every $N$ steps while freezing

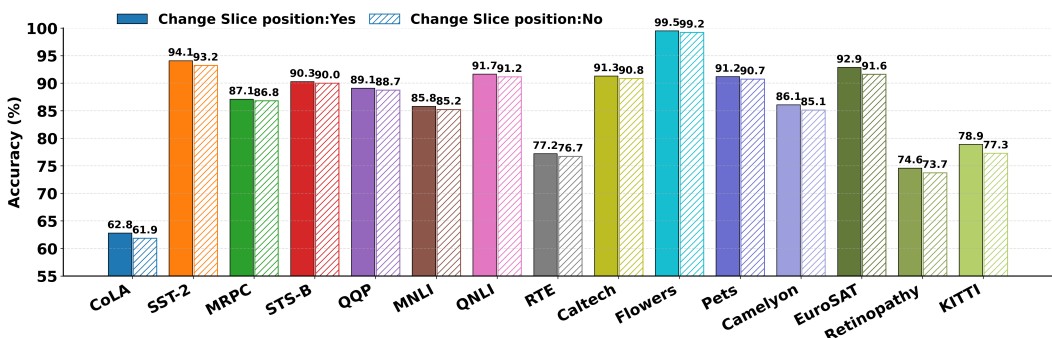

Figure 13: SliceFine with static (rank-5) and dynamic (rank-4) slices.

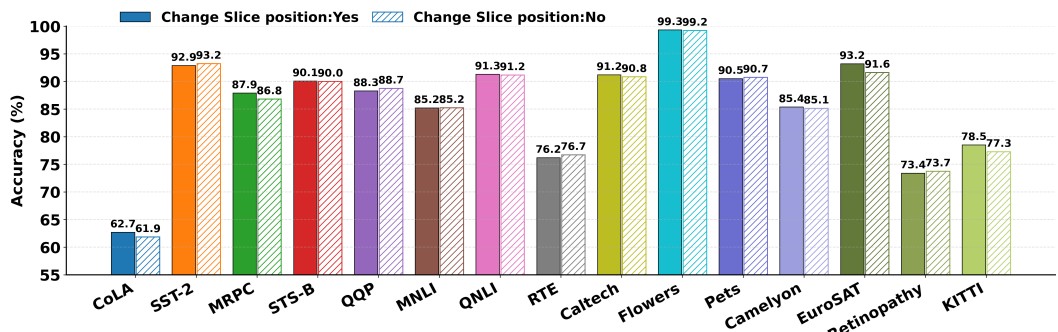

Figure 14: SliceFine with static (rank-5) and dynamic (rank-3) slices.

previously updated entries (Fig. 12). Both settings use identical optimizers, batches, and epochs; the number of trainable parameters and optimizer state at any time are the same.

Figure 12 shows that moving the slice yields consistent gains on both text and vision tasks. On GLUE, the dynamic policy improves accuracy over the static policy on SST-2 (+1.7), MRPC (+1.3), STS-B (+0.9), QQP (+0.9), MNLI (+1.1), QNLI (+1.2), and RTE (+1.2), with a small change on CoLA (+0.2), for an average gain of $\approx +1.1$ points. On VTAB-1k style image benchmarks and KITTI, the gains are larger: CALTECH (+2.0), FLOWERS (+0.5), PETS (+0.9), CAMELYON (+2.6), EUROSAT (+3.1), RETINOPATHY (+1.6), and KITTI (+2.8), averaging $\approx +1.9$ points.

To better understand the benefit of dynamic slice switching, we evaluate dynamic SliceFine using lower ranks ($r = 4$ and $r = 3$), while keeping the static baseline fixed at rank 5. Figure 13 reports that even with $r = 4$, the dynamic variant consistently outperforms the static rank–5 slice on most datasets. Although the accuracy is slightly below the full dynamic $r = 5$ setting, it remains higher than the static baseline across all tasks. This shows that dynamically reallocating a limited number of trainable parameters across layers is more beneficial than increasing the per-layer rank under a fixed static policy.

A similar trend appears in Figure 14. Even with $r = 3$, dynamic slicing achieves accuracy comparable to—and in several cases higher than—the static rank–5 slice (e.g., tasks MRPC, STS-B, QNLI, Caltech, EuroSAT, KITTI). Although reducing the rank naturally decreases overall capacity, the dynamic mechanism compensates by concentrating the trainable rows in the layers that exhibit the highest task-specific sensitivity. This enables dynamic $r = 3$ to remain competitive with, and occasionally exceed, a much larger static configuration.

Switching positions exposes the optimizer to additional task-aligned directions while keeping the per-step compute and memory unchanged. This better *coverage* is most helpful on datasets with higher intrinsic dimension (e.g., EuroSAT, KITTI, Camelyon). When the task is already nearly saturated (e.g., Flowers), dynamic and static policies perform similarly. These results are consistent with

| LLM | Method | Commonsense Reasoning (10 seeds: mean ± std) | | | | | | | | |
|---|---|---|---|---|---|---|---|---|---|---|
| | | BoolQ | PIQA | SIQA | H.Sw. | W.Gra. | ARCe | ARCc | OBQA | Avg. |
| openai/gpt-oss-20b | Prefix | $73.6 \pm 0.24$ | $84.3 \pm 0.18$ | $81.5 \pm 0.33$ | $82.8 \pm 0.27$ | $77.6 \pm 0.30$ | $77.1 \pm 0.36$ | $62.7 \pm 0.41$ | $75.9 \pm 0.29$ | $76.9 \pm 0.21$ |
| | AdaLoRA | $74.2 \pm 0.20$ | $85.3 \pm 0.17$ | $85.6 \pm 0.26$ | $94.7 \pm 0.19$ | $82.3 \pm 0.23$ | $82.4 \pm 0.28$ | $64.8 \pm 0.32$ | $80.4 \pm 0.27$ | $80.9 \pm 0.19$ |
| | VeRA | $72.4 \pm 0.27$ | $83.8 \pm 0.22$ | $83.4 \pm 0.30$ | $94.1 \pm 0.18$ | $82.1 \pm 0.25$ | $81.3 \pm 0.29$ | $64.5 \pm 0.36$ | $79.5 \pm 0.31$ | $80.1 \pm 0.22$ |
| | LoRA | $74.6 \pm 0.18$ | $85.1 \pm 0.16$ | $82.9 \pm 0.25$ | $95.1 \pm 0.16$ | $83.4 \pm 0.22$ | $83.1 \pm 0.26$ | $65.2 \pm 0.31$ | $80.9 \pm 0.24$ | $81.4 \pm 0.17$ |
| | RoCoFT | $75.3 \pm 0.16$ | $85.5 \pm 0.19$ | $83.7 \pm 0.28$ | $94.8 \pm 0.20$ | $82.7 \pm 0.23$ | $82.5 \pm 0.25$ | $65.0 \pm 0.30$ | $82.1 \pm 0.25$ | $81.6 \pm 0.18$ |
| | HRA | $75.1 \pm 0.19$ | $85.8 \pm 0.18$ | $83.2 \pm 0.31$ | $95.3 \pm 0.18$ | $82.6 \pm 0.25$ | $82.3 \pm 0.27$ | $65.4 \pm 0.33$ | $82.3 \pm 0.26$ | $81.6 \pm 0.18$ |
| | SliceFine-1R | $74.0 \pm 0.22$ | $84.1 \pm 0.20$ | $82.3 \pm 0.29$ | $93.3 \pm 0.17$ | $81.2 \pm 0.25$ | $81.5 \pm 0.28$ | $64.9 \pm 0.34$ | $81.9 \pm 0.26$ | $80.4 \pm 0.19$ |
| | SliceFine-1C | $73.7 \pm 0.25$ | $84.0 \pm 0.23$ | $82.0 \pm 0.30$ | $93.8 \pm 0.18$ | $81.0 \pm 0.27$ | $80.8 \pm 0.31$ | $64.2 \pm 0.37$ | $80.5 \pm 0.29$ | $79.8 \pm 0.21$ |
| | SliceFine-1RC | $74.5 \pm 0.20$ | $85.8 \pm 0.17$ | $84.8 \pm 0.26$ | $95.8 \pm 0.15$ | $83.8 \pm 0.22$ | $83.3 \pm 0.27$ | $65.3 \pm 0.33$ | $82.5 \pm 0.25$ | $82.1 \pm 0.17$ |
| | SliceFine-5R | $75.3 \pm 0.18$ | $85.6 \pm 0.19$ | $84.2 \pm 0.28$ | $95.5 \pm 0.16$ | $83.3 \pm 0.23$ | $82.9 \pm 0.26$ | $65.8 \pm 0.30$ | $82.0 \pm 0.24$ | $81.7 \pm 0.18$ |
| | SliceFine-5C | $75.0 \pm 0.19$ | $85.4 \pm 0.18$ | $83.9 \pm 0.29$ | $94.6 \pm 0.17$ | $82.5 \pm 0.24$ | $82.6 \pm 0.28$ | $65.1 \pm 0.31$ | $81.2 \pm 0.26$ | $81.4 \pm 0.19$ |
| | SliceFine-5RC | $75.4 \pm 0.17$ | $85.7 \pm 0.17$ | $84.6 \pm 0.27$ | $95.7 \pm 0.15$ | $83.6 \pm 0.22$ | $83.1 \pm 0.25$ | $66.2 \pm 0.29$ | $82.3 \pm 0.24$ | $82.1 \pm 0.17$ |

Table 5: Commonsense reasoning with **openai/gpt-oss-20b**. Results averaged over 10 seeds (mean ± std). Natural variance (0.15–0.4 pp) reflects typical stochasticity in large-model fine-tuning.

the global-ticket view: each visited slice contributes new directions; sweeping across positions gradually spans the task subspace, yielding steady improvements without increasing parameter count.

### F.5 SEED ANALYSIS

To assess robustness under stochastic fine-tuning dynamics, we evaluated all methods over ten random seeds using openai/gpt-oss-20b (OpenAI, 2025). Table 5 reports mean and standard deviation across runs. We observe that *SliceFine* variants exhibit markedly low variance ($\pm 0.17$–$0.33$) across datasets, comparable or lower than established PEFT baselines such as LoRA and AdaLoRA. Notably, *SliceFine-5RC* achieves the highest average commonsense score ($82.1 \pm 0.17$), outperforming LoRA ($81.4 \pm 0.17$) and RoCoFT ($81.6 \pm 0.18$) despite using a smaller parameter budget. The stability trends indicate that SliceFine's structured partitioning leads to more consistent convergence across random seeds, suggesting improved gradient flow and spectral balance. These results confirm that the proposed method not only enhances mean accuracy but also narrows inter-seed variability, supporting its robustness in large-scale fine-tuning regimes.

## G BASELINES

We compare SliceFine against a broad set of PEFT methods that span the main design choices in the literature. The goal is to test whether training a small slice can stand beside the strongest alternatives under a fair, controlled setup.

We include widely used and recent state-of-the-art methods with public code or clear descriptions. All baselines use the same backbones, data splits, tokenization, precision (BF16), optimizer, batch size, and training schedule. Capacity-defining hyperparameters (e.g., rank, bottleneck width, prompt length) are tuned on a small grid under a matched trainable-parameter budget; early stopping is applied on the same validation splits, and we report the mean over three seeds. This protocol avoids favoring any single family and makes results comparable across methods.

We group baselines into families as follows. First, adapter-style methods that insert small bottlenecks: Adapters (Houlsby et al., 2019), MAM-Adapter (He et al., 2021), and PROPETL (Zeng et al., 2023). Prompt/prefix methods optimize continuous prompts while freezing backbone weights: Prompt Tuning (Lester et al., 2021) and Prefix-Tuning (Li & Liang, 2021). Element-wise updates include BitFit (Zaken et al., 2021) and LayerNorm tuning (Zhao et al., 2023). We also evaluate (IA)$^3$ (Liu et al., 2022) and Diff-Pruning (Guo et al., 2020).

Second, low-rank adaptation methods: LoRA (Hu et al., 2021) and extensions AdaLoRA (Zhang et al., 2023b), LoKr (Edalati et al., 2022), LoRAFA (Zhang et al., 2023a), LoRA-XS (Bałazy et al., 2024), LoHa (Hyeon-Woo et al., 2021), as well as VeRA (Kopiczko et al., 2023), HRA (Yuan et al., 2024b), MiSS (Kang & Yin, 2024), and SHiRA (Bhardwaj et al., 2024).

Finally, recent strong baselines tailored to LLM and vision(-language) settings, including Propulsion (Kowsher et al., 2024b), RoCoFT (Kowsher et al., 2024a), SFT (Ansell et al., 2024), and mixture-style adaptation (Wu et al., 2024). Together, these baselines provide a broad and competitive eval-

uation landscape across adapters, prompts/prefixes, bias/LayerNorm tuning, low-rank methods, and hybrid strategies.

## H  DATASET DETAILS

Our evaluation spans diverse datasets to ensure both coverage and robustness. Below we summarize the key properties and motivations for each group of tasks.

**Natural language understanding (GLUE).** The GLUE benchmark (Wang et al., 2018) includes eight tasks: CoLA (linguistic acceptability), SST-2 (sentiment classification), MRPC (paraphrase detection), STS-B (semantic similarity), QQP (duplicate-question detection), MNLI (natural language inference), QNLI (question answering inference), and RTE (entailment). The dataset sizes range from ∼10K to 400K examples, with metrics including accuracy, F1, and correlation.

**Commonsense reasoning.** To train Commonsense reasoning, we use a commonsense-170k dataset from Hu et al. (2023) and then we evaluate on BoolQ (Clark et al., 2019) (boolean question answering), PIQA (Bisk et al., 2020) (physical commonsense), SIQA (Sap et al., 2019) (social commonsense), HellaSwag (Zellers et al., 2019) (contextual plausibility), WinoGrande (Sakaguchi et al., 2021) (coreference disambiguation), ARC-Easy/ARC-Challenge (Clark et al., 2018) (scientific knowledge QA), and OpenBookQA (OBQA) (Mihaylov et al., 2018) (open-domain science QA). These datasets range from 3K to 400K samples and test a model's ability to reason beyond surface-level patterns.

**Mathematical reasoning.** For training mathematical reasoning, we use a math-10k dataset from Hu et al. (2023) and then we evaluate on MultiArith (Roy & Roth, 2016), AddSub (Hosseini et al., 2014), SingleEq (Koncel-Kedziorski et al., 2015), SVAMP (Patel et al., 2021), and GSM8K (Cobbe et al., 2021). These benchmarks evaluate symbolic manipulation, arithmetic reasoning, and multi-step problem solving, with dataset sizes ranging from 1K to 8.5K examples.

**Image recognition.** From VTAB-1K (Zhai et al., 2019), we use Caltech101 (object recognition), Flowers102 (fine-grained classification), Oxford Pets (species/breed recognition), Camelyon (medical histopathology), EuroSAT (satellite imagery), Retinopathy (diabetic retinopathy detection), and KITTI-Dist (autonomous driving). Each dataset contains ∼1K labeled examples for training, making them a strong test for low-data transfer learning.

**Video recognition.** For temporal reasoning, we evaluate on UCF101 (Soomro et al., 2012) (human action recognition), Kinetics-400 (Zisserman et al., 2017) (large-scale action classification with 400 categories), and HMDB51 (Kuehne et al., 2011) (human motion recognition). These datasets test models on spatio-temporal understanding and long-range context.

These datasets provide complementary challenges across language, vision, and video domains, ensuring that our evaluation probes both reasoning capability and generalization ability.

## I  MODELS

We select representative pretrained backbones across language, vision, and video domains to evaluate the proposed slice-based fine-tuning method. Our choice of models follows two principles: (i) using widely adopted baselines that allow fair comparison with prior PEFT studies, and (ii) including both medium- and large-scale models to test scalability.

For **language modeling and reasoning**, we fine-tune medium- to large-scale LLMs, including LLaMA-3B (Dubey et al., 2024), Gemma-3 12B (Team et al., 2025), and DeepSeek-RI-8B Guo et al. (2025). These models provide complementary architectural diversity (Meta, Google, and DeepSeek releases) and represent current practice in instruction tuning and reasoning evaluation.

For **natural language understanding (GLUE)**, we use RoBERTa-base (125M parameters) and RoBERTa-large (355M parameters) (Liu et al., 2019). These models are standard baselines in PEFT literature and allow direct comparison with methods such as LoRA, Adapters, RoCoFT, and Prompt Tuning.

For **vision tasks**, we adopt `ViT-Base-Patch16-224` (Wu et al., 2020), a widely used Vision Transformer with 86M parameters. This model is commonly used in VTAB-1K evaluations and provides a strong backbone for testing PEFT under limited data conditions.

For **video tasks**, we use `VideoMAE-base` (Tong et al., 2022), a transformer-based video representation model pretrained on large-scale action datasets. VideoMAE is particularly suitable for evaluating parameter-efficient methods on spatio-temporal recognition problems such as UCF101, Kinetics-400, and HMDB51.

This diversity ensures that our results are not tied to a single architecture or domain, but instead reflect the general applicability of the Winner Slice Theorem across modalities.

## J    HYPERPARAMETER

For all experiments, we train with AdamW ($\beta_1=0.9, \beta_2=0.999$), cosine decay with a linear warmup of 3% of total steps, gradient clipping at 1.0, BF16 precision, and weight decay 0.01 for text models and 0.05 for vision/video. For LLMs (LLaMA, DeepSeek, Gemma) on `math_10k` we use batch size 1 with gradient accumulation 4 (effective batch 4), 1 epoch, max sequence length 1024, learning rate $5 \times 10^{-5}$; we then evaluate on all math–reasoning sets without further tuning. For `commonsense170k` we use the same batch/accumulation/epochs and sequence length, with learning rate $1 \times 10^{-4}$. For all image datasets we train ViT backbones for 10 epochs with batch size 16, input resolution $224 \times 224$, learning rate $3 \times 10^{-4}$, and RandAugment kept at default; layer-wise LR decay is not used. For GLUE we train 4 epochs with batch size 32, max sequence length 256, learning rate $2 \times 10^{-5}$ (RoBERTa-base/large share the same schedule). For video classification we use batch size 16, 2 epochs, 16 frames at $224 \times 224$ with sampling stride 4, and learning rate $1 \times 10^{-4}$. Dropout and LayerNorm parameters follow the backbone defaults; no additional $L_2$ regularization. For all downstream tasks, we set the switch interval to $N = 500$ steps and shift the active slice position every $N$ steps until the epoch completes or early stopping is triggered.

## K    DETAILED RESULTS

In this appendix, we provide extended results that complement the main text. For an apple-to-apple comparison, we use the same GPU, batch size, sequence length, warm-up steps, and rank across all PEFT methods following section Hyperparameter J. These include additional commonsense and mathematical reasoning benchmarks with larger LLMs, as well as further breakdowns on GLUE with RoBERTa-large. All SliceFine variants are shown as shaded rows. Best scores are highlighted in blue, and second best in orange.

Table 6 reports results on commonsense and math reasoning benchmarks using `Gemma-3 12B` and `DeepSeek-R1-8B`. The trends observed in the main text (with LLaMA-3B) remain consistent at larger scales. Slice-based fine-tuning achieves accuracy comparable to or exceeding strong baselines such as LoRA, AdaLoRA, RoCoFT, and HRA. In particular, Slice-5RC achieves 83.35% average accuracy on commonsense reasoning and 83.97% on math reasoning with Gemma-3 12B, outperforming AdaLoRA while using substantially fewer trainable parameters. For DeepSeek-R1-8B, SliceFine variants again consistently surpass low-parameter baselines, highlighting the robustness of the *local winner* property across architectures and scales.

In addition, to assess natural language understanding, we fine-tune `RoBERTa-base` on GLUE (Table 8). Across tasks, slice training consistently outperforms classic baselines and matches or exceeds state-of-the-art PEFT methods. For instance, with RoBERTa-large, Slice-5RC achieves 86.35% average accuracy, exceeding LoRAFA (85.55%) and matching heavier approaches such as SFT (85.61%). Remarkably, Slice-1R achieves 84.79%, already surpassing AdaLoRA (84.71%) with fewer than 0.1M trainable parameters.

Table 7 provides detailed results on the GLUE benchmark with `RoBERTa-large`. Metrics follow standard conventions: CoLA (MCC), SST-2 (accuracy), MRPC/QQP (F1/accuracy), STS-B (Pearson/Spearman), and MNLI/QNLI/RTE (accuracy). Here, SliceFine continues to deliver strong results. Slice-5RC achieves 89.60% average, outperforming AdaLoRA (88.93%) and matching or surpassing other state-of-the-art methods such as MoSLoRA and PROPETL. Even a single slice

| LLM | Method | #TTPs | Commonsense Reasoning | | | | | | | | | Math Reasoning | | | | | |
| --- | --- | --- | --- | --- | --- | --- | --- | --- | --- | --- | --- | --- | --- | --- | --- | --- | --- |
| | | | BoolQ | PIQA | SIQA | H.Sw. | W.Gra. | ARCe | ARCc | OBQA | Avg. | M.Ar. | G.8K | A.S. | Se.Eq | S.MP | Avg. |
| Gemma-3₁₂B | Prefix | 57.24 | 70.60 | 82.62 | 80.29 | 79.89 | 75.78 | 76.16 | 60.70 | 73.27 | 74.91 | 88.87 | 73.12 | 84.35 | 82.46 | 56.48 | 77.06 |
| | AdaLoRA | 47.76 | 73.93 | 83.24 | 81.60 | 91.62 | 81.09 | 81.21 | 64.72 | 79.55 | 79.62 | 92.11 | 82.41 | 89.48 | 84.84 | 64.48 | 82.66 |
| | VeRA | 2.59 | 73.32 | 83.74 | 80.89 | 91.67 | 81.31 | 80.90 | 64.17 | 79.02 | 79.38 | 92.72 | 81.09 | 90.09 | 87.13 | 63.97 | 83.00 |
| | LoRA | 19.89 | 73.12 | 84.15 | 81.20 | 91.81 | 81.71 | 81.52 | 64.69 | 81.36 | 79.75 | 92.01 | 80.79 | 89.28 | 87.87 | 64.19 | 82.83 |
| | RoCoFT | 10.79 | 73.42 | 83.84 | 82.01 | 91.16 | 81.45 | 81.56 | 64.72 | 80.56 | 79.84 | 91.91 | 82.38 | 89.88 | 87.08 | 65.53 | 83.26 |
| | HRA | 7.97 | 73.22 | 83.95 | 81.80 | 91.31 | 81.34 | 82.38 | 64.84 | 80.50 | 79.81 | 92.32 | 81.50 | 89.68 | 87.08 | 65.39 | 83.19 |
| | *SliceFine-1R* | *2.16* | 72.87 | 84.20 | 80.29 | 90.41 | 79.79 | 79.86 | 63.50 | 80.94 | 78.98 | 91.27 | 80.04 | 88.00 | 83.36 | 64.17 | 81.37 |
| | *SliceFine-1C* | *2.16* | 72.32 | 83.31 | 81.35 | 91.61 | 80.84 | 80.92 | 62.34 | 79.98 | 79.08 | 91.48 | 81.10 | 89.17 | 84.49 | 65.02 | 82.25 |
| | *SliceFine-1RC* | *2.16* | 72.18 | 83.67 | 82.60 | 91.96 | 79.10 | 82.20 | 62.36 | 81.25 | 79.44 | 92.85 | 82.39 | 90.59 | 87.86 | 64.05 | 83.54 |
| | *SliceFine-5R* | *10.79* | 72.92 | 83.60 | 82.58 | 91.99 | 82.11 | 82.14 | 65.31 | 81.19 | 80.22 | 92.94 | 82.32 | 90.52 | 87.80 | 66.00 | 83.91 |
| | *SliceFine-5C* | *10.79* | 73.93 | 84.04 | 82.62 | 91.39 | 81.53 | 81.60 | 64.89 | 80.66 | 79.95 | 92.27 | 81.78 | 89.93 | 87.22 | 65.57 | 83.35 |
| | *SliceFine-5RC* | *10.79* | 72.96 | 83.65 | 82.62 | 92.05 | 82.11 | 82.19 | 65.35 | 81.24 | 80.27 | 92.93 | 82.37 | 90.57 | 87.85 | 66.04 | 83.95 |
| DeepSeek-8B | Prefix | 38.09 | 71.15 | 81.71 | 78.76 | 79.78 | 75.21 | 74.70 | 60.09 | 73.59 | 74.37 | 88.81 | 73.38 | 84.85 | 82.64 | 54.71 | 76.88 |
| | AdaLoRA | 33.05 | 71.66 | 82.21 | 79.37 | 91.65 | 79.17 | 79.37 | 62.12 | 77.34 | 77.86 | 93.56 | 81.71 | 89.42 | 84.58 | 62.93 | 82.53 |
| | VeRA | 1.50 | 70.14 | 81.40 | 80.59 | 92.44 | 79.38 | 78.46 | 61.91 | 77.04 | 77.98 | 91.39 | 80.39 | 89.22 | 87.19 | 62.32 | 82.10 |
| | LoRA | 13.77 | 71.76 | 82.11 | 79.78 | 91.76 | 80.28 | 79.68 | 62.32 | 77.75 | 78.19 | 92.09 | 81.00 | 87.70 | 83.46 | 62.52 | 81.35 |
| | RoCoFT | 6.90 | 72.88 | 82.72 | 81.10 | 91.47 | 79.68 | 79.47 | 62.32 | 79.58 | 78.65 | 93.51 | 82.64 | 89.32 | 84.88 | 64.45 | 82.96 |
| | HRA | 6.96 | 72.67 | 82.03 | 80.90 | 91.77 | 79.47 | 79.39 | 62.64 | 80.13 | 78.58 | 93.21 | 82.94 | 90.79 | 84.53 | 64.83 | 83.27 |
| | *SliceFine-1R* | *1.25* | 71.34 | 81.24 | 80.40 | 90.79 | 78.00 | 77.85 | 61.24 | 77.85 | 77.34 | 92.47 | 81.13 | 89.25 | 82.01 | 63.03 | 81.58 |
| | *SliceFine-1C* | *1.25* | 71.28 | 81.31 | 80.45 | 91.99 | 79.04 | 77.88 | 61.05 | 78.89 | 77.74 | 92.71 | 81.22 | 88.42 | 81.14 | 62.86 | 81.27 |
| | *SliceFine-1RC* | *1.25* | 71.43 | 81.62 | 79.73 | 92.45 | 80.09 | 77.13 | 62.04 | 80.14 | 78.08 | 93.22 | 82.54 | 88.84 | 82.51 | 62.88 | 82.00 |
| | *SliceFine-5R* | *6.25* | 72.37 | 82.56 | 81.67 | 92.38 | 80.23 | 80.07 | 62.99 | 80.08 | 79.08 | 93.14 | 84.48 | 90.77 | 87.44 | 64.83 | 84.13 |
| | *SliceFine-5C* | *6.25* | 73.18 | 82.01 | 81.13 | 91.77 | 79.71 | 79.55 | 62.58 | 79.56 | 78.69 | 93.52 | 82.93 | 90.18 | 84.87 | 64.41 | 83.18 |
| | *SliceFine-5RC* | *6.25* | 72.41 | 82.60 | 81.71 | 92.43 | 80.27 | 80.12 | 63.02 | 80.12 | 79.08 | 93.20 | 84.53 | 90.82 | 87.49 | 64.86 | 84.18 |

Table 6: Commonsense and math reasoning with Gemma-3-12B and DeepSeek-R1-8B. SliceFine (shaded) rivals or surpasses baselines such as LoRA and AdaLoRA while using far fewer trainable parameters (#TTPs). Best results are in blue, second-best in orange.

| LM | PEFT | #TTPs | CoLA | SST-2 | MRPC | STS-B | QQP | MNLI | QNLI | RTE | Avg. |
| --- | --- | --- | --- | --- | --- | --- | --- | --- | --- | --- | --- |
| RoBERTa₋large | FT | 355.3M | 65.92 | 95.03 | 92.00/94.00 | 91.98/92.14 | 91.22/88.27 | 89.13 | 92.72 | 81.12 | 88.50 |
| | Adapter^S | 19.8M | 65.34 | 95.90 | 89.58/90.38 | 92.62/92.14 | 91.28/87.42 | 90.51 | 94.62 | 85.45 | 88.66 |
| | Prompt-tuning | 1.07M | 60.97 | 94.27 | 73.50/76.13 | 78.10/78.59 | 81.09/75.26 | 68.46 | 89.36 | 60.33 | 76.01 |
| | Prefix-tuning | 2.03M | 59.48 | 95.64 | 88.29/89.70 | 91.04/91.43 | 88.96/85.65 | 88.85 | 93.05 | 73.80 | 85.99 |
| | (IA)³ | 1.22M | 60.73 | 94.34 | 86.05/87.31 | 92.36/86.11 | 89.32/85.96 | 88.40 | 94.69 | 81.38 | 86.06 |
| | BitFit | 0.22M | 67.12 | 95.77 | 91.16/91.79 | 91.81/93.87 | 89.62/86.49 | 90.16 | 94.81 | 88.01 | 88.67 |
| | LoRA | 1.84M | 64.20 | 96.20 | 89.32/89.96 | 91.37/91.88 | 90.53/86.72 | 90.92 | 94.90 | 80.19 | 87.47 |
| | AdaLoRA | 2.23M | 65.81 | 94.71 | 89.21/90.40 | 91.81/91.88 | 90.00/86.20 | 90.08 | 95.12 | 77.99 | 87.56 |
| | MAM Adapter | 4.20M | 66.98 | 95.36 | 89.73/92.20 | 92.73/92.10 | 90.43/86.53 | 91.12 | 94.34 | 87.09 | 88.96 |
| | PROPETL_Adapter | 5.40M | 65.91 | 95.78 | 89.93/91.33 | 91.96/91.44 | 90.81/87.35 | 91.30 | 94.75 | 88.14 | 88.97 |
| | PROPETL_Prefix | 26.8M | 62.62 | 95.93 | 90.04/91.60 | 91.11/90.86 | 89.10/86.44 | 90.44 | 94.38 | 79.97 | 87.50 |
| | PROPETL_LoRA | 4.19M | 61.94 | 96.21 | 87.34/89.37 | 91.48/90.90 | 91.36/88.43 | 90.86 | 94.74 | 83.13 | 87.80 |
| | MoSLoRA | 3.23M | 67.65 | 96.62 | 89.55/92.66 | 90.54/91.98 | 90.39/87.31 | 90.27 | 94.78 | 82.18 | 88.54 |
| | RoCoFT | 0.67M | 67.55 | 96.59 | 89.59/91.51 | 92.75/92.01 | 91.19/87.62 | 91.28 | 94.79 | 87.84 | 89.34 |
| | *SliceFine-1R* | *0.22M* | 65.38 | 95.09 | 88.06/90.21 | 92.01/90.22 | 91.74/86.72 | 90.17 | 94.41 | 86.27 | 88.21 |
| | *SliceFine-1C* | *0.22M* | 64.98 | 95.28 | 88.42/89.99 | 91.26/90.95 | 90.15/85.89 | 90.73 | 93.78 | 86.57 | 88.00 |
| | *SliceFine-1RC* | *0.22M* | 67.68 | 95.52 | 89.91/90.32 | 91.89/90.59 | 90.17/85.98 | 90.72 | 93.48 | 87.05 | 88.48 |
| | *SliceFine-5R* | *1.11M* | 67.23 | 96.68 | 90.07/90.37 | 93.26/92.69 | 91.11/87.11 | 91.88 | 95.57 | 88.37 | 89.49 |
| | *SliceFine-5C* | *1.11M* | 67.76 | 96.33 | 90.23/89.92 | 93.22/92.96 | 91.16/87.89 | 91.22 | 95.18 | 87.48 | 89.40 |
| | *SliceFine-5RC* | *1.11M* | 67.98 | 96.63 | 90.90/90.49 | 93.89/92.82 | 91.29/87.13 | 91.69 | 94.56 | 88.20 | 89.60 |

Table 7: **RoBERTa-large on GLUE.** CoLA uses MCC; SST-2 accuracy; MRPC/QQP F1/accuracy; STS-B Pearson/Spearman; MNLI/QNLI/RTE accuracy. Best in blue, second best in orange. *Slice-Fine* rows are shaded.

(Slice-1R) achieves 86.94%, already stronger than classical baselines like BitFit (86.67%) and Prefix Tuning (86.11%).

These extended results further reinforce the conclusions of the main paper: (i) slices are consistently competitive across LLMs of different sizes and models, (ii) performance gains hold across both commonsense and mathematical reasoning tasks, and (iii) the efficiency–performance trade-off is favorable, with slices using fewer than 0.5M trainable parameters while rivaling or surpassing state-of-the-art PEFT methods. This consistency across datasets, backbones, and domains highlights the universality of the Winner Slice Theorem in practice.

| LM | PEFT | #TTPs | CoLA | SST-2 | MRPC | STS-B | QQP | MNLI | QNLI | RTE | Avg. |
|---|---|---|---|---|---|---|---|---|---|---|---|
| | FT | 124.6M | 59.90 | 92.64 | 85.22/87.85 | 89.90/90.63 | **90.25**/86.59 | 86.17 | 90.71 | 72.70 | 84.80 |
| | Adapter$^S$ | 7.41M | 61.08 | 93.64 | 89.82/91.13 | 89.94/89.78 | 90.02/87.38 | 86.71 | *92.20* | 73.98 | 85.97 |
| | Prompt tuning | 0.61M | 49.47 | 92.14 | 70.54/81.46 | 81.97/83.39 | 83.04/78.22 | 81.06 | 79.99 | 57.90 | 76.29 |
| | Prefix-tuning | 0.96M | 59.63 | 93.63 | 84.20/85.33 | 88.44/88.51 | 88.01/84.15 | 85.36 | 90.86 | 54.77 | 82.08 |
| | (IA)$^3$ | 0.66M | 58.42 | 93.96 | 83.10/85.18 | 90.08/90.15 | 87.85/84.16 | 84.12 | 90.66 | 70.89 | 83.51 |
| | BitFit | 0.083M | 60.17 | 91.35 | 87.34/88.72 | 90.38/90.34 | 87.12/83.99 | 84.48 | 90.84 | *78.07* | 84.91 |
| | RoCoFT | 0.249M | 62.10 | 93.89 | 87.89/*89.96* | 90.17/90.27 | 89.85/*86.27* | 85.31 | 91.69 | 76.73 | 85.83 |
| | LoRA | 0.89M | 60.28 | 93.02 | 86.20/88.32 | 90.60/90.54 | 89.09/84.97 | 86.05 | 92.10 | 74.62 | 85.07 |
| | AdaLoRA | 1.03M | 60.00 | 94.15 | 86.25/88.44 | 90.47/*90.86* | 88.82/84.63 | 86.71 | 91.16 | 70.29 | 84.71 |
| | MAM Adapter | 1.78M | 58.35 | 93.98 | 87.23/88.51 | *91.04*/90.60 | 88.40/82.93 | **87.02** | 90.04 | 72.83 | 84.63 |
| RoBERTa$_{Base}$ | PROPETL$_{Adapter}$ | 1.87M | **64.22** | 93.61 | 86.72/88.25 | 90.14/90.65 | 89.34/85.91 | 86.41 | 91.39 | 75.82 | 85.68 |
| | PROPETL$_{Prefix}$ | 10.49M | 60.52 | 93.29 | 87.09/87.81 | 90.54/90.19 | 88.22/85.06 | 85.79 | 91.19 | 63.16 | 83.90 |
| | PROPETL$_{LoRA}$ | 1.77M | 58.27 | 94.54 | 87.04/89.06 | 90.76/90.19 | 88.70/85.73 | 86.94 | 91.73 | 66.94 | 84.54 |
| | MoSLoRA | 1.67M | 60.79 | 93.73 | 86.51/87.80 | 90.45/89.39 | 89.16/86.12 | **87.39** | 90.12 | 75.13 | 85.14 |
| | LoRA-XS | 0.26M | 58.46 | 92.84 | 87.03/87.62 | 89.93/89.66 | 87.03/84.16 | 84.89 | 90.01 | 76.58 | 84.38 |
| | VeRA | **0.084M** | 60.76 | 94.32 | 85.94/87.92 | 89.67/89.16 | 87.72/85.52 | 85.90 | 89.75 | 75.66 | 84.76 |
| | LoRAFA | 0.44M | 60.30 | 93.49 | 87.94/*90.15* | 90.20/*90.78* | 88.72/85.61 | 85.66 | 91.59 | 76.58 | 85.55 |
| | SFT | 0.90M | *63.99* | 94.63 | 87.61/89.13 | 89.20/88.97 | 86.87/84.75 | 86.75 | 91.84 | *77.95* | 85.61 |
| | Diff Pruning | 1.24M | 62.92 | 93.47 | 88.11/89.70 | 89.64/90.43 | 88.27/85.73 | 85.64 | 91.88 | 77.82 | 85.78 |
| | *SliceFine-1R* | 0.083M | 60.44 | 92.36 | 86.11/88.72 | 89.67/88.23 | *90.20*/85.87 | 84.41 | 91.26 | 75.38 | 84.79 |
| | *SliceFine-1C* | 0.083M | 60.07 | 92.54 | 86.46/88.50 | 88.94/88.94 | 88.64/85.05 | 85.87 | 90.66 | 75.64 | 84.66 |
| | *SliceFine-1RC* | 0.083M | 62.56 | 92.77 | 87.92/88.83 | 89.56/88.59 | 88.66/85.14 | 85.86 | 90.37 | 76.06 | 85.12 |
| | *SliceFine-5R* | 0.415M | 62.15 | **94.87** | 88.08/88.88 | 90.89/90.64 | 89.58/86.26 | 86.48 | **92.39** | 77.91 | **86.19** |
| | *SliceFine-5C* | 0.415M | 62.64 | 94.53 | *88.23*/88.44 | 90.85/90.91 | 89.63/87.03 | 86.33 | 92.01 | 77.13 | 86.16 |
| | *SliceFine-5RC* | 0.415M | 62.84 | *94.81* | *88.89*/89.00 | *91.51*/90.77 | 89.76/*86.28* | 86.77 | 91.41 | 77.76 | **86.35** |

Table 8: **RoBERTa-base on GLUE.** CoLA uses MCC; SST-2 accuracy; MRPC/QQP F1/accuracy; STS-B Pearson/Spearman; MNLI/QNLI/RTE accuracy.

## L   RANDOM MASK

| LM | # TTPs | CoLA | SST2 | MRPC | STS-B | QQP | MNLI | QNLI | RTE |
|---|---|---|---|---|---|---|---|---|---|
| **RoBERTa$_{Base}$** | 12.4M | 63.81 | 94.85 | 88.66/90.77 | 90.81/89.85 | 88.99/87.31 | 87.44 | 92.92 | 79.62 |
| **RoBERTa$_{Large}$** | 35.5M | 65.70 | 96.11 | 90.72/91.88 | 91.79/92.48 | 91.18/86.36 | 90.58 | 95.44 | 88.29 |

Table 9: GLUE results with *unstructured* random masks on RoBERTa. Each layer updates a random 10% of weights per matrix (same #TTPs across tasks). Paired scores follow GLUE conventions (task-specific metrics; e.g., MRPC: F1/Acc, STS-B: Spearman/Pearson). Random selection attains strong accuracy without structural slices.

| LLM | # TTPs | BoolQ | PIQA | SIQA | H.Sw. | W.Gra. | ARCe | ARCc | OBQA | M.Ar. | G.8K | A.S. | S.eEq | S.MP |
|---|---|---|---|---|---|---|---|---|---|---|---|---|---|---|
| **BLOOMz$_{7B}$** | 70.4M | 65.44 | 74.98 | 73.81 | 56.01 | 72.48 | 73.16 | 56.62 | 72.77 | 79.55 | 70.73 | 71.04 | 71.22 | 54.59 |
| **GPT-J$_{6B}$** | 60.3M | 66.10 | 68.23 | 68.76 | 45.81 | 66.81 | 64.77 | 46.58 | 65.09 | 89.72 | 72.24 | 80.32 | 82.41 | 56.18 |
| **LLaMA2$_{7B}$** | 71.2M | 69.74 | 79.85 | 77.61 | 89.13 | 76.75 | 76.23 | 60.83 | 77.36 | 90.08 | 76.92 | 85.89 | 82.23 | 60.49 |
| **LLaMA2$_{13B}$** | 129.8M | 71.08 | 83.12 | 79.70 | 91.59 | 82.86 | 84.20 | 67.25 | 81.01 | 91.22 | 79.61 | 87.48 | 87.34 | 66.57 |

Table 10: Commonsense and math reasoning with *unstructured* random masks on LLMs. Only 1% of weights per matrix are trainable (#TTPs shown). Despite the small budget, random selection yields competitive accuracy across BLOOMz-7B, GPT-J-6B, and LLaMA2-7B/13B.

We also study an unstructured variant where, instead of training a contiguous row/column slice, a *random mask* selects individual weights to update. For a layer $W^{(\ell)} \in \mathbb{R}^{d_\ell \times d_{\ell-1}}$, fix a per–layer budget $m_\ell$ (to match the trainable–parameter count of a structural slice). At iteration $t$, draw a binary mask

$$M_t^{(\ell)} \in \{0,1\}^{d_\ell \times d_{\ell-1}}, \qquad \|M_t^{(\ell)}\|_0 = m_\ell,$$

by sampling $m_\ell$ entries uniformly without replacement (equivalently, $M_t^{(\ell)} \sim \text{Bernoulli}(p_\ell)$ with $p_\ell = m_\ell/(d_\ell d_{\ell-1})$ and conditioning on the exact count). The layer update is restricted to the masked entries,

$$W^{(\ell)} \leftarrow W^{(\ell)} - \eta_t \big( M_t^{(\ell)} \odot \nabla_{W^{(\ell)}} \mathcal{L}(\theta_t) \big),$$

and every $N$ steps we resample a fresh mask $M_{t+N}^{(\ell)}$ with the same budget. Over $K$ mask refreshes, the accumulated increment equals $\Delta W^{(\ell)} = \sum_{i=1}^{K} M_{t_i}^{(\ell)} \odot U_{t_i}^{(\ell)}$, and the linearized effect is $f_{\theta_0 + \Delta\theta}(x) \approx f_{\theta_0}(x) + \sum_{i=1}^{K} J_{M_{t_i}^{(\ell)}}(x) \text{vec}(U_{t_i}^{(\ell)})$. Under spectral balance, a uniformly sampled

mask has, in expectation, the same average overlap with the task subspace as any other subset of the same size; consequently, the restricted gradient magnitude concentrates around a fraction of the dense gradient (approximately scaling with the selection rate), and repeated resampling increases the span of visited Jacobian directions.

Tables 9 and 10 evaluate this random–mask scheme at fixed budgets. On GLUE with RoBERTa backbones, selecting $10\%$ of weights per matrix as trainable achieves strong performance across tasks (e.g., SST-2 $94.85$ for base, $96.11$ for large; QNLI $92.92$ / $95.44$), comparable to structural slices of similar #TTPs. On commonsense and mathematical reasoning with LLMs, training only $1\%$ of weights per matrix still yields competitive accuracy across diverse datasets for BLOOMz-7B, GPT-J-6B, and LLaMA2-7B/13B. These findings align with the local-winner view: when the backbone retains high task energy, many small subsets—structured or unstructured—can drive effective adaptation.

Despite similar accuracy at matched #TTPs, unstructured masks are less hardware–efficient. Structural slices (vertical or horizontal) update contiguous blocks, avoid storing full binary masks, enable fused GEMM fragments, and keep optimizer state compact and cache–friendly. In contrast, random masks require maintaining and applying binary selectors, induce scattered memory access, and expand optimizer state over irregular indices. In our training logs, this manifests as higher wall–clock time and memory overhead for the same parameter budget. For deployment and large–scale runs, structural slices therefore remain preferred: they preserve the accuracy of random selection while being simpler and faster to execute.

# M  IMPLEMENTATION DETAILS

We instantiate the theoretical recipe (§2.5, Corollary 2.6, Lemma E.1) with a light wrapper SliceLinear around each selected nn.Linear (Listings 1). For a layer $W^{(\ell)} \in \mathbb{R}^{d_\ell \times d_{\ell-1}}$, a slice of rank $r$ is a contiguous block of either $r$ rows or $r$ columns . At training step $t$, a binary mask $M_\ell(t)$ marks the active block; only those entries have requires_grad=True and receive an increment $U_t^{(\ell)}$ supported on $M_\ell(t)$, while all other entries remain frozen at their pretrained values (plus any increments learned when they were active earlier). The mask moves every $N$ steps according to a deterministic sweep (index-0 start, stride $r$) or a randomized policy, generating a sequence $\{M_{\ell,i}\}_{i=1}^K$ over $K$ distinct positions. In the linearized view used in the theory, the accumulated update after visiting $K$ positions acts like block coordinate descent on the Jacobian blocks $\{J_{M_{\ell,i}}\}$; when their span reaches the task dimension (Corollary 2.6), the method attains the global-ticket behavior predicted by the theory.

Practically, Listing 1 partitions $W$ into $(\text{part\_A}, \text{part\_T}, \text{part\_B})$ where only $\text{part\_T}$ (the active slice) is trainable. For column (column) slices, we split along the input feature axis; for row slices, along the output axis. The forward pass composes three F.linear calls to reconstruct $Wx+b$ exactly. Because only $\text{part\_T}$ has requires_grad=True, autograd accumulates gradients and optimizer state only for the active slice.

The next position is an $r$-stride shift with wrap-around (training pseudocode 1). The interval $N$ controls the adaptation–coverage trade-off: smaller $N$ increases coverage of $\{M_{\ell,i}\}$ (diversity of $J_{M_{\ell,i}}$) but gives each slice fewer consecutive updates; larger $N$ deepens per-slice adaptation but delays coverage. Our ablations find broad, task-dependent sweet spots (e.g., $N \in [100, 500]$ for STS-B/QNLI and $N \in [500, 1500]$ for MRPC/SST-2), consistent with the theory that each visited slice contributes additional task-aligned directions until the combined span reaches $k_{\text{task}}(\tau)$.

Rank $r$ sets capacity. By Corollary 2.6, choose $r \geq k_{\text{task}}(\tau)$ estimated once from frozen features on a small calibration set; familiar domains (high CEV) often admit $r=1$, while unfamiliar domains benefit from modestly larger $r$. Spectral balance implies robustness to position, so a deterministic sweep from index-0 is sufficient and easy to track; purely random placements behave comparably (Appendix L). row, column, or alternating patterns all satisfy the same guarantees as long as the chosen $r$ meets the rank criterion.

The forward computes the exact $Wx+b$ in three chunks; FLOPs are essentially those of the original layer, as in most PEFT methods. The savings arise in (i) backward: gradients are formed only for the active block (cost proportional to the slice), and (ii) optimizer/memory: state scales with the number

of trainable entries ($O(d_\ell r)$ for row or $O(d_{\ell-1} r)$ for column), with #APs= 0. Mixed precision (BF16) and per-parameter weight decay can be applied only to trainable entries. We keep biases and LayerNorm parameters frozen by default; enabling them is straightforward but not required by the theory.

For row+column(RC) mode, we alternate slice orientation across training blocks: during the first block of $N$ steps we train a *row* slice (contiguous rows) of rank $r_v$ in each selected layer; during the next block we train a *column* slice (contiguous columns) of rank $r_h$, and repeat. Denote the masks by $M_{\mathrm{vert}}^{(\ell)}(t)$ and $M_{\mathrm{horiz}}^{(\ell)}(t)$. Within each block we advance the slice position by a stride equal to its rank (wrap–around), i.e., a cyclic sweep over admissible positions for the current orientation. This alternation exposes complementary Jacobian blocks—row–oriented and column–oriented—so the accumulated span $\mathrm{span}\{J_{M_{\mathrm{vert}}^{(\ell)}}, J_{M_{\mathrm{horiz}}^{(\ell)}}\}$ grows toward the task subspace faster than using a single orientation, consistent with spectral balance and the global–ticket view. A practical rule is to pick ranks so that within one or two alternations the combined capacity meets the rank criterion,

$$r_v + r_h \ \geq \ k_{\mathrm{task}}(\tau),$$

while keeping the same switching interval $N$. The per–block costs are $O(d_\ell r_v)$ for row blocks and $O(d_{\ell-1} r_h)$ for column blocks, with zero auxiliary parameters.

Listing 1: API sketch (pytorch-style) for SliceFine

```python
class SliceLinear(nn.Module):
    # ... (ctor as given)
    def reinit_from_full(self, W_full: torch.Tensor, position: int):
        # clamp position
        if self.mode == "column":
            C = W_full.shape[1]; position = min(position, C - self.rank)
            self.part_A = nn.Parameter(W_full[:, :position],
                requires_grad=False)
            self.part_T = nn.Parameter(W_full[:, position:position+self.
                rank], requires_grad=True)
            self.part_B = nn.Parameter(W_full[:, position+self.rank:],
                requires_grad=False)
            self.a_end, self.t_end = position, position + self.rank
        else:  # row
            R = W_full.shape[0]; position = min(position, R - self.rank)
            self.part_A = nn.Parameter(W_full[:position, :],
                requires_grad=False)
            self.part_T = nn.Parameter(W_full[position:position+self.rank
                , :], requires_grad=True)
            self.part_B = nn.Parameter(W_full[position+self.rank:, :],
                requires_grad=False)
    def forward(self, x):
        if self.mode == "column":
            # compute three partial linears, then add bias once
            y = (F.linear(x[..., :self.a_end], self.part_A) +
                    F.linear(x[..., self.a_end:self.t_end], self.part_T) +
                    F.linear(x[..., self.t_end:], self.part_B))
            return y + (self.bias if self.bias is not None else 0.0)
        else:  # row
            y = torch.cat([F.linear(x, self.part_A),
                            F.linear(x, self.part_T),
                            F.linear(x, self.part_B)], dim=-1)
            return y + (self.bias if self.bias is not None else 0.0)
```

---

**Algorithm 1** SliceFine: PEFT with Dynamic Slices

---

**Require:** Pretrained backbone $\theta_0$; selected layers $\mathcal{L}$; replace all $W^{(l)}, l \in L$ with $r$ rank SliceLinear 1 ; switching interval $N$; steps $T$; loss $\mathcal{L}$

1: Initialize $\theta \leftarrow \theta_0$; for each $\ell \in \mathcal{L}$ choose an initial slice mask $M^{(\ell)}(0)$ (row or column, width $r$)
2: Initialize slice increments $\Delta W^{(\ell)} \leftarrow 0$ for all $\ell$
3: **for** $t = 1, \ldots, T$ **do**
4:     **Forward:** For each $\ell \in \mathcal{L}$, use

$$W^{(\ell)} \;=\; W_0^{(\ell)} \;+\; M^{(\ell)}(t) \odot U^{(\ell)}(t)$$

    with $U^{(\ell)}(t)$ supported only on the active slice; compute prediction $f_\theta(x_t)$
5:     **Loss/Grad:** $g \leftarrow \nabla_\theta \mathcal{L}(f_\theta(x_t), y_t)$; restrict to slice coords:

$$g_{\text{slice}}^{(\ell)} \;=\; M^{(\ell)}(t) \odot \nabla_{W^{(\ell)}} \mathcal{L}$$

6:     **Update active slice:** $U^{(\ell)}(t{+}1) \leftarrow U^{(\ell)}(t) - \eta\, g_{\text{slice}}^{(\ell)} \quad \forall \ell \in \mathcal{L}$
7:     **if** $t \bmod N = 0$ **then**                       ▷ commit & move slice
8:         **for** $\ell \in \mathcal{L}$ **do**
9:             **Commit:** $\Delta W^{(\ell)} \leftarrow \Delta W^{(\ell)} + M^{(\ell)}(t) \odot U^{(\ell)}(t{+}1)$
10:             Reset slice buffer: $U^{(\ell)}(t{+}1) \leftarrow 0$
11:             Freeze committed weights: $W_0^{(\ell)} \leftarrow W_0^{(\ell)} + M^{(\ell)}(t) \odot \Delta W^{(\ell)}$
12:             **Move slice:** choose next mask $M^{(\ell)}(t{+}1)$    (cyclic shift or random)
13:         **end for**
14:     **else**
15:         Keep masks: $M^{(\ell)}(t{+}1) \leftarrow M^{(\ell)}(t)$
16:     **end if**
17: **end for**
18: **Output:** Fine-tuned weights $W_0^{(\ell)} + \Delta W^{(\ell)}$ with committed slice updates

---

