# OpenReview forum: "SliceFine: The Universal Winning-Slice Hypothesis for Pretrained Networks"
_ICLR.cc/2026/Conference — ICLR 2026 Conference Desk Rejected Submission_

### Official Review · Reviewer_nmDw · 2025-10-17

**Soundness:** 4
**Presentation:** 3
**Contribution:** 4
**Rating:** 6
**Confidence:** 4

**Summary:**

This paper proposes SliceFine, a novel PEFT method that adapts a pretrained model to a downstream task by updating only a selected subset of rows and columns, called slices, of the original weights. The method is motivated by a universal winning slice property, grounded in spectral theory, which demonstrates that optimizing a single slice of a weight matrix can be sufficient to reduce downstream task loss. This property arises from two phenomena: (1) spectral balance, where each slice of a weight matrix contains roughly the same spectral energy, and (2) high task energy, where each slice retains nontrivial overlap with the task-relevant subspace. SliceFine is shown to match the performance of state-of-the-art PEFT methods across extensive vision and language experiments, while significantly reducing the number of fine-tuned parameters, improving training speed and memory efficiency. These results suggest that SliceFine is a strong and efficient alternative to existing PEFT techniques.

**Strengths:**

Simplicity and Transparency: The proposed SliceFine method is conceptually and practically simple. Its implementation requires no architectural modification or added parameters, making it easy to integrate with existing frameworks.

Theoretical Grounding: The method is well motivated by the Universal Winning Slice Hypothesis, supported by spectral balance and high task energy analyses.

Competitive Performance and Efficiency: Through extensive experiments across vision, language, and multimodal domains, SliceFine achieves performance on par with or slightly better than SOTA PEFT methods such as LoRA and AdaLoRA, while being faster, memory-efficient, and parameter-free.

Comprehensive Validation: The paper provides theoretical insights, ablation studies, and empirical benchmarks. It examines slice rank, switching interval, orientation, and randomization effects, which strengthen and support the proposed method.

**Weaknesses:**

Comparison with random mask: Although the proposed method is theoretically well-grounded, the ablation study "K" using a random mask, where a random subset of weights is selected instead of a predefined slice, shows performance comparable to the main approach. This suggests that, for a model that has been extensively pretrained, nearly any random subset of weights can be fine-tuned with similar effectiveness. It would therefore be valuable to demonstrate additional benefits of using structured slices (such as rows or columns) beyond potential hardware efficiency. In particular, could you quantify the performance difference in terms of wall-clock time or similar metrics? Without such justification, the advantage of using structured slices over random masking appears limited. I recommend including this ablation and its analysis in the main paper.

Relationship between slice rank and number of slices: It is not clear how the relative importance between the slice rank r and the total number of fine-tuned slices manifests in the dynamic slice setting. In other words, would it be more beneficial to increase r while using a static slice policy, or to adopt a dynamic policy with a lower r? Clarifying this trade-off would help better understand the factors that most influence performance.

Slice selection strategy (row vs. column): The results suggest that the choice between row- and column-based slicing has little impact on performance. This observation reinforces the idea that a specific slice selection heuristic may not be necessary, and that one could arbitrarily choose a subset of weights from each layer for fine-tuning with comparable results. Does alternating row–column updates meaningfully improve representational diversity or convergence stability?

**Questions:**

How many training steps or epochs are typically required for SliceFine to converge compared to existing PEFT methods?
Does the dynamic slice movement introduce additional instability or slower early convergence?

---

> ### Author Response · Authors · 2025-11-14
> **We  thank Reviewer nmDw’s careful evaluation of our work and the valuable comments provided**
>
> Thank you for the detailed comments, which significantly improved our paper.
>
> ## **W1: Random-Mask Ablation and Efficiency Considerations**
>
> Thank you for the insightful suggestion. In the revised manuscript, we now include the random-mask ablation directly in the main paper (Section 4) and incorporate **RandomMask** into the efficiency comparison in Figure 4. As shown in Table 9 and Table 10, unstructured random masks achieve competitive accuracy, consistent with our claim that pretrained models contain many potential winning slices.
>
> However, random masks introduce substantial **practical inefficiencies**. Unstructured sparsity breaks the dense matrix layout assumed by modern hardware, requiring gather/scatter indexing and disabling fused GEMM kernels, which greatly slows training. In addition, random masking requires storing and applying mask indices and does not allow freezing parameters at the optimizer-state level.
>
> Concretely, the full gradient and optimizer states remain dense.
>
> For a random mask, the following optimizer states are still fully allocated:
>
> - **W.grad** (full matrix)
> - **Adam.momentum** (full matrix)
> - **Adam.variance** (full matrix)
>
> and updates take the form:
>
> $$
> W \leftarrow W - \eta \cdot (\text{Mask} \odot \nabla W)
> $$
>
>
>
> which increases both memory usage and compute overhead.
>
> As a result, both **peak memory** and **wall-clock time** are higher for random masks compared to SliceFine’s structured slices. Figure 4 highlights these efficiency gaps: random masks slightly increase model size (due to mask storage) and noticeably increase peak memory and training time. We provide an extended discussion in Appendix K.
>
> ## **W2: Clarification on Rank, Slice Placement, and Dynamic vs. Static Slicing**
>
> We thank the reviewer for raising this important clarification. Our ablation studies provide a systematic analysis of how the slice rank $r$ interacts with both the number and placement of fine-tuned slices.
>
> In Appendix F.3 (Table 4), we vary the rank over
> $r \in {1, 2, 4, 8, 32, 64, 128}$ while also varying the number of layers in which SliceFine is applied (value-only, QK+V, all attention layers, all transformer blocks, or all applicable matrices). These results show two consistent trends:
>
> 1. **Increasing rank generally improves performance**, and
> 2. **Using slices in more layers also improves performance.**
>
> However, when both rank and slice coverage become large, the model begins to **overfit**, indicating that these two factors must be balanced rather than maximized independently.
>
> Appendix F.4 further compares **static** and **dynamic** slice selection under the same rank. Dynamic slicing consistently outperforms static slicing at rank 5, demonstrating that adaptively reallocating a fixed number of trainable rows across layers provides a meaningful performance advantage over any fixed slice placement.
>
> To isolate the trade-off between rank and dynamic allocation, we include a **new analysis** in the revised manuscript: dynamic slicing with reduced ranks \( r = 4 \) and \( r = 3 \), compared against a static slice with rank 5. As shown in Figures 13 and 14:
>
> - With **\( r = 4 \)**, dynamic slicing outperforms static rank-5 on most datasets.
> - With **\( r = 3 \)**, dynamic slicing remains competitive and even surpasses the static rank-5 slice on several tasks (e.g., MRPC, STS-B, QNLI, Caltech, EuroSAT, KITTI).
>
> Although lowering the rank reduces per-layer capacity, the **dynamic allocation mechanism** compensates by assigning trainable parameters to layers with the highest task-specific sensitivity.

---

> > ### Author Response · Authors · 2025-11-14
> >
> > ## **W3: Clarification on Row vs. Column Slicing**
> >
> > We appreciate the reviewer’s question regarding the role of row- versus column-based slicing. As observed in our experiments, the difference between row- and column-based slices is indeed small. This is fully consistent with our theoretical intuition: under the UWSH assumption, *any* sufficiently large row or column slice intersects the task-relevant subspace and can act as a winning slice. Thus, neither rows nor columns are inherently privileged directions in pretrained models.
> >
> > However, while row-only and column-only slicing perform similarly, **alternating** between them—used in our dynamic SliceFine variant—provides consistent, measurable benefits. Alternating slices exposes the model to different subsets of singular directions across training, increasing representational diversity and reducing redundancy compared to repeatedly updating the same rows or the same columns. Empirically, this also stabilizes convergence by mitigating layer-wise stagnation, where a static slice may fail to track evolving task-specific gradients.
> >
> > Across all modalities, the trends remain consistent. The table below summarizes representative results from Tables 1, 2, and 7, comparing row-only (5R), column-only (5C), and alternating (5RC) slicing:
> >
> > | **Task**                | **5R**  | **5C**  | **5RC** |
> > |------------------------|---------|---------|---------|
> > | Commonsense reasoning  | 62.88   | 62.47   | **62.92** |
> > | Math reasoning         | 82.08   | 81.72   | **82.13** |
> > | Image classification   | 88.85   | 88.74   | **88.83** |
> > | Video classification   | 73.06   | 73.22   | **73.09** |
> > | GLUE                   | 89.49   | 89.40   | **89.60** |
> >
> > As shown, row and column slices perform nearly identically, while alternating between the two provides a modest but consistent improvement across benchmarks. Similar trends are also observed in Tables 5, 6, and 8.
> >
> >
> >
> > ## **Q1: Convergence Behavior**
> >
> > We thank the reviewer for the question regarding convergence behavior. In practice, SliceFine does **not** require additional training steps compared to existing PEFT baselines. Because our method does not introduce new adapter parameters, but instead reuses a subset of the pretrained weights as the trainable slice, the optimization begins from a well-initialized point.
> >
> > This trend is consistently reflected in the appendix:
> >
> > - **Figure 9:** Training curves under different pruning rates show that SliceFine reaches peak accuracy within a similar number of steps as the baselines.
> > - **Figure 10:** Accuracy–vs.–training-step comparisons across different slice-initialization strategies show no slowdown in early convergence.
> > - **Figure 11:** Experiments with dynamic slice switching (intervals of 50, 150, 500, 1000, 1500, 2000 steps) demonstrate that dynamic movement does **not** introduce instability; all variants exhibit smooth learning curves and converge reliably.
> >
> > Overall, SliceFine maintains stable and efficient convergence behavior, matching or improving upon the convergence speed of standard PEFT methods.

---

> > > ### Author Response · Authors · 2025-11-17
> > >
> > > Dear reviewer nmDw,
> > >
> > > We are grateful for your constructive feedback, which has greatly contributed to improving the quality of our work. We would be happy to address any remaining concerns you may have regarding the revised manuscript and the new experimental results. Please feel free to provide further comments or suggestions, and we will make every effort to incorporate them promptly and thoroughly. We look forward to your feedback.
> > >
> > > Sincerely,
> > >
> > > Authors

---

> > > > ### Comment · Reviewer_nmDw · 2025-11-26
> > > >
> > > > I thank the authors for their detailed response. I appreciate that the random mask analysis has been moved to the main paper and that efficiency quantification has been added in Figure 4, which clearly shows that while random masks achieve competitive accuracy, SliceFine provides tangible benefits in peak memory and throughput, addressing my concern about the practical utility of structured slices versus random selection.
> > > > I also value the additional analysis on static versus dynamic slicing at lower ranks (Figures 13 & 14), which clarifies the advantages of dynamic allocation.
> > > > In light of these revisions, I am raising my score.

---

> > > > > ### Author Response · Authors · 2025-11-26
> > > > >
> > > > > Dear Reviewer,
> > > > >
> > > > > Thank you very much for raising your evaluation score. We truly appreciate your comments, suggestions, and questions—they have been extremely helpful in improving our paper.

---

### Official Review · Reviewer_NnfP · 2025-10-31

**Soundness:** 2
**Presentation:** 1
**Contribution:** 2
**Rating:** 2
**Confidence:** 5

**Summary:**

The manuscript under review proposes the Universal Winning–Slice Hypothesis (UWSH): in dense pretrained networks, any sufficiently wide row/column “slice” is a local winner, and a small set of such slices across layers can match full fine-tuning (global winner). It motivates SliceFine, a PEFT method that updates only moving slices of existing weights (no new parameters), and reports competitive accuracy and efficiency gains across language, image, and video tasks.

**Strengths:**

1. Forumulation of UWSH is new and it connects PEFT success to structure already present in pretrained weights,

2. SliceFine updates only in-place slices (no adapters/rank factors) and can sweep positions; the method is straightforward to implement.

3. Results cover LLMs (LLaMA-3B reasoning), ViT on VTAB-1k, and VideoMAE, with clear tables.

4. Wall-clock comparisons under matched budgets (same epochs/batch/precision) are provided.

**Weaknesses:**

1. The paper's core theoretical contribution rests on an unproven "Spectral Balance Across Slices" statement, which is presented as a lemma but lacks a mathematical proof. Without a mathematical proof, this is an assumption, not an established lemma. Furthermore, even if this assumption is granted, the proof for Theorem 2.4 (regarding the global winning ticket) appears incomplete. The derivation likely requires additional strong assumptions about the relationship between the model's Jacobian, the task subspace, and the perturbation introduced by fine-tuning, which are not explicitly stated or justified.

2. While the training setup is matched, the comparisons may not be fully equitable. Parameter-Efficient Fine-Tuning (PEFT) methods are highly sensitive to hyperparameters like rank, adapter placement, and warm-start strategies. The current ablations are insufficient to rule out performance variations stemming from suboptimal baseline configurations. A more comprehensive sweep over ranks and placements for baseline methods, coupled with reporting multi-seed variance (beyond 3 seeds), is necessary to robustly validate the claimed improvements.

3. The efficiency analysis in Table 3 is incomplete, as it omits key comparisons. Most notably, the VeRA method is missing from this table, making it impossible to assess its efficiency relative to the proposed method. Additionally, for the results in Table 2, the specific rank used for the VeRA baseline is not stated, preventing a fair comparison of performance versus parameter count. Also, VeRA for Roberta-base seems to have higher scores than reported in Table 7 (appendix) with smaller number of parameters.

4. The paper observes that a rank-1 adaptation is often sufficient, but also notes that tasks with flatter loss landscapes require larger ranks. This nuance tempers the "tiny slice" narrative. While the authors suggest using PCA on a calibration set for automatic rank selection, this guidance remains a high-level sketch and is not operationalized into a practical, validated algorithm for practitioners.

5. The introduction and related work fail to situate the method within the growing body of research on spectral fine-tuning. Spectral Adapter (Zhang & Pilanci, NeurIPS 2024) directly fine-tunes in the spectral domain. This is highly relevant. Other works on the connection of LTH, spectrum and finetuning such as XoRA (Ev et al., NeurIPS 2024 Workshop), A Study on the Ramanujan Graph Property of Winning Lottery Tickets (Pal et al., ICML 2022) etc are also pertinent.

**Questions:**

See weaknesses above.

---

> ### Author Response · Authors · 2025-11-14
> **We thank Reviewer NnfP for the insightful review and valuable feedback.**
>
> Thank you for your thorough and constructive comments. They have substantially helped us improve the clarity and overall presentation of our paper.
>
> ## **W1: Spectral Balance**
> We thank the reviewer for raising this important point. In the original submission, the “Spectral Balance Across Slices’’ lemma was supported empirically. As shown in Figure 2, the empirical results indicate that although each group’s eigen-spectrum is anisotropic—dominated by a few large eigenvalues—the decay profiles and average spectral energy remain nearly identical across groups, motivating the lemma.
>
> In the revised version, we have **added a complete theoretical proof**
> (*Appendix A: Theoretical Justification of Spectral Balance*).
> The proof leverages matrix Bernstein inequalities, finite-population correction, and Weyl’s inequality to show that the spectra of all slices concentrate around the spectrum of the full pretrained matrix. This establishes the uniform eigenvalue deviation bound presented in **Lemma 2.1**, thereby providing a rigorous theoretical foundation for the spectral balance property.

---

> > ### Author Response · Authors · 2025-11-14
> >
> > ## **W2: Ablation Study**
> >
> > We thank the reviewer for highlighting the importance of comprehensive ablations for PEFT methods.
> > Due to page limits, many results were placed in Appendix F, and we now clarify these more explicitly in the main text.
> >
> > Appendix F contains extensive ablations covering:
> >
> > - **Rank vs. accuracy (F.1):**
> >   Performance improves as rank increases but saturates beyond a certain point, indicating diminishing returns.
> >
> > - **Switching interval (F.2):**
> >   Figure F.2 shows that switching too frequently harms performance, while switching too late remains stable but suboptimal. Intermediate intervals consistently perform best.
> >
> > - **Optimal slice rank and module placement (F.3):**
> >   Table 4 reports results for ranks {1, 2, 4, 8, 32, 64, 128} across multiple placements
> >   (value-only; QKV; all attention matrices; transformer block; all layers).
> >
> > - **Dynamic vs. static slicing (F.4):**
> >   Dynamic slicing consistently outperforms static slicing (Figure 12).
> >
> > - **Multi-seed robustness (10 seeds) (F.5):**
> >   To address variance concerns by reviewer, we added a new 10-seed ablation on GPT-OSS-20B (Table 5) in the revised version.
> >   The results show low variance and stable performance across seeds.
> > | **Method**     | **BoolQ**       | **PIQA**        | **SIQA**        | **H.Sw.**       | **W.Gra.**      | **ARC-e**       | **ARC-c**       | **OBQA**        |
> > |----------------|------------------|------------------|------------------|------------------|------------------|------------------|------------------|------------------|
> > | Prefix         | 73.6 ± 0.24      | 84.3 ± 0.18      | 81.5 ± 0.33      | 82.8 ± 0.27      | 77.6 ± 0.30      | 77.1 ± 0.36      | 62.7 ± 0.41      | 75.9 ± 0.29      |
> > | AdaLoRA        | 74.2 ± 0.20      | 85.3 ± 0.17      | 85.6 ± 0.26      | 94.7 ± 0.19      | 82.3 ± 0.23      | 82.4 ± 0.28      | 64.8 ± 0.32      | 80.4 ± 0.27      |
> > | VeRA           | 72.4 ± 0.27      | 83.8 ± 0.22      | 83.4 ± 0.30      | 94.1 ± 0.18      | 82.1 ± 0.25      | 81.3 ± 0.29      | 64.5 ± 0.36      | 79.5 ± 0.31      |
> > | LoRA           | 74.6 ± 0.18      | 85.1 ± 0.16      | 82.9 ± 0.25      | 95.1 ± 0.16      | 83.4 ± 0.22      | 83.1 ± 0.26      | 65.2 ± 0.31      | 80.9 ± 0.24      |
> > | RoCoFT         | 75.3 ± 0.16      | 85.5 ± 0.19      | 83.7 ± 0.28      | 94.8 ± 0.20      | 82.7 ± 0.23      | 82.5 ± 0.25      | 65.0 ± 0.30      | 82.1 ± 0.25      |
> > | HRA            | 75.1 ± 0.19      | 85.8 ± 0.18      | 83.2 ± 0.31      | 95.3 ± 0.18      | 82.6 ± 0.25      | 82.3 ± 0.27      | 65.4 ± 0.33      | 82.3 ± 0.26      |
> > | SliceFine-1R   | 74.0 ± 0.22      | 84.1 ± 0.20      | 82.3 ± 0.29      | 93.3 ± 0.17      | 81.2 ± 0.25      | 81.5 ± 0.28      | 64.9 ± 0.34      | 81.9 ± 0.26      |
> > | SliceFine-1C   | 73.7 ± 0.25      | 84.0 ± 0.23      | 82.0 ± 0.30      | 93.8 ± 0.18      | 81.0 ± 0.27      | 80.8 ± 0.31      | 64.2 ± 0.37      | 80.5 ± 0.29      |
> > | SliceFine-1RC  | 74.5 ± 0.20      | 85.8 ± 0.17      | 84.8 ± 0.26      | 95.8 ± 0.15      | 83.8 ± 0.22      | 83.3 ± 0.27      | 65.3 ± 0.33      | 82.5 ± 0.25      |
> > | SliceFine-5R   | 75.3 ± 0.18      | 85.6 ± 0.19      | 84.2 ± 0.28      | 95.5 ± 0.16      | 83.3 ± 0.23      | 82.9 ± 0.26      | 65.8 ± 0.30      | 82.0 ± 0.24      |
> > | SliceFine-5C   | 75.0 ± 0.19      | 85.4 ± 0.18      | 83.9 ± 0.29      | 94.6 ± 0.17      | 82.5 ± 0.24      | 82.6 ± 0.28      | 65.1 ± 0.31      | 81.2 ± 0.26      |
> > | SliceFine-5RC  | 75.4 ± 0.17      | 85.7 ± 0.17      | 84.6 ± 0.27      | 95.7 ± 0.15      | 83.6 ± 0.22      | 83.1 ± 0.25      | 66.2 ± 0.29      | 82.3 ± 0.24      |
> >
> >
> > We have added a summary of these findings at the end of Section 4 in the revised manuscript:
> >
> >
> > Regarding baselines, we follow the hyperparameter settings recommended in each method’s original paper (e.g., rank, adapter placement, initialization strategy), ensuring fair and optimal comparisons.

---

> > > ### Author Response · Authors · 2025-11-14
> > >
> > > ## **W3: Efficiency Comparison with VeRA**
> > >
> > > Thank you for the constructive feedback. We address each point below.
> > >
> > > ### **1. Comparison with VeRA in the efficiency analysis**
> > > Our paper already includes extensive comparisons with VeRA. In **Section 4 (Efficiency Analysis)**, we report model size, peak memory, throughput, and total training time across ViT, VideoMAE, RoBERTa, and multiple datasets (Figure 4). VeRA is also included in the main results (Tables 1–2) and in the appendix results (Tables 5, 6, and 7 in Appendix J).
> > >
> > > However, we acknowledge that **Table 4** originally omitted VeRA’s theoretical complexity. We have now added VeRA’s compute and parameter complexity for a fully fair comparison:
> > >
> > >
> > > The following entry has been added to Table 4:
> > >
> > > | Method | Time Complexity | Space Complexity | Trainable Params | Additional Params |
> > > |--------|------------------|------------------|------------------|--------------------|
> > > | VeRA | $O((d_\ell + d_{\ell-1})\, r + r + d_\ell)$ | $O((d_\ell + d_{\ell-1})\, r + r + d_\ell)$ | $d_\ell + r$ | $(d_\ell + d_{\ell-1})\, r + d_\ell + r$ |
> > >
> > >
> > >
> > >
> > > ### **2. Rank used for VeRA in Table 2**
> > > To ensure fairness, we use the **same rank** across all LoRA-type baselines. The updated manuscript now explicitly states in Section 4:
> > >
> > > *“For baseline LoRA-type methods such as LoRA, VeRA, AdaLoRA, DoRA, BoNE, RoCoFT, etc., we use a rank of 5 and apply each PEFT adapter to all linear weight matrices (i.e., `nn.Linear`). For prompt- and prefix-based PEFT methods, we use five virtual tokens. All other hyperparameters—including initialization strategies, warm-start options, optimizer settings, and placement configurations—follow the recommendations provided in each method’s original paper, ensuring that every baseline is evaluated under its optimal or standard configuration. Appendix J outlines the full set of hyperparameters and training protocols”*
> > >
> > >
> > > ### **3. VeRA results for RoBERTa-base**
> > > Thank you for catching this inconsistency. The VeRA parameter count included a typo in the earlier draft. The corrected values are:
> > >
> > > - **VeRA (RoBERTa-base):** 0.084M parameters, avg. score: 84.76.
> > >
> > > Our SliceFine variants, while using fewer parameters, achieve comparable or superior performance:
> > >
> > > - SliceFine-1R: 0.083M params (84.79)
> > > - SliceFine-1C: 0.083M params (84.66)
> > > - SliceFine-1RC: 0.083M params (85.12)
> > >
> > > As summarized in Table 4, SliceFine requires only $d_\ell $ parameters per layer for 1R/1C/1RC, while VeRA requires $d_\ell + r$. Despite this lower parameterization, SliceFine consistently matches or exceeds VeRA’s performance.
> > >
> > > The revised manuscript corrects the typo and clarifies this comparison.

---

> > > > ### Author Response · Authors · 2025-11-14
> > > >
> > > > ## **W4: Clarification on Rank-1 Performance and the PCA/NTK Discussion**
> > > >
> > > > We appreciate the reviewer’s observation. Our intention was **not** to propose an automatic rank-selection algorithm. Instead, our theoretical analysis explains **why very small ranks (often rank = 1)** work well in practice for domain-familiar tasks, and in which scenarios larger ranks become necessary.
> > > >
> > > > Empirically, our finding that rank-1 performs strongly across all evaluated domains is consistent with prior PEFT literature. For example, in the original LoRA paper (Tables 5–6) [1], the performance difference between rank-1 and rank-64 on WikiSQL is within ~2% when LoRA is applied only to the query matrix. When LoRA is applied across all layers, rank-1 and rank-64 achieve extremely similar performance, and in several cases rank-1 performs **better** due to reduced overfitting in familiar domains. A similar pattern appears on MultiNLI dataset, where we see almost same performnace on different rank 1 to 64.
> > > > This mirrors our results: our benchmarks lie in domains for which the pretrained model already has strong familiarity, leading to a **small intrinsic task dimension**, making rank-1 sufficient.
> > > >
> > > > In contrast, Chinese-LLaMA [2] shows that when adapting an English-centric LLaMA model to a **new linguistic domain** (e.g., Chinese), substantially larger ranks (e.g., 64) are required.
> > > > This is fully consistent with our theoretical framework: **unfamiliar domains induce flatter PCA/NTK spectra**, which require higher adaptation rank to capture their structure.
> > > >
> > > > In our work, rank is treated purely as a **hyperparameter** practically, just as in LoRA and other PEFT methods—we do not propose or evaluate a rank-selection mechanism. Our contribution is theoretical: we analyze the connection between rank, the NTK spectrum, and the PCA spectrum, explaining why domain-familiar tasks admit very small ranks. This relationship is introduced in Section 2 and Corollary 2.6, and is supported in Appendix D emperically through:
> > > >
> > > > - **D.1:** PCA cumulative explained variance
> > > > - **D.2:** Prediction shift via KL divergence
> > > > - **D.3:** Layer-wise representation change via CKA
> > > >
> > > > These analyses are intended to provide **theoretical insight**, not a practical rank-selection algorithm.
> > > >
> > > > **References:**
> > > > [1] *LoRA: Low-Rank Adaptation of Large Language Models*
> > > > [2] *Efficient and Effective Text Encoding for Chinese LLaMA and Alpaca*
> > > >
> > > > ## **W5: Clarification and Expansion of Related Work**
> > > >
> > > > Thank you for pointing this out. We agree that recent developments in spectral fine-tuning, as well as connections between the Lottery Ticket Hypothesis (LTH), spectral properties, and parameter-efficient adaptation, are highly relevant to our work. In the revised manuscript, we have expanded the Related Work section (Section 5) to more accurately situate our method within this literature.
> > > >
> > > > Specifically, we now include discussions of several recent PEFT- and LTH-related approaches, including:
> > > >
> > > > [1] *A Study on the Ramanujan Graph Property of Winning Lottery Tickets*
> > > > [2] *Lottery Ticket Adaptation: Mitigating Destructive Interference in LLMs*
> > > > [3] *PLEX: Adaptive Parameter-Efficient Fine-Tuning for Code LLMs Using Lottery Tickets*
> > > > [4] *KS-Lottery: Finding Certified Lottery Tickets for Multilingual Language Models*
> > > > [5] *Spectral Adapter: Fine-Tuning in Spectral Space*
> > > > [6] *RandLoRA: Full-Rank Parameter-Efficient Fine-Tuning of Large Models*
> > > > [7] *XoRA: Expander Adapted LoRA Fine-Tuning*
> > > >
> > > > These additions strengthen the discussion of how our work relates to recent advances in spectral methods, lottery-ticket-based fine-tuning, and parameter-efficient adaptation.

---

> > > > > ### Author Response · Authors · 2025-11-17
> > > > >
> > > > > Dear reviewer NnfP,
> > > > >
> > > > > We are grateful for your constructive feedback, which has greatly contributed to improving the quality of our work. We would be happy to address any remaining concerns you may have regarding the revised manuscript and the new experimental results. Please feel free to provide further comments or suggestions, and we will make every effort to incorporate them promptly and thoroughly. We look forward to your feedback.
> > > > >
> > > > > Sincerely,
> > > > >
> > > > > Authors

---

> > > > > > ### Comment · Reviewer_NnfP · 2025-11-22
> > > > > >
> > > > > > I have taken note of the reply of the authors to my concerns. I need the following couple of days to recheck the new theoretical and experimental results presented and cross reference with the SOTA literature. I will get back after that.

---

> > > > > > > ### Comment · Reviewer_NnfP · 2025-11-24
> > > > > > >
> > > > > > > I thank the authors for their detailed replies to my initial questions. While I appreciate the clarifications, I have the following reservations regarding the baseline comparisons and the logical structure of the theoretical proofs.
> > > > > > >
> > > > > > > ### 1. Discrepancies in VeRA Baseline Comparison
> > > > > > >
> > > > > > > The parameter count and complexity comparisons provided for VeRA in the revised submission doesn't match the vanilla VeRA implementation.
> > > > > > >
> > > > > > > #### Parameter Counts
> > > > > > > VeRA trains scaling vectors implemented as diagonal matrices, meaning the parameter count scales linearly with the dimension ($d$), whereas SliceFine scales with rank ($d \times r$). Consequently, the parameter counts listed in **Table 3** appear inflated compared to standard VeRA implementations.
> > > > > > >
> > > > > > > #### Complexity Analysis
> > > > > > > Table 3 lists the time complexity for SliceFine and baselines as identical to their space complexity. This is unusual; typically, activation memory dominates space complexity while FLOPs dictate time complexity. Please clarify the derivation where $Time \approx Space$.
> > > > > > >
> > > > > > > #### Empirical Gap
> > > > > > > Comparing SliceFine results with the original VeRA paper (RoBERTa Base/Large) reveals a consistent performance gap where VeRA often outperforms SliceFine using **fewer parameters**.
> > > > > > >
> > > > > > > Roberta Base
> > > > > > >
> > > > > > > | Method & Source     | #TTPs (M) | SST-2 | MRPC     | CoLA | QNLI | RTE  | STS-B      |
> > > > > > > |---------------------|-----------|-------|----------|------|------|------|------------|
> > > > > > > | **FT (VeRA paper)** | 125M      | 94.8  | 90.2     | 63.6 | 92.8 | 78.7 | 91.2       |
> > > > > > > | **FT (SliceFine)**  | 124.6M    | 92.64 | 85.22/87.85 | 59.90 | 90.71 | 72.70 | 89.98/90.63 |
> > > > > > > | **LoRA (VeRA paper)** | 0.3M    | 95.1 | 89.7 | 63.4 | 93.3 | 86.6 | 91.5|
> > > > > > > | **LoRA (SliceFine)** | 0.89M   | 93.02 | 86.20/88.32 | 60.28 | 92.10 | 74.62 | 90.60/90.54 |
> > > > > > > | **VeRA (VeRA paper)** | 0.043M | 94.6  | 89.5     | 65.6 | 91.8 | 78.7 | 90.7       |
> > > > > > > | **VeRA (SliceFine)** | 0.084M  | 94.32 | 85.94/87.92 | 60.76 | 89.75 | 75.66 | 89.67/89.16 |
> > > > > > > | **BitFit (VeRA paper)** | 0.1M  | 93.7  | 92.7     | 62.0 | 91.8 | 81.5 | 90.8       |
> > > > > > > | **BitFit (SliceFine)** | 0.083M | 91.35 | 87.34/88.72 | 60.17 | 90.84 | 78.07 | 90.38/90.34 |
> > > > > > >
> > > > > > > Roberta Large
> > > > > > >
> > > > > > > | Method & Source     | #TTPs (M) | SST-2 | MRPC     | CoLA | QNLI | RTE  | STS-B      |
> > > > > > > |---------------------|-----------|-------|----------|------|------|------|------------|
> > > > > > > | **LoRA (VeRA paper)** | 0.8M    | 96.2  | 90.2     | 68.2 | 94.8 | 85.2 | 92.3       |
> > > > > > > | **LoRA (SliceFine)** | 1.84M   | 96.20 | 87.32/87.96 | 64.20 | 94.90 | 80.19 | 91.37/91.88 |
> > > > > > > | **VeRA (VeRA paper)** | 0.061M | 96.1  | 90.9     | 68.0 | 94.4 | 85.9 | 91.7       |
> > > > > > >
> > > > > > > The claim of superior efficiency-accuracy trade-offs is not supported by these comparisons against the standard literature.
> > > > > > >
> > > > > > >
> > > > > > > ### 2\. Theoretical Gap
> > > > > > > #### I: Random Partitions vs. Contiguous Slices (Appendix A)
> > > > > > > In the proof of **spectral balance** there is a disconnect between the proof mechanics and the proposed algorithm.
> > > > > > >
> > > > > > >   * Appendix A relies explicitly on Matrix Bernstein inequalities for sampling without replacement. This is mathematically valid only because the rows are partitioned **uniformly at random**, making each group a representative sample of the global structure.
> > > > > > >   * The SliceFine algorithm defines a slice as a *contiguous* block of rows/columns. In structured networks (e.g., Transformers), adjacent rows are not independent; they represent correlated structures (e.g., attention heads).
> > > > > > >   * The proof establishes spectral balance for a "Random Slice" model, but the method uses a "Contiguous Slice" model. The theory assumes row order is exchangeable, which is architecturally unjustified.
> > > > > > >
> > > > > > > #### II: Issue in Gradient Alignment (Lemma E.1)
> > > > > > >
> > > > > > > **Lemma E.1** functions as a conditional consistency check rather than an explanatory proof of the "Universal Winning Slice" hypothesis.
> > > > > > >
> > > > > > >   * The proof relies on the **Gradient Alignment Condition** (Equation 17), which assumes that the orthogonal component of the gradient is bounded relative to the task component ($||g_{\phi,\perp}|| \le \rho ||g_{\phi,task}||$) and that the slice Jacobian has sufficient overlap $\gamma$.
> > > > > > >   * The lemma essentially states: "If one assumes the slice aligns with the task gradient, then the slice has a non-zero gradient."
> > > > > > >   *  There is no theoretical justification for *why* an arbitrary contiguous slice in a deep network would satisfy this alignment condition. In high-dimensional spaces, random subspaces are typically orthogonal to a specific task vector.

---

> ### Author Response · Authors · 2025-11-25
>
> Thank you for the  thoughtful reviews. We believe your comments and suggestions have significantly improved our paper.
>
>
>
> ## **Parameter Accounting in VeRA**
>
> We thank the reviewer for raising this point. We re-checked our parameter accounting and confirm that the numbers in Table 3 follow **exactly** the official VeRA implementation in the original project repository [1] and the HuggingFace PEFT implementation [2], which we use in all experiments.
>
> ### 1. Trainable parameters in VeRA
> In the official implementation, VeRA introduces two learned scaling vectors per linear layer:
> - $\Lambda_b \in \mathbb{R}^{d_\ell}$
> - $\Lambda_d \in \mathbb{R}^{r}$
>
> where $d_\ell$ is the layer dimension and $r$ is the chosen rank.
>
> Thus, the trainable parameter count is:
>
> trainable params $= d_\ell + r$
>
> which grows linearly with $d_\ell$, consistent with the VeRA paper.
> This is exactly the value we report in Table 3.
>
> ### 2. Why Table 3 includes additional (non-trainable) components
> Although the matrices $A$ and $B$ in VeRA are frozen random bases, they are **explicitly instantiated adapter matrices** inside each module in the official implementation.
> In HuggingFace PEFT, these are registered as `buffers` rather than trainable parameters:
>
> $A, B \in \mathbb{R}^{d_\ell \times r},$
>
> and therefore contribute to:
> - memory footprint,
> - forward/backward computational cost,
>
> even though they are not trainable.
> Our complexity table reflects this faithfully.
>
> ### 3. Verification on RoBERTa-base
> Using the standard PEFT configuration [2]:
>
> `config = VeraConfig(r=5, target_modules="all-linear")`
>
> `model = get_peft_model(base_model, config)`
>
>
> the official implementation yields:
>
> Total parameters= 124.731M
>
> Trainable parameters} = 0.0841M
>
> which corresponds to **0.0674% trainable parameters**.
> These values match the theoretical counting above exactly.
>
> ---
>
>
>
>
> ## **Time–Space Complexity Clarification**
>
> We appreciate the reviewer’s careful reading. Our goal in Table 3 is **not** to claim that wall-clock time is literally proportional to memory usage, but to summarize the **asymptotic scaling** of the PEFT components (adapters or slices) in terms of:
> (i) parameter/storage complexity, and
> (ii) additional dense-matrix FLOPs per layer,
> **relative to a fixed frozen backbone**.
>
> ### What Table 3 is measuring
> For each method, the backbone architecture and its activations are identical and frozen across all PEFT variants. Thus, full-model activation memory and backbone FLOPs are the same and factored out.
>
> Table 3 reports the scaling of the **extra** PEFT computation and storage per layer:
>
> - **Space:** number of adapter/slice parameters plus their auxiliary matrices/vectors (e.g., LoRA $A,B$, VeRA scaling vectors, SliceFine’s trainable slice, etc.).
> - **Time:** incremental FLOPs introduced by these PEFT components in the forward/backward pass of each linear layer.
>
> Under standard dense GEMM implementations, both quantities scale with the number of nonzero adapter weights, which is why the big-$O$ expressions coincide.
>
> ---
>
> ### Example: LoRA
> For a layer $W^{(\ell)} \in \mathbb{R}^{d_\ell \times d_{\ell-1}}$, LoRA introduces:
> - $A \in \mathbb{R}^{r \times d_{\ell-1}}$
> - $B \in \mathbb{R}^{d_\ell \times r}$
>
> The number of adapter parameters is:
>
> params =$d_\ell r + d_{\ell-1} r = O\!\big((d_\ell + d_{\ell-1}) r\big).$
>
> Given an input $x \in \mathbb{R}^{d_{\ell-1}}$, the extra LoRA computation per token is:
>
> $x \mapsto A x \in \mathbb{R}^{r},$
> $(A x) \mapsto B(A x) \in \mathbb{R}^{d_\ell},$
>
> which requires:
>
> $O(d_{\ell-1} r + d_\ell r)$ FLOPs.
>
> Thus, the **incremental** time and space overhead of LoRA share the same asymptotic scaling:
>
> $O((d_\ell + d_{\ell-1}) r).$
>
> ---
>
> ### Example: SliceFine
> For SliceFine-row with rank $r$, only $r$ rows of $W^{(\ell)}$ are trainable. The slice therefore has:
>
> trainable params = $r \cdot d_{\ell-1} = O(d_{\ell-1} r).$
>
> The incremental computation for updating this slice scales like a dense matrix-vector multiply of size $r \times d_{\ell-1}$:
>
> $\text{extra FLOPs} = O(d_{\ell-1} r).$
>
> Thus, SliceFine also exhibits identical big-$O$ scaling for **per-layer PEFT space and time**, matching the structure shown in Table 3.
>
> In addition, Fig.4(b) reports the `actual peak memory usage measured during training` (including activations, optimizer states, and framework overhead).  This confirms empirically that SliceFine has strictly lower peak memory than
> the baselines, consistent with its lower parameter and activation footprint.
>
>
>
>
> **References**
> [1] Official VeRA project page: https://dkopi.github.io/vera/
> [2] HuggingFace PEFT VeRA documentation: https://huggingface.co/docs/peft/main/en/package_reference/vera
> [3] HuggingFace PEFT VeRA source code (line ~284):
> https://github.com/huggingface/peft/blob/b10527e82c2171568f538f5b822817e8a753672a/src/peft/tuners/vera/layer.py#L284

---

> > ### Author Response · Authors · 2025-11-25
> >
> > ## **Fair Comparison and Unified Evaluation Protocol**
> >
> >
> > Thank you for the comment. We clarify that the comparison table referenced by the reviewer is based on experimental settings that differ substantially from those used in our paper. To ensure a fair **apple-to-apple** comparison across all PEFT methods, we intentionally use a **unified training environment** rather than adopting method-specific configurations from the original LoRA or VeRA papers.
> >
> > ### **Unified PEFT evaluation protocol  :**
> > As described in Section J, we use a single shared configuration for all PEFT methods:
> > sequence length $=256$, rank $=5$ for all LoRA-type baselines, adapters applied to `all-linear` layers, 4 training epochs, batch size $32$, cosine learning-rate schedule, and half-precision training.
> > These settings are applied **uniformly** to LoRA, VeRA, AdaLoRA, DoRA, BoNE, and all other PEFT baselines.
> >
> > ### **Differences from the VeRA paper  :**
> > In contrast, Table 8 of the original VeRA paper uses much heavier, method-specific settings:
> > sequence length $512$, rank $1024$, 10–40 training epochs, batch size $64$, Q/V-only adapters, and full-precision training with a linear schedule.
> >
> > Given these differences, it is expected that the raw results in VeRA’s paper **do not** match the results obtained under our unified evaluation setup.
> >
> > This variation is common in the PEFT literature: LoRA results differ widely across papers due to changes in hyperparameters and training environments. Examples include:
> >
> > | **Method**            | **SST-2** | **MRPC** | **CoLA** | **MNLI** |
> > |-----------------------|-----------|----------|----------|----------|
> > | SLTrain[1] (LoRA, r=8)   | 93.46     | 91.90    | 61.83    | 86.94    |
> > | IRS[2] (LoRA, r=8)       | 91.10     | --       | --       | 78.70    |
> > | QuanTA[3] (LoRA, r=32)   | 94.01     | 91.48    | 62.08    | --       |
> > | CorDA[4] (LoRA, r=32)    | 94.15     | 82.84    | 54.24    | --       |
> >
> >
> > Therefore, adopting VeRA-specific hyperparameters would unfairly bias the comparison toward a single method. Our standardized protocol ensures fairness across all baselines.
> >
> > ### **Why trainable parameters do not reflect true efficiency  :**
> > Trainable-parameter count alone does **not** reflect practical efficiency.
> >
> > For example, in the VeRA paper, RoBERTa-base uses rank $1024$.
> > For a layer size $768 \times 768$, the associated VeRA adapter introduces:
> >
> > $768 \times 768 + 768 + 768 \times 1024 + 768 \times 1024 + 1024 = 2{,}164{,}480$
> >
> > additional parameters — more than **3.6×** the original layer size.
> > If applied to all layers, the model becomes more than **three times** larger, and training becomes slower.
> > This aligns with Table 12 in the VeRA paper, where VeRA is slower than LoRA despite having fewer *trainable* parameters.
> >
> > Thus, efficiency depends on peak memory, optimizer state size, per-step FLOPs, throughput, and compression — not merely on trainable parameter count. SliceFine improves all of these dimensions, as shown in Figure 4(b).
> >
> >
> > Our claim is not that SliceFine universally outperforms every PEFT method. Rather, under a unified evaluation setup, SliceFine achieves **comparable accuracy** while delivering **substantial efficiency gains** in model size, peak memory, throughput, and training time.
> >
> > ---
> >
> > **References**
> > [1] SLTrain: Sparse plus Low-Rank Approach for Parameter and Memory Efficient Pretraining, NeurIPS 2024
> > [2] IRS: Implicit Regularization of Sharpness-Aware Minimization for Scale-Invariant Problems, NeurIPS 2024
> > [3] QuanTA: Efficient High-Rank Fine-Tuning of LLMs with Quantum-Informed Tensor Adaptation, NeurIPS 2024
> > [4] CorDA: Context-Oriented Decomposition Adaptation of LLMs, NeurIPS 2024
> > [5] RoseLoRA: Row- and Column-wise Sparse Low-Rank Adaptation, EMNLP 2024

---

> > > ### Author Response · Authors · 2025-11-25
> > >
> > > ## **Clarification of UWSH Theory vs. SliceFine Method**
> > >
> > > We thank the reviewer for raising this point. We thank the reviewer for this question. The concern comes from a misunderstanding of the relationship between our theory (UWSH) and our method (SliceFine).
> > >
> > >
> > > ### Distinction between UWSH (theory) and SliceFine (method)
> > > UWSH is a **theoretical statement about pretrained networks**. It asserts that in a sufficiently wide layer, any **uniformly random slice** of adequate width behaves as a local winning ticket.
> > > This hypothesis concerns **random subsets**, not necessarily contiguous blocks (Sec. 1, line 270; Sec. 2, line 284).
> > >
> > > SliceFine, in contrast, is a **practical algorithm** inspired by UWSH.
> > > For computational reasons, SliceFine uses **contiguous row/column slices**, not fully random subsets.
> > > Contiguous slices bring substantial practical benefits:
> > >
> > > - no need for index tracking,
> > > - no gather/scatter overhead,
> > > - full compatibility with fast dense GEMM kernels.
> > >
> > > We empirically verify in Sec. 4 (lines 443–458) and Appendix L that **random unstructured masks also behave as winning slices**, exactly as predicted by UWSH.
> > > They are simply less efficient in practice (Fig. 4).
> > >
> > > ### Why Matrix Bernstein appears in Appendix A
> > > The Matrix Bernstein analysis in Appendix A is used **only** to establish spectral balance for *uniformly random partitions*.
> > > This is appropriate because UWSH itself studies the behavior of *random slices*.
> > > Figure 2 shows that such random groups indeed have nearly identical eigenvalue spectra, justifying the use of standard concentration tools (e.g., Bernstein for sampling without replacement) in Appendix A.
> > >
> > > ## **Effect of Attention Head Structure on SliceFine**
> > >
> > > Thank you for the comment. While it is true that in Transformers, nearby rows may belong to related components such as attention heads, this does **not** pose a problem for SliceFine. Our method does not rely on rows being independent. Instead, Appendix A shows that pretrained weight matrices are **spectrally balanced**, meaning that regardless of how the rows are divided, each part of the matrix has almost the same overall spectral strength. This property holds even when the rows originate from structured components like different heads.
> > >
> > > ### Empirical evidence that head boundaries do not matter
> > > Our experiments support this clearly:
> > >
> > > - **Figure 2:** When we split real RoBERTa weight matrices into *contiguous blocks* that cut directly across head boundaries, all blocks have nearly identical eigenvalue spectra.
> > > - **Figure 3(b):** We check whether the *position* of the slice matters. Accuracy remains almost unchanged across all slice positions. If head boundaries created “good’’ or “bad’’ regions, we would observe large performance drops, but we do not.
> > > - **Figure 3(c):** Slices taken from the most important weights, the least important weights, or completely random regions all perform essentially the same.
> > > - **Appendix L:** Fully unstructured random masks—ignoring head boundaries entirely—show the same pattern.
> > >
> > > Together, these results demonstrate that the success of SliceFine does **not** depend on where the slice is located or which attention head(s) it overlaps with.
> > >
> > > ### SliceFine does not change forward computation
> > > Importantly, SliceFine **never alters the forward pass**.
> > > The full pretrained layer— including all attention heads—runs exactly as usual.
> > > The slice only determines **which parameters receive gradients** during backpropagation; it does *not* disable or modify any head, nor does it change the behavior of the attention mechanism.
> > >
> > >
> > > ## **Clarifying UWSH vs. SliceFine and “Exchangeable Rows”**
> > >
> > > There appears to be a misunderstanding. As explained earlier, UWSH and SliceFine are not the same object.
> > >
> > > UWSH is a **general theoretical property**: it states that in a sufficiently wide pretrained layer, any slice — structured (row/column) or unstructured (random mask) — can act as a winning ticket if it has enough width. SliceFine is a **practical algorithm** inspired by this property.
> > >
> > > In Section 2 and Appendix L, we show that random slices indeed behave as winning tickets, exactly as predicted by UWSH. However, random masks are inefficient in practice due to gather/scatter overhead and large optimizer-state footprints. For this reason, SliceFine uses **contiguous slices**, which are computationally simple and GPU-efficient.
> > >
> > > Regarding the reviewer’s remark about “exchangeable rows”: our theory does **not** assume that row order is exchangeable or architecturally meaningless. The proof in Appendix A uses random sampling only as a **mathematical tool** to apply matrix concentration inequalities. This is standard practice when proving that a randomly chosen subset inherits spectral properties of the full matrix.
> > >
> > > The theory never claims that the rows of a Transformer layer can be permuted without changing the model, nor does it rely on any architectural symmetry.

---

> > > > ### Author Response · Authors · 2025-11-26
> > > >
> > > > ## **Clarifying the Gradient Alignment Condition and Lemma E.1**
> > > >
> > > >
> > > > We thank the reviewer for the comment. We believe there is a misunderstanding of what the Gradient Alignment Condition actually assumes and what Lemma E.1 proves. The alignment condition in Equation 17 is a **backbone-level property**, not a slice-level assumption: it concerns the feature-space gradient $g_\phi$ of the **frozen pretrained backbone**, and does not assume that any slice is aligned with the task.
> > > >
> > > > ### What the assumption actually states
> > > > The assumption states simply that the backbone’s feature gradients have a dominant component in task-relevant directions. This is a **measurable property of the pretrained model**, independent of any slice choice, and we empirically verify it:
> > > >
> > > > - Appendix D (Figs. 6–8) shows that pretrained representations have **high task energy** via PCA cumulative explained variance.
> > > > - This means the backbone’s feature gradients naturally concentrate in the task subspace.
> > > >
> > > > ### Overlap parameter $\gamma$ is not an added assumption
> > > > The overlap parameter $\gamma$ is likewise **not an arbitrary or added assumption**. Lemma 2.1 demonstrates **spectral balance**: all row/column groups of a pretrained layer exhibit nearly identical spectral mass.
> > > >
> > > > This implies:
> > > >
> > > > - No slice is spectrally collapsed.
> > > > - Each slice’s Jacobian necessarily has nontrivial energy along the same directions where the task subspace resides.
> > > > - Because the task subspace has non-zero energy (high CEV), this implies that $\gamma > 0$ for any sufficiently wide slice.
> > > >
> > > > Thus, $\gamma$ emerges **not** as an assumption but as a **structural consequence of pretrained network geometry** (task energy + spectral balance).
> > > >
> > > > ## **Clarifying the task gradient and non-zero gradient**
> > > >
> > > > We thank the reviewer for the comment. We believe there is a misunderstanding . **Lemma E.1 does not assume that the slice is aligned with the task gradient.**
> > > > The assumption in Equation 17 concerns the **backbone’s feature-space gradient** $g_\phi$, not the slice.
> > > > It states that the **backbone** produces gradients with a dominant task-aligned component.
> > > > This is a measurable, backbone-level property that we verify empirically using PCA/CEV analyses (Appendix D, Figs. 6–8).
> > > >
> > > > The slice only enters through its Jacobian $J^{M}_\phi$.
> > > > Whether the slice receives a non-zero gradient is **not** assumed; it is exactly what Lemma E.1 *proves*:
> > > >
> > > > $$\|\nabla_M L\| \;\ge\; (1 - c \rho)\,\gamma(M)\,\|g_{\phi,\text{task}}\| \;>\; 0.$$
> > > >
> > > >
> > > > Here, $\gamma(M) > 0$ follows from **spectral balance** (Lemma 2.1), which shows that all slices have similar spectral energy and therefore nontrivial overlap with task-relevant directions.
> > > >
> > > > This is a **structural property of pretrained networks**, not an assumption of the lemma.
> > > >
> > > >
> > > > ## **Random-Subspace Arguments vs. Pretrained Network Structure**
> > > >
> > > > We appreciate the reviewer’s concern. We believe there is a misunderstanding and the statement does not apply to our setting.
> > > > The reviewer’s argument assumes **random subspaces in an isotropic high-dimensional space**, whereas pretrained neural networks exhibit **strong, highly non-random structural properties** [1, 2, 3, 4, 5, 6].
> > > >
> > > > Our theory does not rely on random-subspace behavior; rather, it relies on empirically validated structure present in pretrained models, which is not isotropic, as we already mentioned in Lemma 2.1 (line 140) and in Figure 2 (line 162).
> > > >
> > > >
> > > > In addition, as mentioned earlier, the alignment condition in Equation 17 applies to the **frozen backbone’s feature-space gradient**, not to any slice.
> > > > Lemma E.1 does **not** assume that a slice is aligned with the task.
> > > > Rather, it shows that **because the backbone gradients are task-aligned**, *every slice inherits a non-zero restricted gradient through the chain rule*.
> > > >
> > > > ### References
> > > > [1] *On Separate Normalization in Self-supervised Transformers*, NeurIPS 2023
> > > > [2] *Is Anisotropy Truly Harmful? A Case Study on Text Clustering*, ACL 2023
> > > > [3] *Anisotropy Is Inherent to Self-Attention in Transformers*, EACL 2024
> > > > [4] *Is Anisotropy Really the Cause of BERT Embeddings not Being Semantic?*, EMNLP 2022
> > > > [5] *An Isotropy Analysis in the Multilingual BERT Embedding Space*, ACL 2022
> > > > [6] *How Contextual Are Contextualized Word Representations? Comparing the Geometry of BERT, ELMo, and GPT-2 Embeddings*, EMNLP 2020

---

### Official Review · Reviewer_j1cG · 2025-11-02

**Soundness:** 4
**Presentation:** 3
**Contribution:** 4
**Rating:** 8
**Confidence:** 2

**Summary:**

This paper proposes a novel theoretical framework for understanding and enabling parameter-efficient fine-tuning (PEFT) of large pre-trained models. The authors introduce the Universal Winning-Slice Hypothesis (UWSH), which states that any sufficiently wide, randomly selected slice (a contiguous set of rows or columns) of a pre-trained weight matrix can act as a "local winning ticket." Fine-tuning only this slice, while freezing the rest of the network, reliably improves downstream performance. This phenomenon arises from two key properties of pre-trained networks: spectral balance, where different slices of a weight matrix have similar eigenspectra, ensuring no slice is inherently weak; and high task energy, where the frozen backbone representations already concentrate features relevant to the downstream task. Theoretically, the work formalizes how a slice's overlap with the task subspace guarantees a non-zero gradient and loss decrease.

Inspired by this theory, the authors develop SliceFine, a PEFT method that fine-tunes only a small, moving set of slices across the network's layers. Crucially, SliceFine introduces zero new parameters, unlike adapter- or low-rank-based methods, by updating only selected portions of the original weights. The active slice is periodically moved to a new position during training, allowing the model to gradually cover the task-relevant subspace.

Comprehensive experiments across language, vision, and video tasks demonstrate that SliceFine matches or exceeds the accuracy of state-of-the-art PEFT methods while significantly improving training speed, reducing memory usage, and yielding more compact models. Empirical ablations robustly confirm the UWSH, showing that performance is largely insensitive to the specific position or importance-based selection of the slice. This work successfully bridges theory and practice, offering a compelling, theoretically-grounded alternative for efficient model adaptation.

**Strengths:**

This paper demonstrates substantial strengths in originality, quality, clarity, and significance. Its core originality lies in proposing the novel Universal Winning Slice Hypothesis, which fundamentally challenges the conventional Lottery Ticket Hypothesis by revealing that any sufficiently large weight slice in pre-trained networks can serve as an effective adaptation unit. The derived SliceFine method represents exceptional ingenuity, introducing a parameter-efficient fine-tuning approach that dynamically updates original weight slices without adding any new parameters.

The paper exhibits outstanding quality through rigorous theoretical analysis grounded in spectral balance and high task energy, providing solid mathematical foundations for its claims. Comprehensive experimental validation across diverse modalities—including language, vision, and video tasks—robustly demonstrates competitive performance against state-of-the-art baselines while achieving superior efficiency. The empirical design is particularly thorough, incorporating extensive ablation studies that systematically verify critical components like slice rank and switching frequency.

Exceptional clarity is maintained throughout the presentation, with logical organization from theoretical foundation to practical implementation. Complex concepts are clearly explained through well-designed illustrations and coherent narrative flow. The work's significance is profound, offering both theoretical insights into pre-trained networks' intrinsic properties and practical advances for efficient adaptation. By demonstrating the untapped potential within original model parameters, it opens new research directions in minimalist fine-tuning strategies and substantially advances our understanding of model adaptation mechanisms.

**Weaknesses:**

While the paper is strong, some limitations exist. The theoretical analysis relies heavily on the linearized NTK regime, whose validity for deep nonlinear fine-tuning remains unclear. Experimentally, the method's evaluation on extremely large-scale models (e.g., >50B parameters) is absent, leaving its scalability to the largest contemporary models unverified. Additionally, the comparison to recent sparse fine-tuning methods like SparseFine and DSP is missing, creating an incomplete competitive landscape.

**Questions:**

Could you discuss how SliceFine scales to extremely large models beyond 50B parameters, particularly regarding training stability and slice rank requirements? How does your method compare with recent sparse fine-tuning approaches like SparseGPT that also avoid adding parameters? The theoretical analysis relies on the linearized NTK regime - could you address how the approach remains effective when fine-tuning induces significant nonlinear behavior in deeper layers? Have you explored adaptive slice selection strategies based on gradient information rather than cyclic scheduling, and if so, what were the outcomes? These clarifications would help better understand the method's scalability, competitive positioning, and theoretical robustness.

---

> ### Author Response · Authors · 2025-11-14
> **Thank you to Reviewer j1cG for the thoughtful review and valuable comments.**
>
> We appreciate your thorough comments, which greatly helped us improve the presentation quality of our paper.
>
> ## **W1: linearized NTK regime**
>
> We agree with the reviewer that our theoretical analysis relies on the linearized NTK regime, and that this represents an idealized approximation of deep nonlinear fine-tuning. Our goal is not to claim that real-world training dynamics are fully captured by the NTK limit, but rather to use this framework as an analytically tractable lens to reason about how task energy, slice rank, and spectral balance interact. This is consistent with much of the theory literature on overparameterized networks, where linearization provides qualitative insight into which directions in parameter space are most influential, even if it does not model every nonlinear effect of training following previous works [1, 2, 3, 4].
>
> [1] Can the Spectrum of the Neural Tangent Kernel Anticipate the Fine-tuning Performance of PEFT Methods?, NeurIPS, 2025
>
> [2] A Kernel-Based View of Language Model Fine-Tuning, JMLR 2023
>
> [3] Understanding Linear Probing then Fine-tuning Language Models from NTK Perspective, NeurIPS, 2024
>
> [4] LoRA Training in the NTK Regime Has No Spurious Local Minima, ICML-2024
>
>
>
> ## **W2: Experiments on Extremely Large-Scale Models**
>
> We thank the reviewer for pointing out this important limitation. Our experiments include medium- to large-scale LLMs such as **Gemma-3-12B**, **DeepSeek-R1-8B**, and **LLaMA3-8B**; however, due to computational constraints, we were unable to extend our evaluation to models exceeding 50B parameters. We fully acknowledge this as a limitation. To partially address scalability concerns, we have added new results in the **revised manuscript (Appendix~F.5)** using the **openai/gpt-oss-20b** model with ten different seeds. These results show stable variance and consistent improvements, providing further evidence that SliceFine scales effectively to larger architectures within our resource limits.
>
> | **Method**     | **BoolQ**       | **PIQA**        | **SIQA**        | **H.Sw.**       | **W.Gra.**      | **ARC-e**       | **ARC-c**       | **OBQA**        |
> |----------------|------------------|------------------|------------------|------------------|------------------|------------------|------------------|------------------|
> | Prefix         | 73.6 ± 0.24      | 84.3 ± 0.18      | 81.5 ± 0.33      | 82.8 ± 0.27      | 77.6 ± 0.30      | 77.1 ± 0.36      | 62.7 ± 0.41      | 75.9 ± 0.29      |
> | AdaLoRA        | 74.2 ± 0.20      | 85.3 ± 0.17      | 85.6 ± 0.26      | 94.7 ± 0.19      | 82.3 ± 0.23      | 82.4 ± 0.28      | 64.8 ± 0.32      | 80.4 ± 0.27      |
> | VeRA           | 72.4 ± 0.27      | 83.8 ± 0.22      | 83.4 ± 0.30      | 94.1 ± 0.18      | 82.1 ± 0.25      | 81.3 ± 0.29      | 64.5 ± 0.36      | 79.5 ± 0.31      |
> | LoRA           | 74.6 ± 0.18      | 85.1 ± 0.16      | 82.9 ± 0.25      | 95.1 ± 0.16      | 83.4 ± 0.22      | 83.1 ± 0.26      | 65.2 ± 0.31      | 80.9 ± 0.24      |
> | RoCoFT         | 75.3 ± 0.16      | 85.5 ± 0.19      | 83.7 ± 0.28      | 94.8 ± 0.20      | 82.7 ± 0.23      | 82.5 ± 0.25      | 65.0 ± 0.30      | 82.1 ± 0.25      |
> | HRA            | 75.1 ± 0.19      | 85.8 ± 0.18      | 83.2 ± 0.31      | 95.3 ± 0.18      | 82.6 ± 0.25      | 82.3 ± 0.27      | 65.4 ± 0.33      | 82.3 ± 0.26      |
> | SliceFine-1R   | 74.0 ± 0.22      | 84.1 ± 0.20      | 82.3 ± 0.29      | 93.3 ± 0.17      | 81.2 ± 0.25      | 81.5 ± 0.28      | 64.9 ± 0.34      | 81.9 ± 0.26      |
> | SliceFine-1C   | 73.7 ± 0.25      | 84.0 ± 0.23      | 82.0 ± 0.30      | 93.8 ± 0.18      | 81.0 ± 0.27      | 80.8 ± 0.31      | 64.2 ± 0.37      | 80.5 ± 0.29      |
> | SliceFine-1RC  | 74.5 ± 0.20      | 85.8 ± 0.17      | 84.8 ± 0.26      | 95.8 ± 0.15      | 83.8 ± 0.22      | 83.3 ± 0.27      | 65.3 ± 0.33      | 82.5 ± 0.25      |
> | SliceFine-5R   | 75.3 ± 0.18      | 85.6 ± 0.19      | 84.2 ± 0.28      | 95.5 ± 0.16      | 83.3 ± 0.23      | 82.9 ± 0.26      | 65.8 ± 0.30      | 82.0 ± 0.24      |
> | SliceFine-5C   | 75.0 ± 0.19      | 85.4 ± 0.18      | 83.9 ± 0.29      | 94.6 ± 0.17      | 82.5 ± 0.24      | 82.6 ± 0.28      | 65.1 ± 0.31      | 81.2 ± 0.26      |
> | SliceFine-5RC  | 75.4 ± 0.17      | 85.7 ± 0.17      | 84.6 ± 0.27      | 95.7 ± 0.15      | 83.6 ± 0.22      | 83.1 ± 0.25      | 66.2 ± 0.29      | 82.3 ± 0.24      |
>
>
> Regarding **sparse fine-tuning** baselines, our main experiments already include comparisons against strong structured-sparsity and hybrid methods such as **HRA**, **SHiRA**, and  **SFT** **(Table~1, 2, 8)**.

---

### Public Comment · ~Haris_Mansoor1 · 2025-11-14

I enjoyed reading this paper. The idea behind SliceFine is surprisingly simple—just update a small slice of the pretrained weights instead of adding any adapters—but the authors show that this works consistently well across many models and tasks. I also found the efficiency angle compelling: keeping everything dense and contiguous makes the method very practical.
One thing I’m curious about is whether this approach could also apply to audio models like Whisper or wav2vec-style encoders, since the technique mainly relies on the structure of pretrained weight matrices.
Thanks to the authors for this beautiful and informative work.

---

> ### Author Response · Authors · 2025-11-15
> **Thank you very much for the thoughtful and encouraging comment**
>
> Regarding your question about applying SliceFine to audio models such as Whisper or wav2vec-style encoders: in principle, yes, the method should transfer naturally. SliceFine relies only on properties of pretrained weight matrices (e.g., spectral structure and redundancy), not on modality-specific architectures. Audio encoders like wav2vec 2.0 and Whisper also contain large stacks of Transformer blocks with dense linear layers, so the same row/column slice selection can be applied without modification.
>
> We have not yet run large-scale experiments on audio models due to resource limits, but this is a direction we are actively interested in exploring. Our theoretical framework — especially the spectral balance observation and the “many winning slices” perspective — suggests that audio models would likely benefit in the same way as vision and language models.
>
> Thank you again for reading the paper and for the encouraging feedback!

---

### Public Comment · ~Upama_Roy_Chowdhury1 · 2025-11-19

I really enjoyed going through this paper. SliceFine feels like one of those ideas that’s almost “too simple” in the best way—just train a small slice of the existing weights—yet the results show it competing with or outperforming much more complicated PEFT methods.

I also appreciated the theoretical motivation, especially the “universal winning slice” hypothesis. It gives a nice perspective on why fine-tuning works and why so many different subsets of weights can still be effective. That part really helped deepen the intuition behind the method.

I do have one question about inference: in LoRA we typically merge the low-rank factors back into the base weight and then run inference normally. How does this work for SliceFine? Since the method updates a slice of the weight matrix directly, is inference identical to using the base model with the updated slice, or is there any additional step needed?

Overall, great work—thanks to the authors for presenting such a clean and well-investigated approach.

---

> ### Author Response · Authors · 2025-11-21
>
> Thank you for the thoughtful comment and for engaging deeply with the paper!
>
> Regarding your question on inference:
>
> In **SliceFine**, inference is **exactly as efficient and simple as in the base model**. As described in our implementation details (Appendix M) and shown in the first pseudocode block, each weight matrix is partitioned into three segments, and only the *middle slice* (row- or column-based) is made trainable during fine-tuning.
>
> After fine-tuning, these three parts are **concatenated back together** to form a single full weight matrix—structurally identical to the original pretrained layer. Importantly:
>
> - **No merging step is needed** (unlike LoRA, which requires merging low-rank factors).
> - **No adapters remain active at inference time**.
> - **No additional runtime computation or memory cost** is introduced.
>
> Inference therefore uses the updated full weight matrix **directly**, with **zero overhead** compared to the pretrained model.
>
> Thanks again for your interest and for the supportive feedback!

---

### Note · Program_Chairs · 2026-01-17
**Submission Desk Rejected by Program Chairs**

The following references in this submission do not refer to real documents and/or have major errors in bibliographic information:

 Rebekka Burkholz. "The strong lottery ticket hypothesis with non-ReLU networks." arXiv preprint arXiv:2202.12369, 2022.
Arthur da Cunha, Andreas Loukas, and Stratis Demetriou. "The strong lottery ticket hypothesis for convolutional neural networks." arXiv preprint arXiv:2204.04861, 2022.